# Multimarginal flow matching with optimal transport potentials

**Raghav Kansal** [1]   **David Crair** [1]   **Nghia Nguyen** [1]   **Scott Pope** [1]   **Bradley Parry** [1]

## Abstract

Flow matching (FM) has emerged as a powerful framework for learning dynamic transport maps between two empirical distributions. However, less explored is the setting with intermediate observed marginals that can help constrain the flows between the endpoints. This "multimarginal" regime is central to modeling temporal evolution in dynamical systems in many scientific domains that can sample sequential distributions. We tackle this problem with a novel approach that leverages the connection between FM and dynamic optimal transport (OT), softly steering the flow towards the intermediate marginals through potential terms in the dynamic OT action. By extending the conditional FM learning target to incorporate these potentials, we derive an efficient, simulation-free algorithm for multimarginal FM that offers considerable flexibility in the spatiotemporal dynamics of the learned flows. We demonstrate state-of-the-art performance and training efficiency of OT-potential FM (OTP-FM) on diverse single-cell RNA sequencing, oceanographic, and meteorological datasets. Our code is available at https://github.com/Bexorg-Inc/OTP-FM.

## 1. Introduction

Understanding the complex nonlinear dynamics of physical systems is of central importance in many scientific domains, including transcriptomic state transitions in developmental biology, disease progression in neurodegenerative diseases, and climate modeling. There has been significant recent progress in these disciplines towards developing a corpus of static snapshots of these systems during their evolution, such as with longitudinal single-cell RNA sequencing (scRNA-seq) measurements; however, inferring accurate per-sample trajectories from these, often independent, snapshots remains a critical challenge for mechanistic understanding, therapeutic target identification, and predictive inference.

*Conditional flow matching* (CFM) (Lipman et al., 2023; Tong et al., 2024; Liu et al., 2023; Albergo et al., 2025) has emerged as a leading framework for this problem, learning continuous-time transport maps through efficient, simulation-free regression of simple conditional trajectories, which are commonly conditional optimal transport (OT) solutions between paired source and target samples. When intermediate marginals are available, the natural extension has been to apply CFM *piecewise* between consecutive marginals, stitching conditional paths end-to-end (Fig. 1). However, this produces trajectories with unphysical discontinuities at each marginal boundary. Recent multimarginal methods such as MMFM (Rohbeck et al., 2025) and 3MSBM (Theodoropoulos et al., 2026) have tried to smooth these, but with prescriptive, ad-hoc strategies that we argue need not describe the physical system.

We propose a principled relaxation: we show that piecewise CFM corresponds to conditional dynamic OT with *hard constraints* on each intermediate marginal, and relax these to smooth, finite-strength potential energy terms in the dynamic OT action with OT-potential FM (OTP-FM)—recovering piecewise CFM as a limiting case. Our contributions are:

- **A multimarginal generalization of dynamic OT** with exact conditional solutions and theoretical bound on alignment to the ground-truth intermediate marginals, controlled by the potential strength and training loss.

- **A flexible, simulation-free training algorithm** with a broad design space in the potential parameters, effectively allowing the data to determine the interpolated dynamics through optimization over this space rather than the prescriptive approaches of prior methods.

- **State-of-the-art (SOTA) performance and training efficiency** on diverse scientific datasets, with systematic ablations and concrete recommendations for applying OTP-FM to new datasets.

---

[1]Bexorg, Inc., New Haven, CT, USA. Correspondence to: Raghav Kansal <raghav.kansal@bexorg.com>, Bradley Parry <brad.parry@bexorg.com>.

*Proceedings of the 43rd International Conference on Machine Learning*, Seoul, South Korea. PMLR 306, 2026. Copyright 2026 by the author(s).

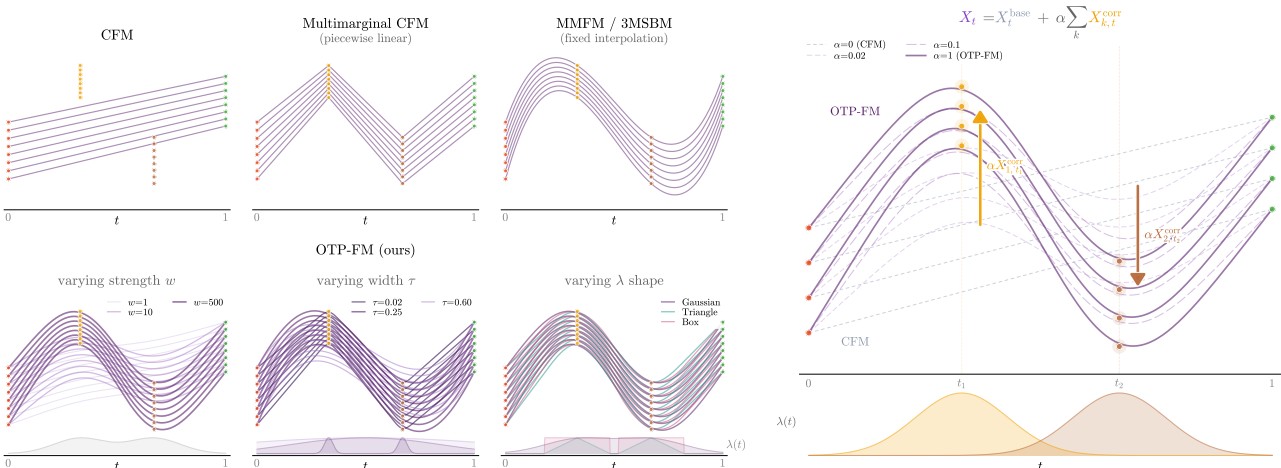

*Figure 1.* **(Left)** Comparing standard CFM — straight-line trajectories ignoring intermediate marginals; multimarginal CFM — stitching CFM trajectories piecewise between consecutive marginals; prescriptive approaches such as MMFM and 3MSBM — using fixed interpolation strategies to smooth kinks; and OTP-FM, whose soft potential-driven dynamics with tunable strength $w$, temporal width $\tau$, and $\lambda$ shape yields smooth *and* flexible trajectories. **(Right)** Method overview: trajectories $X_t$ are decomposed as a base CFM path $X_t^{\text{base}}$ plus marginal-driven corrections $X_{k,t}^{\text{corr}}$ that are gradually scaled by the curriculum parameter $\alpha$ to converge to the OTP-FM solution.

## 2. Background and preliminaries

### 2.1. Optimal transport

The original static Monge OT problem solves for a transport map $\psi \colon \mathbb{R}^d \to \mathbb{R}^d$ between source and target measures $\mu_0$ and $\mu_1$, such that the push-forward operation $\psi_\# \mu_0 = \mu_1$, that is *optimal* with respect to a cost $c \colon \mathbb{R}^d \times \mathbb{R}^d \to \mathbb{R}$ (Villani, 2009): $\mathcal{L}_{\text{OT}} := \min_{\psi:\psi_\#\mu_0=\mu_1} \int c(x, \psi(x)) \mathrm{d}\mu_0(x)$. For cost $c_p(x, y) = \|x - y\|^p$, $\mathcal{L}_{\text{OT}}^{1/p}$ is the Wasserstein $p$-distance $\mathcal{W}_p$.

Particularly relevant is the *dynamic OT* (DOT) formulation (Benamou & Brenier, 2000), where we define a dynamic probability path $\rho_t \colon [0, 1] \times \mathbb{R}^d \to \mathbb{R}^+$ interpolating $\mu_0$ to $\mu_1$. This path is generated by a velocity field $u_t$ via the continuity equation $\partial_t \rho = -\nabla \cdot (\rho u)$, with sample trajectories satisfying $\dot{X}_t = u_t(X_t)$ for $X_0 \sim \rho_0$, and the corresponding map, or *flow*, $\psi_t \colon [0, 1] \times \mathbb{R}^d \to \mathbb{R}^d$ transports the samples along the trajectory $X_t = \psi_t(X_0) \sim \rho_t$. The objective for squared Euclidean cost $c_2/2$ is:[1]

$$\mathcal{L}_{\text{DOT}} := \min_{\rho_t, u_t} \int_0^1 \int_{\mathbb{R}^d} \frac{1}{2} \|u_t(x)\|^2 \mathrm{d}\rho_t(x) \mathrm{d}t, \quad (1)$$

subject to the continuity equation and boundary conditions $\rho_0 = \mu_0$, $\rho_1 = \mu_1$.

Interestingly, this can be interpreted as minimizing the action $S[L]$ of a fluid with Lagrangian $L = T - V$, kinetic energy $T = \frac{1}{2}\rho\|u\|^2$, and potential energy $V = 0$:

---

[1] We adopt the conventional factor of $1/2$ to maintain the analogy with physical kinetic energy.

$$S[L] = \int_0^1 \int_{\mathbb{R}^d} \Big[ \frac{1}{2}\rho_t(x)\|u_t(x)\|^2 - V(x, t) +$$
$$\varphi_t(x)[\partial_t \rho_t(x) + \nabla \cdot (\rho_t(x) u_t(x))] \Big] \mathrm{d}x\mathrm{d}t, \quad (2)$$

where $\varphi$ is a Lagrange multiplier enforcing the continuity equation. Minimizers for $V = 0$ satisfy the Euler-Lagrange (E-L) equations:

$$u_t = \nabla \varphi_t, \qquad \partial_t \varphi_t(x) + \frac{\|\nabla \varphi_t(x)\|^2}{2} = 0, \quad (3)$$

and follow straight-line trajectories $X_t = (1 - t)X_0 + t\psi(X_0)$, $\dot{X}_t = \psi(X_0) - X_0$, where $\psi$ is the static OT map for cost $c_2$. A detailed derivation is provided in App. A.1. These straight-line minimizers correspond to the conditional paths commonly used in CFM, as we describe next, while in OTP-FM we explore the case of non-zero $V$.

### 2.2. Conditional flow matching

Conditional flow matching (CFM) similarly aims to find a flow between measures by learning a parametric velocity $u_t^\theta$. Obtaining a valid *marginal* training target to which to regress this is often intractable; however, Lipman et al. (2023); Tong et al. (2024) show that we can derive an equivalent, simpler *conditional* target by constructing the marginal $\rho_t(x) = \int \rho_t(x|z) q(z) \mathrm{d}z$ as a mixture of conditional paths $\rho_t(x|z)$, with associated conditional velocity $u_t(x|z)$, conditioned on a latent variable $z \sim q(z)$, and satisfying $\rho_0(x) = \mu_0(x)$, $\rho_1(x) = \mu_1(x)$.

The simplest and most common way to do so is choosing $z = (x_0, x_1) \sim \pi(x_0, x_1)$, where $\pi$ is some joint distribution of the endpoints, and $u_t(x|z)$ the *conditional dynamic*

*OT solution* (for $V = 0$) between $\rho_0(x|z) = \delta(x - x_0)$ and $\rho_1(x|z) = \delta(x - x_1)$, defining the CFM objective:

$$X_t(x|z) = (1-t)x_0 + tx_1, \qquad (4)$$

$$u_t(x|z) = \dot{X}_t(x|z) = x_1 - x_0, \qquad (5)$$

$$\mathcal{L}_{\text{CFM}}(\theta) := \mathbb{E}_{t,z,x\sim\rho_t(x|z)} \left\| u^\theta(t,x) - u_t(x|z) \right\|^2. \quad (6)$$

Perhaps surprisingly, this far simpler objective is gradient-equivalent to the marginal FM objective, thereby providing an efficient and scalable training algorithm for learning flows. In OTP-FM, we generalize this regression target to non-zero potentials $V$ that flexibly incorporate intermediate marginal constraints.

## 2.3. Few-step and consistency models

As we will describe in Sec. 4.1, OTP-FM can require evaluating sample positions $X_{t_k}$ at intermediate times during training. To avoid costly ODE simulation for this, we train a *consistency model* for few-step inference. Broadly, consistency models aim to learn (variations of) the *flow map* $\Psi_{t_1,t_2} : [0,1]^2 \times \mathbb{R}^d \to \mathbb{R}^d$ — a generalization of $\psi_t$ that can transport samples between two arbitrary time points $t_1$ and $t_2$: $\Psi_{t_1,t_2}(X_{t_1}) = X_{t_2}$, with boundary condition $\Psi_{t,t}(x) = x$ (Boffi et al., 2025; 2026).

While OTP-FM is agnostic to the particular training procedure used to learn $\Psi_{t_1,t_2}$, for our experiments we primarily employ that of *improved MeanFlow* (iMF) (Geng et al., 2026b), which, at the time of writing, is SOTA in one- and two-step inference. Namely, we parameterize $\Psi_{t_1,t_2}$ in terms of the *mean velocity* between $t_1$ and $t_2$, $v_{t_1,t_2}(X_{t_1}) = \frac{1}{t_2-t_1} \int_{t_1}^{t_2} u_t(X_t)\mathrm{d}t$, from which the regression target is derived:

$$\mathcal{L}_{\text{iMF}}(\theta) := \mathbb{E}_{t_1,t_2,z,x\sim\rho_{t_1}(\cdot|z)} \left\| V_{t_1,t_2}^\theta(x) - u_{t_1}(x|z) \right\|^2 \quad (7)$$

$$V_{t_1,t_2}^\theta(x) \equiv v^\theta(x) - (t_2 - t_1) \, \text{sg}\left[v_{t_1,t_1}^\theta(x)\partial_x v^\theta + \partial_{t_1} v^\theta\right],$$

where sg is the stop-gradient operator, $u_t(x|z)$ is the *instantaneous* conditional velocity target used in CFM, and $V_{t_1,t_2}^\theta$ is parametrization of the model $v_{t_1,t_2}^\theta$ that directly regresses this target.[2] Further discussion of consistency methods and a derivation of our iMF objective appear in App. C, and results with alternatives in App. G, demonstrating OTP-FM's flexibility to the choice of consistency model.

## 3. The dynamic OTP problem

We now generalize CFM to the multimarginal trajectory inference problem: given empirical marginals $\{\mu_{t_k}\}_{k=0}^{K+1}$ (two endpoint and $K$ intermediate) at times $t_k \in [0, 1]$, we aim

---

[2] Our formulation is largely equivalent to the original of Geng et al. (2026b) except for a modification of the training target to flow *forward* in time. Details are provided in App. C.2.

---

to learn a velocity field $u_t^\theta$ whose flow induces densities $\rho_t$ aligned with $\mu_{t_k}$ while describing physically plausible interpolated trajectories, as measured by alignment to held-out marginals (Sec. 6). We first observe that standard piecewise CFM targets can be recast as conditional solutions of dynamic OT with *hard* penalty terms, or singular potentials (Sec. 3.1). This motivates a smooth relaxation to what we call the dynamic OT + potentials (OTP) problem (Sec. 3.2), in which the intermediate marginal constraints appear instead as *soft* potential terms in the OT action. Finally, we derive sample trajectories and discuss the design space of potentials (Secs. 3.3 and 3.4).

## 3.1. Piecewise CFM as OT with singular potentials

The standard multimarginal extension of CFM stitches conditional OT solutions piecewise between consecutive marginals (e.g. Tong et al. (2024; 2023), see App. A.2), whose target densities satisfy $\rho_{t_k} = \mu_{t_k}$ exactly at each intermediate time. This is equivalent to imposing *hard constraints* (HC) on the conditional dynamic OT problem (Eq. 1), to whose solutions we regress:

$$\mathcal{L}_{\text{HC}} := \min_{\rho_t, u_t} \int_0^1 \int_{\mathbb{R}^d} \left[\tfrac{1}{2}\rho_t\|u_t\|^2 + \varphi_t[\partial_t\rho_t + \nabla\cdot(\rho_t u_t)]\right]\mathrm{d}x\mathrm{d}t,$$

$$\text{s.t. } \rho_{t_k} = \mu_{t_k} \; \forall k. \quad (8)$$

Because the kinetic-energy integral is additive in time and the constraints fix the densities at each $t_k$, this problem decouples across consecutive intervals into independent dynamic OT problems between $\mu_{t_k}$ and $\mu_{t_{k+1}}$ (Lemma A.3 in App. A.2), recovering piecewise CFM as its conditional solution (Lemma A.4).

Equivalently, these constraints can be enforced by adding singular penalties based on a metric statistical distance $\mathcal{D}$ such as $\mathcal{W}_2^2$, time-localized with Dirac deltas, and taking the penalty strength $w_k \to \infty$:

$$\mathcal{L}_{\text{HC}} := \min \int_0^1 \int_{\mathbb{R}^d} \Big[\tfrac{1}{2}\rho_t\|u_t\|^2 + \overbrace{\sum_{k=1}^{K} w_k \, \delta(t - t_k) \, \mathcal{D}_t^k \rho_t}^{\equiv -V_{\text{HC}}(x,t)}$$

$$+ \varphi_t[\partial_t\rho_t + \nabla \cdot (\rho_t u_t)]\Big]\mathrm{d}x\mathrm{d}t, \quad (9)$$

where $\mathcal{D}_t^k \equiv \mathcal{D}[\rho_t, \mu_{t_k}] : \mathcal{P}(\mathbb{R}^d) \times \mathcal{P}(\mathbb{R}^d) \to \mathbb{R}^+$ is the statistical distance between $\rho_t$ and $\mu_{t_k}$. The limit $w_k \to \infty$ enforces $\mathcal{D}_t^k = 0 \Leftrightarrow \rho_{t_k} = \mu_{t_k}$ (since $\mathcal{D}_t^k$ is a metric), recovering the hard constraint (Prop. 3.1). Thus, we can interpret the penalties as singular potential terms in the action.

## 3.2. The OTP problem: a smooth relaxation

This viewpoint suggests a natural relaxation to smooth the resultant unphysical kinks in piecewise CFM: take $w_k \in \mathbb{R}$ *finite* and soften the Dirac deltas to normalized temporal

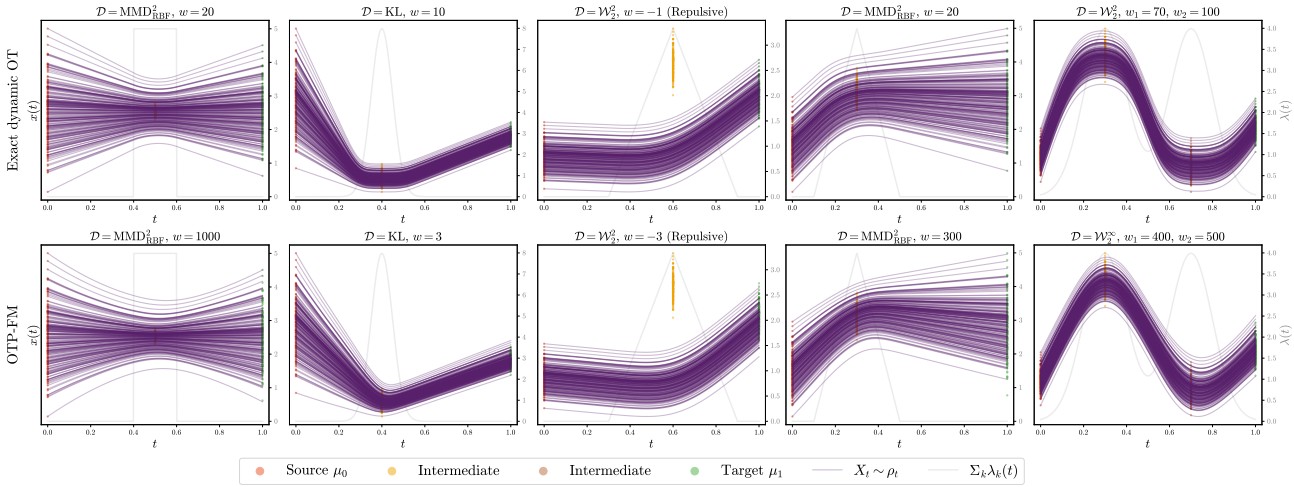

*Figure 2. Top:* Exact solutions to the marginal dynamic OTP problem for 1D Gaussian marginals for varying potentials, strengths, and $\lambda_k(t)$. *Bottom:* OTP-FM solutions for the same marginals and potentials, except the rightmost plot, which demonstrates $\mathcal{D} = \mathcal{W}_2^{\infty}$.

kernels $\lambda_k : [0,1] \to \mathbb{R}^+$ centered around $t_k$ with characteristic width $\tau \in \mathbb{R}^+$. The choice of statistical distance $\mathcal{D}$ remains free; in fact, in the soft setting we can further admit *non-metric divergences* such as the Kullback-Leibler divergence (KLD), which sacrifice exact constraint enforcement in the hard limit but expand the design space (Sec. 3.4). These generalizations yield the OTP variational problem:

$$\mathcal{L}_{\mathrm{OTP}} := \min \int_0^1 \int_{\mathbb{R}^d} \left[ \frac{1}{2} \rho_t \|u_t\|^2 + \overbrace{\sum_{k=1}^K w_k \lambda_k(t) \mathcal{D}_t^k \rho_t}^{\equiv -V_{\mathrm{OTP}}(x,t)} \right. $$
$$\left. + \varphi_t [\partial_t \rho_t + \nabla \cdot (\rho_t u_t)] \right] \mathrm{d}x \mathrm{d}t, \quad (10)$$

subject to the boundary conditions $\rho_0 = \mu_0$, $\rho_1 = \mu_1$. Each term $w_k \lambda_k(t) \mathcal{D}_t^k$ thus acts as a per-marginal potential energy in the action: smoothed in time around $t_k$ by $\lambda_k(t)$; of finite strength $w_k$; and shaped in density space by the choice of $\mathcal{D}$. Proposition 3.1 states that in the singular-potential limit, the OTP problem recovers piecewise CFM and is thus a strict generalization; the precise statement and proof based on $\Gamma$-convergence are provided in App. A.2.

**Proposition 3.1** (OTP converges to the hard-constrained problem in the hard-potential limit, informal). *In the singular limit $w_k \to \infty$, $\lambda_k \to \delta(t - t_k)$ for all $k$, the OTP problem (Eq. 10) with $\mathcal{D} = \mathcal{W}_2^2$ converges to the hard-constrained problem (Eq. 8) and any sequence of OTP minimizers converges to its unique piecewise OT solution between consecutive marginals.*

### 3.3. Minimizers and sample trajectories

To further understand what we have gained with this relaxation, we derive the E-L equations and corresponding sample trajectories of the OTP problem.

**Theorem 3.2.** *Minimizers of the OTP problem (Eq. 10) satisfy the following E-L equations:*

$$u_t(x) = \nabla \varphi_t(x), \quad (11)$$

$$\partial_t \varphi_t(x) + \frac{\|\nabla \varphi_t(x)\|^2}{2} = \sum_{k=1}^K w_k \lambda_k(t) g_k(x,t), \quad (12)$$

*along with the continuity equation, where $g_k(x,t) \equiv \delta \mathcal{D}_t^k / \delta \rho_t(x)$ is the functional derivative of $\mathcal{D}_t^k$ w.r.t. $\rho_{t_k}(x)$.*

**Corollary 3.3.** *Minimizers of the OTP problem (Eq. 10) satisfy the following sample trajectories:*

$$\ddot{X}_t = \sum_{k=1}^K w_k \lambda_k(t) \nabla g_k(X_t, t). \quad (13)$$

Proofs are provided in App. A.3. We can interpret $\nabla g_k(X_t, t)$ as the *force* applied by the marginal $\mu_{t_k}$ to each sample in analogy to Newton's second law. Clearly, it is an important object, and in general not trivial to compute; however, we show that it has convenient forms for common choices of $\mathcal{D}$ such as $\mathcal{W}_2^2$, MMD, and KLD in Table 1 (derived in App. B). Solving these trajectories for the *marginal* problem remains computationally challenging; however, the *conditional* solutions can be computed highly efficiently, as we show in Sec. 4.

### 3.4. Design space of potentials

One case where we *can* solve the marginal problem exactly is for isotropic Gaussian marginals (App. D), illustrated in Fig. 2 for various configurations to visualize our rich design space along three axes: the strength $w_k$, the temporal kernel $\lambda_k(t)$, and the statistical distance $\mathcal{D}$. Overall, we observe intuitive qualitative behavior: trajectories are driven towards

*Table 1.* Functional derivatives $g_k(x,t) \equiv \delta\mathcal{D}_t^k/\delta\rho_t(x)$ and their gradients for different statistical distances $\mathcal{D}_t^k \equiv \mathcal{D}[\rho_t, \mu_{t_k}]$. $\varphi^*$ and $\psi^*$ are the OT H-J potential and map for $c_2$ cost, respectively.

| $\mathcal{D}_t^k$ | $g_k(x,t)$ | $\nabla g_k(x,t)$ |
|---|---|---|
| $\mathcal{W}_2^2$ | $-\varphi^*(x)$ | $x - \psi^*(x)$ |
| $\mathcal{D}_{\mathrm{MMD}^2}^{(\mathrm{kernel}\ k)}$ | $2\int k(x,y)[\rho_t(y) - \mu_{t_k}(y)]\,\mathrm{d}y$ | $2\int \nabla_x k(x,y)[\rho_t(y) - \mu_{t_k}(y)]\,\mathrm{d}y$ |
| $\mathcal{D}_{\mathrm{KL}}$ | $\ln\rho_t(x) - \ln\mu_{t_k}(x) + 1$ | $\nabla\ln\rho_t(x) - \nabla\ln\mu_{t_k}(x)$ |

(or away from) intermediates depending on $w_k$, with the smoothness and temporal dynamics controlled by $\lambda_k(t)$.[3] We discuss practical and performance considerations regarding this design space in the next two sections.

# 4. The OTP-FM algorithm

We now leverage the OTP dynamics to formulate a practical training algorithm by deriving a CFM-style regression objective. We first construct conditional OTP-FM trajectories—conditioned on endpoint and intermediate-marginal samples—and obtain an explicit consistency-model loss (Sec. 4.1). We then derive the regression targets for different statistical distances (Sec. 4.2), identifying $\mathcal{W}_2^{2/\infty}$—with either OT or independent couplings—as a particularly clean, efficient, and stable choice that we recommend in practice. Algorithmic details on the loss and curriculum follow in Sec. 4.3, and we conclude with theoretical bounds on the alignment between learned and ground-truth marginals—controlled by the potential strengths and training loss—in Sec. 4.4.

## 4.1. Conditional OTP-FM and training objective

**Temporal localization.** Since $\nabla g_k(X_t, t)$ in Eq. 13 depends on $\rho_t$ and $X_t$ throughout the support of $\lambda_k(t)$, using these dynamics directly as a regression target would require simulating the trajectory at every training step. Instead, we can exploit the temporal localization of $\lambda_k(t)$: on its effective support of width $\tau$ around $t_k$, $X_t - X_{t_k}$ and $\rho_t - \rho_{t_k}$ are both $\mathcal{O}(\tau)$ (assuming bounded velocity), so $\nabla g_k(X_t, t) = \nabla g_k(X_{t_k}, t_k) + \mathcal{O}(\tau)$. Substituting this into Eq. 13 yields the *OTP-FM dynamics*:

$$\ddot{X}_t = \sum_{k=1}^{K} w_k \lambda_k(t) \nabla g_k(X_{t_k}, t_k). \quad (14)$$

As the force $\nabla g_k(X_t, t)$ now depends only on the trajectory at the discrete marginal times $\{t_k\}$, we are able to formulate the fully *simulation-free* training objective below. Furthermore, we observe qualitatively identical behavior to

---

[3]Counter-intuitively, boundary conditions cause *repulsive* forces (negative $V$) to drive trajectories towards intermediates; the sign convention in Eq. 10 ensures positive $w_k$ corresponds to "attractive" dynamics.

the exact OTP solutions (Fig. 2), and retain the important theoretical bounds (Sec. 4.4) with this approximation.

**Conditional solutions.** Integrating Eq. 14 once and twice yields the velocity and trajectory:

$$u_t(X_t) = u_i + \sum_{k=1}^{K} w_k \nabla g_k(X_{t_k}, t_k)\mathcal{I}[\lambda_k](t), \quad (15)$$

$$X_t = X_0 + u_i t + \sum_{k=1}^{K} w_k \nabla g_k(X_{t_k}, t_k)\mathcal{I}^{(2)}[\lambda_k](t), \quad (16)$$

where $\mathcal{I}^{(n)}[f](t)$ denotes the $n$-th time integral from $0$ to $t$ and $u_i \in \mathbb{R}^d$ is an integration constant fixed by the boundary conditions. As in CFM, we obtain tractable regression targets by conditioning on endpoints $(x_0, x_1)$, as well as, in general, a minibatch $\mathcal{B}$ of samples drawn from a joint distribution $\pi_{\mathrm{all}}$ over all marginals. Imposing $X_0 = x_0$ and $X_1 = x_1$ pins down $u_i$, and the resulting conditional trajectory and velocity decompose into a CFM base term plus potential-driven corrections:

$$u_t(X_t | x_0, x_1, \mathcal{B}) = \underbrace{x_1 - x_0}_{\equiv u^{\mathrm{base}}} +$$

$$\sum_{k=1}^{K} w_k \underbrace{\nabla g_k(X_{t_k}, t_k, \mathcal{B})\left[\mathcal{I}[\lambda_k](t) - \mathcal{I}^{(2)}[\lambda_k](1)\right]}_{\equiv u_{k,t}^{\mathrm{corr}}}. \quad (17)$$

$$X_t(x_0, x_1, \mathcal{B}) = \underbrace{x_0 + (x_1 - x_0)t}_{\equiv X_t^{\mathrm{base}}} +$$

$$\sum_{k=1}^{K} w_k \underbrace{\nabla g_k(X_{t_k}, t_k, \mathcal{B})\left[\mathcal{I}^{(2)}[\lambda_k](t) - \mathcal{I}^{(2)}[\lambda_k](1)\,t\right]}_{\equiv X_{k,t}^{\mathrm{corr}}}, \quad (18)$$

The base terms recover the CFM straight-line path and velocity (Corollary A.2), while each correction $X_{k,t}^{\mathrm{corr}}, u_{k,t}^{\mathrm{corr}}$ captures the impulse from the $k$-th potential.

**The self-consistent fixed-point problem** Note that Eq. 18 is implicit in $\{X_{t_k}\}$: each $\nabla g_k(X_{t_k}, t_k, \mathcal{B})$ is evaluated at $X_{t_k}$, which itself depends on the trajectory through $\nabla g_k(X_{t_k}, t_k, \mathcal{B})$. Solving for $\{X_{t_k}\}$ that are self-consistent with the trajectory they induce is therefore a fixed-point problem, whose structure and solver depend on the choice of distance $\mathcal{D}$ and how its force is estimated, as discussed next in Sec. 4.2.

**The OTP-FM training objective** We train a consistency model $v_{t_1,t_2}^{\theta}$ (Sec. 2.3) by regressing it onto the conditional OTP-FM velocity $u_t(x|z)$ (Eq. 17). Using the iMF formulation (Eq. 7), to be explicit, we obtain the loss:

$$\mathcal{L}_{\mathrm{OTP-FM}}(\theta) := \mathbb{E}_{t_1,t_2,z,x}\|V_{t_1,t_2}^{\theta}(x) - u_t(x|z)\|^2, \quad (19)$$

where $z = (x_0, x_1, \mathcal{B})$ and $V_{t_1, t_2}^\theta(x)$ is the parametrization of $v^\theta$ defined in Eq. 7. We reiterate, however, that OTP-FM is agnostic to the choice of consistency model; we ablate alternative objectives in App. C.

### 4.2. Computing forces and fixed-points

Eq. 19 introduced the general conditioning variable $z = (x_0, x_1, \mathcal{B})$ with $\mathcal{B}$ a minibatch of size $M$ from the joint coupling $\pi_{\text{all}}$, from which the force estimator $\nabla g_k(X_{t_k}, t_k, \mathcal{B})$ is built. For the $\mathcal{W}_2^2$ potential, $\nabla g_k(x) = x - \psi^*(x)$ is well-defined pointwise, so $\mathcal{B}$ *reduces to a single sample per marginal* and the estimator is linear—together yielding the closed-form fixed point and a clean training signal. MMD and KLD, by contrast, are functionals of the full density $\rho_{t_k}$ and degenerate on Diracs, so their estimators require all $M$ samples and yield nonlinear fixed points (Sec. 6.2).

**The $\mathcal{W}_2^2$ distance**  With a single sample $x_{t_k} \sim \mu_{t_k}$ per marginal, the conditional OT map collapses to $\psi^*(X_{t_k}|x_{t_k}) = x_{t_k}$, yielding the one-sample estimator $\nabla g_k(X_{t_k}, t_k, \mathcal{B}) = X_{t_k} - x_{t_k}$ at $\mathcal{O}(M)$ cost per training step. Substituted into Eq. 18 at $t \in \{t_1, \ldots, t_K\}$, this yields a linear system in $X_T \equiv [X_{t_1}, \ldots, X_{t_K}]^\top$ with the direct closed-form solution

$$X_T^{\text{FP}} = (\mathbb{1} - A)^{-1}\left(X_T^{\text{base}} - A\, x_T\right),$$
$$A_{ik} \equiv w_k\left[\mathcal{I}^{(2)}[\lambda_k](t_i) - \mathcal{I}^{(2)}[\lambda_k](1)\, t_i\right], \quad (20)$$

where $x_T \equiv [x_{t_1}, \ldots, x_{t_K}]^\top$ and the $K \times K$ inverse $(\mathbb{1} - A)^{-1}$ depends only on the time grid and kernel and is precomputed once per training (further details in App. E.3). The remaining design choice is the joint coupling $\pi_{\text{all}}$ across times; we use $\pi_{\text{all}}^{\text{OT}} \equiv \prod_{i=0}^{K} \pi_{t_i, t_{i+1}}^{\text{OT}}$, the product of OT couplings between consecutive marginals, precomputed once across the dataset. This is closest in spirit to the $\mathcal{W}_2^2$ distance on the marginal problem. We could alternatively condition on the full minibatch $\mathcal{B}$ and estimate $\psi^*$ via minibatch OT (Fatras et al., 2020) at each step, but find the per-iteration overhead impractical relative to this precomputed variant.

**The $\mathcal{W}_2^\infty$ distance**  While precomputing the OT map has been possible for all experiments in this work, we propose replacing $\pi_{\text{all}}^{\text{OT}}$ with the independent coupling $\pi_{\text{all}}^{\text{ind}}$ for applications to larger-scale, higher-dimensional datasets where the exact OT coupling may not be practical. This is akin to couplings based on entropically regularized OT, $\mathcal{W}_2^\varepsilon$ (Cuturi, 2013), in the limit of the regularization parameter $\varepsilon \to \infty$; hence, we refer to this as the $\mathcal{W}_2^\infty$ potential. Eq. 20 applies unchanged; only $x_T$ on the RHS is now drawn from $\pi_{\text{all}}^{\text{ind}}$ rather than $\pi_{\text{all}}^{\text{OT}}$. The resulting estimator $\nabla g_k(X_{t_k}, t_k, \mathcal{B}) = X_{t_k} - x_{t_k}$ paired with $\pi_{\text{all}}^{\text{ind}}$ likewise costs $\mathcal{O}(M)$ per training step and is hence extremely efficient; furthermore, it does not require the precomputation

of the OT map across all marginals.

**Beyond $\mathcal{W}_2^2$: MMD and KLD potentials**  As exploratory generalizations of the design space, we also consider MMD and KLD potentials, with corresponding forces $\nabla g_k$ derived in Table 1. The MMD admits a natural $\mathcal{O}(M^2)$ empirical estimator using a tunable RBF kernel, while the KLD requires score estimators for $\rho_{t_k}$ and $\mu_{t_k}$, for which we explore an $\mathcal{O}(M)$ Gaussian fit and an $\mathcal{O}(M^2)$ RBF kernel-density estimate (App. B.2). Both yield *nonlinear* fixed-point systems in $X_{t_k}$ and noisier finite-sample gradients than the linear $\mathcal{W}_2^2$ estimator, requiring $\leq 5$ Picard iterations with damped Anderson acceleration (Anderson, 1965) per training step and proving less performant overall (Sec. 6.2).

### 4.3. Algorithmic details

---

**Algorithm 1** The $i$th training iteration of OTP-FM

---

1: **Input:** $v^\theta(x, t_1, t_2)$: model; $M$: batch size; $\alpha(i)$: curriculum parameter;
2: Sample $\mathcal{B} \sim \pi_{\text{all}}$  *# shape: $M \times (K+2) \times D$*
3: Sample $(t_1, t_2) \sim p_{t_1, t_2}$  *# Eq. 114*
4: $x_0, x_1 \leftarrow \mathcal{B}[:, 0], \mathcal{B}[:, -1]$
5: $u^{\text{base}} \leftarrow x_1 - x_0$
6: Compute fixed points $X_{t_k}^{\text{FP}}(\mathcal{B})$ (Eq. 20 for $\mathcal{W}_2^2$ or FP iterations for MMD / KLD)
7: Compute $u_{k, t_1}^{\text{corr}}$ and $X_{k, t_1}^{\text{corr}}$ using Eqs. 17, 18 and $X_{t_k}^{\text{FP}}$
8: *# $\alpha$-weighted Eqs. 17 and 18*
9: $u_{t_1} \leftarrow u^{\text{base}} + \alpha(i) \sum_{k=1}^{K} u_{k, t_1}^{\text{corr}}$  *# $M \times D$*
10: $X_{t_1} \leftarrow x_0 + (x_1 - x_0)t_1 + \alpha(i) \sum_{k=1}^{K} X_{k, t_1}^{\text{corr}}$
11: *# MeanFlow target (Eq. 19), computed using $\texttt{jvp}$*
12: $v_{\text{tgt}} \leftarrow u_{t_1} + (t_2 - t_1)\left[u_{t_1}\partial_x + \partial_{t_1}\right] v^\theta(X_{t_1}, t_1, t_2)$
13: Compute loss $\mathcal{L}_{\text{OTP-FM}}\left(v^\theta(X_{t_1}, t_1, t_2), \text{sg}(v_{\text{tgt}})\right)$
14: Update $v^\theta$ using $\nabla \mathcal{L}_{\text{OTP-FM}}$

---

**Consistency training**  We use a consistency model such as iMF (Geng et al., 2026b) for few-step inference. This is crucial for efficient training of the MMD/KLD potentials, which require evaluation of the learned $X_{t_k}$ during training to compute the forces $\nabla g_k(X_{t_k}, t_k, \mathcal{B})$ and perform FP iterations (Sec. 4.2). While not necessary for the $\mathcal{W}_2^2$ potential, it nevertheless enables efficient inference. As is common for image generation, we additionally maintain an exponential moving average (EMA) of the model weights $u_\theta^{\text{EMA}}$ for both inference and evaluation of $X_{t_k}$ during training.

**Loss function**  We experiment with three weightings of the squared-L2 regression target (Eq. 19): 1) unweighted MSE; 2) the adaptive weighting of MeanFlow, replacing $\|\Delta\|_2^2$ with $\|\Delta\|_2^2 / \text{sg}(\|\Delta\|_2^2 + c)^p$, where $\Delta$ is the regression residual, $c$ a small constant, and $p$ a tunable exponent; and 3) a learnt log-variance weighting (Karras et al., 2024). We

find the adaptive weighting with $p = 1$ generally the most performant; comparisons are shown in Apps. F and G.

**Homotopy curriculum** As the fixed points of Eq. 18 are not guaranteed to be attractive for arbitrary initializations, we find it beneficial to introduce a training curriculum that scales the correction terms by a smoothly increasing parameter $\alpha(i) \in [0, 1]$ over training iterations $i$, rewriting Eq. 18 as a family of fixed-point maps:

$$X_t = F_\alpha(X_t) \equiv X_t^{\text{base}} + \alpha \sum_{k=1}^{K} X_{k,t}^{\text{corr}}, \quad \forall t \in \{t_k\}. \quad (21)$$

This is an application of the *homotopy continuation* technique (Allgower & Georg, 2003): we start by allowing the model to learn at $\alpha \approx 0$, where the problem reduces to CFM with a trivially attractive fixed point, and gradually transition to the full OTP-FM dynamics at $\alpha = 1$. Under standard regularity conditions, the implicit function theorem guarantees the existence of a locally unique and smooth solution branch $\alpha \mapsto X_{t_k}^{\text{FP}}(\alpha)$, and increasing $\alpha$ gradually allows the training dynamics to track this branch (App. E.2). In practice, we use a sigmoid schedule with tunable mean and slope, with ablations in Sec. 6.2. Interestingly, we find this more performant even for the $\mathcal{W}_2^2$ potentials, for which we have a closed-form solution.

### 4.4. Theoretical bounds

As in standard CFM, $\mathcal{L}_{\text{OTP−FM}}$ is gradient-equivalent to a marginal FM loss regressing $v^\theta$ onto the induced marginal velocity $u_t(x) = \int u_t(x|z)\,\rho_t(z|x)\,\mathrm{d}z$ along $\rho_t(x) = \int \rho_t(x|z)\,q(z)\,\mathrm{d}z$.[4] The following two propositions show that OTP-FM with the $\mathcal{W}_2^2$ potential *guarantees alignment of this induced $\rho_t$ and the ground-truth intermediate marginals $\mu_{t_k}$, with the $\mathcal{W}_2$ bound between them controlled by the potential strength $w$ and training loss $\mathcal{L}_{\text{OTP−FM}}$*. Proofs are provided in App. A.4.

**Proposition 4.1.** *(Marginal alignment bound). For $\mathcal{W}_2^2$-type potentials of strength $w$, away from singular configurations: $\mathcal{W}_2^2(\rho_{t_k}, \mu_{t_k}) \leq C_k/w^2 + \mathcal{O}(w^{-3})$, where $C_k$ is a problem-dependent constant independent of $w$.*

**Proposition 4.2.** *(End-to-end bound). For Eulerian and Lagrangian consistency losses (App. C), combining Prop. 4.1 with flow-map learning bounds from Boffi et al. (2025): $\mathcal{W}_2(\rho_{t_k}^\theta, \mu_{t_k}) \leq D_k\sqrt{\mathcal{L}_{\text{OTP−FM}}} + \sqrt{C_k}/w + \mathcal{O}(w^{-3/2})$, where $C_k$ and $D_k$ are problem- and consistency-loss-dependent constants, respectively, both independent of $w$.*

---

[4]This follows by direct application of Tong et al. (2024, Thms. 3.1–3.2) with the OTP-FM conditioning variables $z = (x_0, x_1, \mathcal{B}) \sim q$.

## 5. Related work

Like OTP-FM, previous work in trajectory inference has focused on modeling dynamics through neural ordinary or stochastic differential equations (ODEs or SDEs) (Chen et al., 2018). Originally, these methods were trained by repeatedly simulating the ODE or SDE, with examples such as TrajectoryNet (Tong et al., 2020), MIOFlow (Huguet et al., 2022), and Schrödinger-bridge methods like DMSB (Chen et al., 2023), NLSB (Koshizuka & Sato, 2023), and Deep-RUOT (Zhang et al., 2025). These, while flexible, generally prove computationally prohibitive for large-scale, high-dimensional data.

More recently, flow- and score-matching methods successfully pioneered *simulation-free* training as described above. However, most focus only on trajectories between two endpoint marginals, simply stitching trajectories piecewise in the case of multiple marginals, as in I-/OT-CFM and [SF]²M (Tong et al., 2024; 2023). MMFM (Rohbeck et al., 2025) and 3MSBM (Theodoropoulos et al., 2026) are two recent methods that smooth these trajectories, through cubic spline interpolation and lifting to phase space, respectively. However, as evidenced by our experiments below, these prescribed smoothing strategies need not describe the dynamics of the physical system. In contrast, OTP-FM provides a principled framework for deriving smooth, flexible dynamics that lets the data, rather than an ad-hoc interpolation rule, determine the trajectories through optimization over the broad design space. New methods such as MFM (Kapusniak et al., 2024) and VGFM (Wang et al., 2026) represent orthogonal improvements to CFM—learning the data manifold and modeling unbalanced cell growth, respectively—that can be combined with our approach.

Finally, methods such as WLF (Neklyudov et al., 2024), JKOnet(*) (Bunne et al., 2022; Terpin et al., 2024), and iJKOnet (Persiianov et al., 2026) directly solve a multimarginal OT-like variational problem over the entire dataset, and are promising but fundamentally different computational approaches to solving the *conditional* OT problem within the CFM framework, as in OTP-FM. We benchmark against all of the above methods.

## 6. Experiments

### 6.1. Synthetic data

We first validate OTP-FM on 1D Gaussian marginals where exact dynamic OT solutions are available (App. D). Figure 2 illustrates that OTP-FM faithfully reproduces the qualitative behavior across potential types and configurations. Empirically, $\mathcal{W}_2^2$ with the $\pi_{\text{all}}^{\text{OT}}$ coupling is the most performant and stable to train, with $\mathcal{W}_2^\infty$ and $\pi_{\text{all}}^{\text{ind}}$ comparably effective while avoiding the OT precomputation overhead; the kernel-based MMD and KLD potentials, by contrast, are

*Table 2.* Ablation of OTP-FM design choices on EB 100D L2O ($\overline{\text{MMD}}$ on held-out times, lower is better). The optimal value is in **bold**.

| Potential | $\overline{\text{MMD}}\downarrow$ | $w$ | $\overline{\text{MMD}}\downarrow$ | $\lambda(t)$ | $\overline{\text{MMD}}\downarrow$ | Curriculum | $\overline{\text{MMD}}\downarrow$ |
|---|---|---|---|---|---|---|---|
| $\mathcal{W}_2^2$ | **0.0675** | $w=10$ | 0.1431 | **Gaussian** | **0.0675** | Linear ($\alpha = 0 \to 1$) | 0.0800 |
| $\mathcal{W}_2^\infty$ | 0.0813 | $w=100$ | 0.0726 | Box | 0.0681 | Constant ($\alpha = 1$) | 0.0687 |
| MMD | 0.0861 | $w=1000$ | **0.0675** | Triangle | 0.0723 | **Sigmoid ($\mu = 0$, $\beta = 6$)** | **0.0675** |
| KL | 0.0876 | $w=10000$ | 0.0683 | | | Sigmoid ($\mu = 1$, $\beta = 6$) | 0.0758 |
| | | | | $\tau = 0.2$ | 0.0690 | Sigmoid ($\mu = 0$, $\beta = 3$) | 0.0683 |
| | | | | $\tau = 0.33$ | **0.0675** | Sigmoid ($\mu = 0$, $\beta = 12$) | 0.0681 |
| | | | | $\tau = 0.5$ | 0.0693 | | |

less stable due to their nonlinear fixed-point problem and noisier gradient estimates (further details in App. F).

### 6.2. Ablation studies

We ablate the key design choices of OTP-FM on the EB 100D L2O benchmark (described below) in Table 2 to show sensitivity to 1) potential type, 2) potential strength, 3) $\lambda_k(t)$ width and shape, and 4) curriculum. We vary one parameter at a time around the optimum (bold) and use an 8-layer MLP ResNet architecture for $v^\theta$ (further details in App. G.3).

Comparing potential types is the central ablation: $\mathcal{W}_2^2$ is most performant and $\mathcal{W}_2^\infty$ second; combined with their efficiency and the training-stability issues of MMD/KLD, this establishes $\mathcal{W}_2^{2/\infty}$ as the preferred potentials for OTP-FM. Surprisingly, a sigmoid curriculum outperforms a constant one even for the $\mathcal{W}_2^2$ potential—despite its original motivation as homotopy continuation for MMD/KLD (Sec. 4.3)—suggesting training still benefits from starting from the simpler CFM target. The sensitivity to potential strength and temporal dynamics demonstrates the flexibility OTP-FM offers over prescribed-interpolation approaches: these are crucial axes for the SOTA performance shown next. Further ablations on the consistency model and loss function, as well as trajectory visualizations, are in App. G.

### 6.3. Single-cell RNA sequencing

We perform a comprehensive comparison in Table 3 with all baselines in Sec. 5 on two scRNA-seq datasets: developing embryoid bodies (EB, 5 time intervals $t_0$—$t_4$) (Moon et al., 2019) and CITE-seq of human CD34+ HSPCs (CITE, 4 time intervals) (Burkhardt et al., 2022), operating on up to a 100D PCA representation. To ensure at least one reported point of comparison with each baseline, we adopt six established protocols: EB 5D leave-one-marginal-out (L1O) evaluating $\mathcal{W}_1$ at the held-out time (Tong et al., 2020); EB 100D L0O (no holdouts, MMD averaged over training times) and L1O (MMD over held-out times) (Persiianov et al., 2026); EB 100D L2O, holding out $t_1$ and $t_3$ with MMD averaged over *all four* times after $t_0$ (Theodoropoulos et al., 2026); and CITE 5D/50D L1O, averaging $\mathcal{W}_1$ over held-out times (Kapusniak et al., 2024); 5D/50D refer to the first 5/50 PCs. Evaluating on both training and held-out marginals tests

how well each method recovers the training distribution *and* learns physically plausible trajectories that accurately *interpolate* unseen marginals.

We additionally measure each method's training time on consistent hardware and plot training time vs. performance for CITE 5D and EB 100D L2O in Fig. 3. Overall, OTP-FM achieves SOTA results on nearly all metrics in both performance *and* training efficiency, completing training within 3–5 minutes on an NVIDIA L40S GPU in all settings. The one exception in terms of fidelity is the EB 100D L0O experiment, where it is outperformed by plain piecewise CFM and [SF]²M; this is the only setting that does not measure interpolation ability — precisely the limitation of those methods that OTP-FM is designed to address. Additional results, trajectory visualizations, and dataset, model, and benchmarking details are in App. G.

### 6.4. Gulf of Mexico and Beijing air quality

We further evaluate on two non-biological datasets: ocean current particle transport in the Gulf of Mexico (2D, 9 timepoints) (Shen et al., 2025) and hourly particle concentrations from Beijing air quality monitoring stations (1D, 13 timepoints) (Chen, 2017). Following Theodoropoulos et al. (2026), Table 4 reports the average $\mathcal{W}_2$ across four held-out timepoints each as well as the remaining training timepoints. We compare against the two simulation-free methods closest in motivation to OTP-FM—MMFM and 3MSBM, both of which learn smooth multimarginal trajectories through *prescribed* interpolation strategies (Fig. 1, Sec. 5). OTP-FM significantly outperforms both at interpolating the held-out times, particularly on Beijing, which we attribute directly to its flexible, data-driven dynamics; trajectory visualizations and further analysis are in Apps. H and I.

## 7. Conclusion

We introduced OTP-FM, a principled generalization of multimarginal CFM grounded in dynamic OT with soft potential energy terms to steer learnt trajectories toward intermediate marginals. It recovers piecewise CFM in the singular-potential limit and unlocks a broad design space for the spatiotemporal dynamics. We derived exact conditional solutions and an efficient simulation-free training algorithm,

*Table 3.* Results on the EB and CITE scRNA-seq datasets for different experimental protocols, as defined in the text. Mean and std. dev. over 5 seeds is shown where available (lower is better for all metrics). Results for all methods described in Sec. 5 are shown. A * indicates our own training of the method using the provided code, while gray indicates we were unable to produce reasonable results on that experiment; otherwise, the result is from the prior work. Best performing method is in **bold** and second-best in *italics*.

| | EB 5D | EB 100D | | | CITE 5D | CITE 50D |
|---|---|---|---|---|---|---|
| Method | L1O $\overline{\mathcal{W}_1}\downarrow$ | L0O $\overline{\mathrm{MMD}}\downarrow$ | L1O $\overline{\mathrm{MMD}}\downarrow$ | L2O $\overline{\mathrm{MMD}}\downarrow$ | L1O $\overline{\mathcal{W}_1}\downarrow$ | L1O $\overline{\mathcal{W}_1}\downarrow$ |
| *Neural ODE / SDE: Simulation-based training* | | | | | | |
| TrajectoryNet | 0.784 | | | | | |
| NLSB | 0.970 | 0.660 | 0.373±0.000 | | 0.767* | |
| MIOFlow | 0.881* | 0.230 | 0.453±0.000 | 1.164* | 0.494* | 38.333±0.002 |
| DMSB | 1.775±0.429 | 1.976* | 0.579* | 0.518* | 1.705±0.160 | 61.188* |
| DeepRUOT | 0.774 | 1.096* | 1.793* | 0.718* | 0.845 | 37.892±0.002 |
| *Variational / OT* | | | | | | |
| WLF-UOT | 0.800±0.002 | 1.414* | 0.190* | 0.262* | 0.733±0.063 | 37.035±0.079 |
| JKOnet* | 3.055* | 0.229±0.052 | 0.249±0.010 | 2.016* | 1.237* | 242.784* |
| iJKOnet | 1.137* | 0.085±0.024 | *0.119±0.001* | 1.347* | 0.594* | 92.540* |
| *Neural ODE / SDE: Simulation-free training* | | | | | | |
| 3MSBM | | | | 0.135±0.014 | | |
| I-CFM | 0.872±0.087 | *0.028** | 0.390* | 0.284* | 0.965±0.111 | 41.834±3.284 |
| OT-CFM | 0.790±0.068 | 0.079* | 0.151* | 0.093* | 0.882±0.058 | 38.756±0.010 |
| [SF]²M | 0.793±0.066 | **0.023*** | 0.248* | 0.157* | 0.920±0.049 | 38.524±0.293 |
| OT-MFM | 0.713±0.039 | 0.188* | 0.201* | 0.105* | 0.724±0.070 | **36.394** |
| VGFM | *0.676* | 0.129* | 0.208* | 0.131* | 0.745 | 37.386 |
| MMFM | 0.899±0.010* | 0.247±0.006* | 0.286±0.008* | 0.101±0.003* | 0.553±0.003* | 44.433±0.399* |
| OTP-FM ($\mathcal{W}_2^2$) | **0.675±0.019** | 0.041±0.002 | **0.117±0.010** | **0.068±0.001** | **0.435±0.003** | *36.966±0.398* |
| OTP-FM ($\mathcal{W}_2^\infty$) | 0.804±0.023 | 0.173±0.015 | 0.151±0.004 | *0.081±0.002* | *0.466±0.022* | 40.312±0.239 |

*Figure 3.* Training time vs. performance of different methods for the CITE 5D L1O (top) and EB 100D L2O (bottom) experiments. Red stars denote OTP-FM. Top right is better.

*Table 4.* Results on GoM and Beijing air quality datasets showing average $\mathcal{W}_2$ across held-out and training timepoints (lower is better). MMFM results are with our training (marked with *) and 3MSBM from the original paper, which does not report training $\mathcal{W}_2$ scores on the Beijing air quality dataset. Best performing method is in **bold** and second-best in *italics*.

| | Gulf of Mexico | | Beijing Air Quality | |
|---|---|---|---|---|
| Method | Holdout $\overline{\mathcal{W}_2}\downarrow$ | Train $\overline{\mathcal{W}_2}\downarrow$ | Holdout $\overline{\mathcal{W}_2}\downarrow$ | Train $\overline{\mathcal{W}_2}\downarrow$ |
| 3MSBM | 0.135±0.012 | **0.050±0.030** | 38.09±3.67 | |
| MMFM | 0.150±0.005* | 0.083±0.008* | 38.89±3.92* | **11.88±2.81*** |
| OTP-FM ($\mathcal{W}_2^2$) | **0.107±0.001** | *0.059±0.006* | **17.28±0.56** | *16.52±2.33* |
| OTP-FM ($\mathcal{W}_2^\infty$) | *0.121±0.003* | 0.069±0.009 | *23.15±1.08* | 24.08±1.79 |

proved Wasserstein bounds on the alignment with ground-truth marginals controlled by potential strength and training loss, and demonstrated that OTP-FM pushes the frontier in both performance and training efficiency across biological, oceanographic, and meteorological datasets.

**Practical recommendations** Across the design space, $\mathcal{W}_2^{2/\infty}$ potentials are clearly preferred: they yield closed-form, low-variance gradients, a linear fixed-point system, and the best empirical performance. Our out-of-the-box recipe is $\mathcal{W}_2^\infty$ with strength $w \approx 1000$, equally-spaced Gaussian $\lambda_k(t)$, and a sigmoid curriculum (Sec. 6.2, App. G); $w$ is the most impactful parameter to tune, with $\lambda_k(t)$ shape and width having secondary effects. Switching to $\mathcal{W}_2^2$ with the $\pi_{\mathrm{all}}^{\mathrm{OT}}$ coupling can provide further improvement when the $\mathcal{O}(N^3)$ OT precomputation is feasible.

**Limitations and future work** While matching intermediate marginals encourages more plausible trajectories, it does not alone guarantee physically meaningful interpolated states, particularly for sparsely-sampled or noisily-labeled data; methods that explicitly learn the data manifold such as MFM (Kapusniak et al., 2024) and pixel Mean-Flow (Lu et al., 2026) are important orthogonal directions for improvement. Future work will also explore incorporating domain-specific inductive biases such as cell growth in VGFM (Wang et al., 2026), and conditional inference as in MMFM (Rohbeck et al., 2025). Like CFM, OTP-FM solves the *conditional* OTP problem given a joint coupling $\pi_{\mathrm{all}}$, rather than jointly optimizing the dynamics and coupling toward the full marginal variational problem. A rectified-flow-style retraining (Liu et al., 2023) or combining our conditional formulation with new direct multimarginal-OT solvers such as WLF (Neklyudov et al., 2024) are interesting directions toward self-consistent dynamics–coupling pairs.

Beyond the $\mathcal{W}_2^2$ potential, our MMD and KLD force estimators rely on kernel-based estimates of the full marginal density, which are noisier than $\mathcal{W}_2^2$'s linear, pointwise gradient and yield nonlinear fixed-point systems. Improving them via adaptive kernels or learned score models could unlock a richer family of physically-motivated potentials. Finally, while $\mathcal{W}_2^\infty$ is a strong default, OTP-FM's broad design space imposes a tuning burden absent from prescriptive methods; automating this search, exploring a broader parameter space, and deploying to novel scientific domains are all exciting directions for future work.

## Impact Statement

This work introduces a principled and computationally efficient framework for learning the latent dynamics of physical systems from discrete empirical snapshots. In biological contexts, OTP-FM enables reconstructing continuous trajectories from sparsely sampled, high-dimensional observations. In climate science and oceanography, the same formulation improves dynamical modeling from limited temporal data. The efficiency and performance of OTP-FM lowers the barrier to trajectory inference on new, large-scale scientific datasets.

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

# A. Proofs of theorems

## A.1. Dynamic OT without a potential

Proofs for the E-L equations and straight-line trajectories for the dynamic OT problem can be found in standard texts such as Villani (2009), and are reproduced here for completeness and later reference in App. A.3.

**Theorem A.1** (E-L equations). *For $V = 0$, minimizers of Eq. 2 satisfy the following Euler–Lagrange (E-L) equations:*

$$u_t = \nabla \varphi_t, \qquad \partial_t \varphi_t(x) + \frac{\|\nabla \varphi_t(x)\|^2}{2} = 0, \tag{22}$$

*along with the continuity equation.*

*Proof.* We start with the dynamic OT action without a potential:

$$S[\mathcal{L}] = \int_0^1 \int_{\mathbb{R}^d} \left[ \frac{1}{2} \|u_t(x)\|^2 \rho_t(x) + \varphi_t(x)[\partial_t \rho_t(x) + \nabla \cdot (\rho_t(x) u_t(x))] \right] dx dt. \tag{23}$$

Integrating the second term by parts yields:

$$\int_0^1 \int_{\mathbb{R}^d} \varphi_t(x)[\partial_t \rho_t(x) + \nabla \cdot (\rho_t(x) u_t(x))] = \int_{\mathbb{R}^d} [\varphi_t(x)\rho_t(x)]_0^1 \, dx - \int_0^1 \int_{\mathbb{R}^d} [\partial_t \varphi_t(x) + u_t(x) \cdot \nabla \varphi_t(x)] \, \rho_t(x) dx dt \tag{24}$$

For fixed boundary conditions, the first term is simply a constant with respect to $u$ and $\rho$.

The minima can thus be found by taking the functional derivatives of $S$:

$$\frac{\delta S}{\delta \varphi} = \partial_t \rho_t(x) + \nabla \cdot (\rho_t(x) u_t(x)) = 0 \qquad \text{(Continuity equation)} \tag{25}$$

$$\frac{\delta S}{\delta u} = (u_t(x) - \nabla \varphi_t(x)) \, \rho_t(x) = 0 \qquad\qquad \Rightarrow u_t(x) = \nabla \varphi_t(x) \tag{26}$$

$$\frac{\delta S}{\delta \rho} = \frac{\|u_t(x)\|^2}{2} - [\partial_t \varphi_t(x) + u_t(x) \cdot \nabla \varphi_t(x)] = 0 \quad \Rightarrow \partial_t \varphi_t(x) + \frac{\|\nabla \varphi_t(x)\|^2}{2} = 0 \quad \text{(H-J equation)} \tag{27}$$

where in Eq. 27 we plugged in Eq. 26. $\qquad \square$

**Corollary A.2** (Straight-line trajectories). *Minimizers of Eq. 2 (with $V = 0$) yield straight-line sample trajectories $X_t = (1-t)X_0 + t\psi(X_0)$ with $\dot{X}_t = \psi(X_0) - X_0$, where $\psi$ is the static OT map for cost $c_2$.*

*Proof.* We first take the gradient of the H-J equation:

$$\partial_t \nabla \varphi + (\nabla^2 \varphi) \nabla \varphi = 0, \tag{28}$$

where $\nabla^2 \varphi$ is the Hessian of $\varphi$. Next, the time derivative of the sample velocity $\dot{X}_t$ (Eq. 26):

$$\begin{aligned}
\frac{du(X_t)}{dt} &= \frac{d}{dt} \nabla \varphi_t(X_t) \\
&= \partial_t \nabla \varphi_t(X_t) + (\nabla^2 \varphi)(X_t) \dot{X}_t \\
&\overset{(i)}{=} \partial_t \nabla \varphi_t(X_t) + (\nabla^2 \varphi)(X_t) \nabla \varphi_t(X_t) \\
&\overset{(ii)}{=} 0,
\end{aligned} \tag{29}$$

where in $(i)$ we applied Eq. 26 again and in $(ii)$ used Eq. 28. Thus, the sample trajectories obey

$$\ddot{X}_t = 0. \tag{30}$$

As the dynamic OT solution is equivalent to that of the static OT problem with cost $c_2$, given a sample $X_0 \sim \mu_0$, its trajectory must lead to $X_1 = \psi(X_0) \sim \mu_1$. Therefore, integrating Eq. 30 with boundary conditions $X_0$ at $t = 0$ and $\psi(X_0)$ at $t = 1$ yields $X_t = (1-t)X_0 + t\psi(X_0)$ and $\dot{X}_t = \psi(X_0) - X_0$. $\qquad \square$

## A.2. Piecewise multimarginal CFM as the hard-potential limit of OTP

We make the connection between standard piecewise multimarginal CFM and the OTP variational problem (Sec. 3) precise via two main results: piecewise multimarginal CFM is equivalent to a single CFM training regressing onto conditional solutions of the hard-constrained problem $\mathcal{L}_{\mathrm{HC}}$ (Lemma A.4), and OTP minimizers converge to the solutions of the hard-constrained problem in the singular, hard-penalty limit of $w_k \to \infty$, $\lambda_k \to \delta(t - t_k)$ (Prop. A.5, stated for $\mathcal{D} = \mathcal{W}_2^2$). In order to prove the former, we first establish that $\mathcal{L}_{\mathrm{HC}}$ decomposes additively into $K + 1$ independent Benamou–Brenier (BB) subproblems (Lemma A.3), where $K$ is the number of intermediate marginals.

Throughout this subsection, we assume $0 = t_0 < t_1 < \cdots < t_K < t_{K+1} = 1$ and that all marginals $\{\mu_{t_k}\}_{k=0}^{K+1}$ are absolutely continuous probability measures on $\mathbb{R}^d$, uniformly bounded in $\mathcal{P}_2(\mathbb{R}^d)$ — i.e., with finite second moments — so that $\mathcal{W}_2^2(\mu_{t_k}, \mu_{t_{k+1}}) < \infty$ for all $k$. An *admissible flow* is a pair $(\rho_t, u_t)_{t \in [0,1]}$ with $\rho_t \in \mathcal{P}_2(\mathbb{R}^d)$ a curve of probability measures and $u_t$ a velocity field satisfying the continuity equation and endpoint boundary conditions $\rho_0 = \mu_0$, $\rho_1 = \mu_1$. We define intervals $I_k \equiv [t_k, t_{k+1}]$ and write the rescaled time $s_k(t) := (t - t_k)/(t_{k+1} - t_k) \in [0, 1]$ for convenience.

We refer to *piecewise multimarginal CFM* as the standard procedure from, e.g., Tong et al. (2024; 2023): pick any joint coupling $\pi_{\mathrm{all}}$ of $(\mu_{t_0}, \ldots, \mu_{t_{K+1}})$ and train $K + 1$ independent CFM models $u_{\theta_k}^k : \mathbb{R}^d \times I_k \to \mathbb{R}^d$ $(k = 0, \ldots, K)$, one per consecutive interval $I_k$ between $(\mu_{t_k}, \mu_{t_{k+1}})$, against the standard CFM regression losses $\mathcal{L}_k^{\mathrm{piecewise}}(\theta_k)$ with endpoint pairs drawn from the marginal of $\pi_{\mathrm{all}}$ on $(x_{t_k}, x_{t_{k+1}})$ and conditional-OT solutions between them as regression targets (Eq. 5), then stitch the $K + 1$ learned flows at each $\{t_k\}_{k=1}^K$:

$$u_\theta^{\mathrm{piecewise}}(x, t) = \sum_{k=0}^K \mathbf{1}_{I_k}(t)\, u_{\theta_k}^k(x, t), \qquad \theta \equiv (\theta_0, \ldots, \theta_K), \tag{31}$$

where $\mathbf{1}_{I_k}$ is the indicator function of $I_k$.[5]

**Lemma A.3** (Additive decomposition of $\mathcal{L}_{\mathrm{HC}}$). *Under the setup above, the hard-constrained action $\mathcal{L}_{\mathrm{HC}}$ (Eq. 8) decomposes additively into $K + 1$ independent Benamou–Brenier dynamic OT subproblems between consecutive marginals,*

$$\mathcal{L}_{\mathrm{HC}} = \sum_{k=0}^K \min_{(\rho, u)|_{I_k}} \int_{I_k}\!\!\int \tfrac{1}{2}\|u_t\|^2 \rho_t \,\mathrm{d}x\,\mathrm{d}t = \sum_{k=0}^K \frac{\mathcal{W}_2^2(\mu_{t_k}, \mu_{t_{k+1}})}{2(t_{k+1} - t_k)}, \tag{32}$$

*where the inner minimization on $I_k$ is over admissible flows constrained by $\rho_{t_k} = \mu_{t_k}$ and $\rho_{t_{k+1}} = \mu_{t_{k+1}}$.*

*Proof.* Let $(\rho_t, u_t)$ be an admissible flow that additionally satisfies $\rho_{t_k} = \mu_{t_k}$ for all $k$. The kinetic energy is additive over the intervals $I_k$,

$$\int_0^1\!\!\int \tfrac{1}{2}\|u_t\|^2 \rho_t \,\mathrm{d}x\,\mathrm{d}t = \sum_{k=0}^K \int_{I_k}\!\!\int \tfrac{1}{2}\|u_t\|^2 \rho_t \,\mathrm{d}x\,\mathrm{d}t, \tag{33}$$

and the only active constraints over each interval are the continuity equation $(\rho_t, u_t)|_{I_k}$ and the endpoint conditions $\rho_{t_k} = \mu_{t_k}$, $\rho_{t_{k+1}} = \mu_{t_{k+1}}$; no constraint couples distinct intervals. Hence, the joint minimization decouples into $K + 1$ independent subproblems:

$$\mathcal{L}_{\mathrm{HC}|I_k} = \min_{(\rho, u)|_{I_k}} \int_{I_k}\!\!\int \tfrac{1}{2}\|u_t\|^2 \rho_t \,\mathrm{d}x\,\mathrm{d}t \quad \text{s.t.} \ \ \partial_t \rho + \nabla\cdot(\rho u) = 0, \ \ \rho_{t_k} = \mu_{t_k}, \ \ \rho_{t_{k+1}} = \mu_{t_{k+1}}. \tag{34}$$

Each of these subproblems thus is the standard BB problem between $\mu_{t_k}$ and $\mu_{t_{k+1}}$, most easily seen by time rescaling $s \equiv s_k(t)$ as above and defining $\tilde{u}(s, x) \equiv (t_{k+1} - t_k)\, u(t, x)$ to satisfy the continuity equation in $s$:

$$\begin{aligned} \mathcal{L}_{\mathrm{HC}|I_k} &= \frac{1}{t_{k+1} - t_k} \min_{(\rho, u)} \int_0^1\!\!\int \tfrac{1}{2}\|u_s\|^2 \rho_s \,\mathrm{d}x\,\mathrm{d}s \quad \text{s.t.} \ \ \partial_s \rho + \nabla\cdot(\rho u) = 0, \ \ \rho_0 = \mu_{t_k}, \ \ \rho_1 = \mu_{t_{k+1}}. \\ &= \frac{\mathcal{W}_2^2(\mu_{t_k}, \mu_{t_{k+1}})}{2(t_{k+1} - t_k)} \end{aligned} \tag{35}$$

Summing over all intervals yields Eq. 32. □

---

[5]The overlap at the $K$ interior endpoints $\{t_k\}_{k=1}^K$ is measure zero and so irrelevant to any time integral.

**Lemma A.4** (Piecewise multimarginal CFM is equivalent to CFM regression onto conditional solutions of $\mathcal{L}_{\mathrm{HC}}$)**.** *Under the setup above and any joint coupling $\pi_{\mathrm{all}}$ of $(\mu_{t_0}, \ldots, \mu_{t_{K+1}})$, conditioning on a sampled endpoint tuple $z = (x_{t_0}, \ldots, x_{t_{K+1}}) \sim \pi_{\mathrm{all}}$ reduces each per-interval subproblem in Eq. 32 to the dynamic OT problem between the Dirac delta functions $\delta(x - x_{t_k})$ and $\delta(x - x_{t_{k+1}})$, whose unique conditional solution is the straight-line trajectory:*

$$X_t^{\mathrm{HC}}(x|z) \;=\; (1 - s_k(t))\, x_{t_k} + s_k(t)\, x_{t_{k+1}}, \qquad u_t^{\mathrm{HC}}(x|z) \;=\; \frac{x_{t_{k+1}} - x_{t_k}}{t_{k+1} - t_k}, \qquad t \in I_k, \tag{36}$$

*with conditional path $\rho_t^{\mathrm{HC}}(x \mid z) = \delta(x - X_t^{\mathrm{HC}}(x \mid z))$. The corresponding CFM regression loss*

$$\mathcal{L}^{\mathrm{HC-CFM}}(\theta) \;:=\; \mathbb{E}_{t \sim U[0,1]}\, \mathbb{E}_{z \sim \pi_{\mathrm{all}}}\, \mathbb{E}_{x \sim \rho_t^{\mathrm{HC}}(\cdot|z)} \left\| v^\theta(x, t) - u_t^{\mathrm{HC}}(x \mid z) \right\|^2 \tag{37}$$

*of a single shared model $v^\theta(x, t)$ on $[0, 1]$ then equals the total piecewise multimarginal CFM loss under $\pi_{\mathrm{all}}$,*

$$\mathcal{L}^{\mathrm{HC-CFM}}(\theta) \;=\; \sum_{k=0}^{K} \mathcal{L}_k^{\mathrm{piecewise}}(\theta_k),$$

*when the $K + 1$ piecewise models are tied to a shared parameterization via $u_{\theta_k}^k(x, t) = \mathbf{1}_{I_k} v^\theta(x, t)$.*

*Proof. Conditional HC solution.* Let the CFM conditioning variable $z = (x_{t_0}, \ldots, x_{t_{K+1}}) \sim \pi_{\mathrm{all}}$. By Lemma A.3, $\mathcal{L}_{\mathrm{HC}}$ decomposes into $K + 1$ independent BB subproblems (Eq. 34); the conditional version of each replaces the marginal boundary constraints $\rho_{t_k} = \mu_{t_k}, \rho_{t_{k+1}} = \mu_{t_{k+1}}$ with their conditional counterparts $\rho_{t_k}(x \mid z) = \delta(x - x_{t_k})$, $\rho_{t_{k+1}}(x \mid z) = \delta(x - x_{t_{k+1}})$, yielding a dynamic OT problem between two Dirac delta functions on $I_k$.

After rescaling to $[0, 1]$ as in Lemma A.3, Corollary A.2 provides its unique minimizer: the straight-line trajectory $X_s = (1 - s)\, x_{t_k} + s\, x_{t_{k+1}}$; undoing the rescaling yields Eq. 36.

*Loss equality.* Splitting the time integral in Eq. 37 across the $K + 1$ subintervals and using $\rho_t^{\mathrm{HC}}(x \mid z) = \delta(x - X_t^{\mathrm{HC}})$ yields:

$$\mathcal{L}^{\mathrm{HC-CFM}}(\theta) \;=\; \sum_{k=0}^{K} \int_{I_k} \mathbb{E}_{(x_{t_k}, x_{t_{k+1}}) \sim \pi_{\mathrm{all}}^{t_k, t_{k+1}}} \left\| v^\theta(X_t^{\mathrm{HC}}, t) - \frac{x_{t_{k+1}} - x_{t_k}}{t_{k+1} - t_k} \right\|^2 \mathrm{d}t, \tag{38}$$

where we marginalized the components of $z$ not appearing in the $k$-th summand. Each summand is exactly the standard CFM loss on $I_k$ under the consecutive-pair coupling $\pi_{\mathrm{all}}^{t_k, t_{k+1}}$; i.e., $\mathcal{L}_k^{\mathrm{piecewise}}$ evaluated at $u_{\theta_k}^k(x, t) = v^\theta(x, t)$ for $t \in I_k$. Hence, $\mathcal{L}^{\mathrm{HC-CFM}}(\theta) = \sum_{k=0}^{K} \mathcal{L}_k^{\mathrm{piecewise}}(\theta_k)$. $\qquad \square$

**Proposition A.5** (The OTP problem converges to the hard-constrained problem in the hard-potential limit)**.** *Consider the OTP problem (Eq. 10) with $\mathcal{D} = \mathcal{W}_2^2$ and temporal kernels of the form $\lambda_k(t) = \frac{1}{\tau} \eta_k\left(\frac{t - t_k}{\tau}\right)$, with any fixed nonnegative shapes $\eta_k$ normalized so $\int \eta_k = 1$. Then, in the joint limit $w \to \infty$, $\tau \to 0$ with $w\tau \to 0$,[6] the OTP problem converges to the hard-constrained problem (Eq. 8), and any sequence of OTP minimizers converges to the unique piecewise OT solutions of the Benamou–Brenier subproblems of Lemma A.3.*

*Proof.* We use the fundamental theorem of $\Gamma$-convergence (Braides, 2006): *equi-coercivity* + $\Gamma$-*convergence* $\Rightarrow$ convergence of minimum problems. $\Gamma$-convergence implies that the OTP energies converge to that of the hard-constrained problem, while equi-coercivity is required for the minimizers to converge as well. We prove both ingredients. Note that since we assume absolute continuity of the marginals above, by Villani (2009, Thm. 9.4), the piecewise OT solution of the hard-constrained problem (Lemma A.3) is unique.

**Setup.** We define the family of functionals:

$$F_{w,\tau}[\rho, u] \equiv \int_0^1 \int \left[ \tfrac{1}{2} \|u_t\|^2 \rho_t + w \sum_k \lambda_k(t)\, \mathcal{W}_2^2(\rho_t, \mu_{t_k})\, \rho_t \right] \mathrm{d}x\, \mathrm{d}t, \tag{39}$$

---

[6]The hard-constrained limit only requires that $\tau$ shrink faster than $1/w$ along the sequence considered, e.g., $\tau = w^{-2}$.

fix sequences $w_n \to \infty$, $\tau_n \to 0$ with $w_n \tau_n \to 0$, and write $F_n \equiv F_{w_n, \tau_n}$ and $\lambda_k^n$ for the kernel at width $\tau_n$. We define the limit functional to incorporate the hard constraints as follows:

$$F_{\mathrm{HC}}(\rho, u) \equiv \begin{cases} \int_0^1 \int \frac{1}{2} \|u_t\|^2 \rho_t \, \mathrm{d}x \, \mathrm{d}t & \text{if } (\rho, u) \text{ solves the hard-constrained (HC) problem,} \\ +\infty & \text{otherwise,} \end{cases} \tag{40}$$

where solving the HC problem means that $(\rho, u)$ is admissible as defined above *and* $\rho_{t_k} = \mu_{t_k}$ for all $k$, so $\min F_{\mathrm{HC}} = \mathcal{L}_{\mathrm{HC}}$ (Eq. 8).

**Equi-coercivity.** Equi-coercivity means that all minimizers of $F_n$ must be confined to a compact set independent of $n$; i.e., they cannot escape to infinity if $F_n$ is bounded. Our proof is based on the formal definition from Dal Maso (1993, Prop. 7.7): the sequence $F_n$ is equi-coercive if and only if there exists a lower semicontinuous (l.s.c.) coercive function $\Psi$ such that $F_n \geq \Psi$ for all $n$. We take the kinetic energy as our candidate $\Psi$:

$$\Psi(\rho, u) = \int_0^1 \int \frac{1}{2} \|u_t\|^2 \rho_t \, \mathrm{d}x \, \mathrm{d}t; \tag{41}$$

since the OTP penalty is non-negative, $F_n \geq \Psi$. Also, since $\frac{1}{2}\|u_t\|^2$ is convex and l.s.c., by Ambrosio et al. (2008, Thm. 5.4.4(ii)), $\Psi$ is l.s.c. under this convergence.

What is left to show is that $\Psi$ is coercive, for which we use Dal Maso (1993, Def. 1.12): $\Psi$ is coercive if the set of all bounded energies $\{\Psi(\rho^n, u^n) \leq E\}$ is relatively compact[7] for all $E \in \mathbb{R}$. From Ambrosio et al. (2008, Thm. 8.3.1), if $\Psi(\rho^n, u^n) \leq E$, then

$$\mathcal{W}_2^2(\rho_{t_1}^n, \rho_{t_2}^n) \leq |t_1 - t_2| \int_{t_1}^{t_2} \int \|u_t^n\|^2 \rho_t^n \, \mathrm{d}x \, \mathrm{d}t \leq 2E |t_1 - t_2| \tag{42}$$

$$\Rightarrow \quad \mathcal{W}_2(\rho_{t_1}^n, \rho_{t_2}^n) \leq \sqrt{2E} |t_1 - t_2|^{1/2}, \tag{43}$$

which means all bounded-energy paths are uniformly $\frac{1}{2}$-Hölder continuous in $\mathcal{W}_2$ and thus, *equi-continuous*. Since they all start from the same distribution $\mu_0 \in \mathcal{P}_2$, they also have uniformly bounded second moments throughout time. Thus, by Ascoli–Arzelà theorem (Santambrogio, 2015, Box 1.7), $\{\Psi(\rho^n, u^n) \leq E\}$ is relatively compact, and $\Psi$ coercive. $\Psi$ is therefore an l.s.c. coercive function, which means $F_n$ is equi-coercive.

**$\Gamma$-convergence.** $\Gamma$-convergence requires both *liminf* and *limsup inequalities* (Dal Maso, 1993), essentially showing that the limit of $F_n$ is bounded both below and above by $F_{\mathrm{HC}}$ and, therefore, $F_n$ converges to $F_{\mathrm{HC}}$.

*Limsup inequality.* We seek to show for every $(\rho, u)$ there exists a sequence[8] $(\rho^n, u^n) \to (\rho, u)$ with $\limsup_n F_n(\rho^n, u^n) \leq F_{\mathrm{HC}}(\rho, u)$. If $(\rho, u)$ is not HC-admissible then $F_{\mathrm{HC}}(\rho, u) = +\infty$ and the inequality is trivial. For HC-admissible $(\rho, u)$, we simply consider the constant sequence $(\rho^n, u^n) = (\rho, u)$. By definition, $F_{\mathrm{HC}}(\rho, u)$ equals the kinetic energy $\Psi(\rho, u) \equiv E \in \mathbb{R}$. Subtracting it from $F_n$ leaves only the penalty:

$$F_n(\rho, u) - E = \sum_{k=1}^K w_n \int_0^1 \lambda_k^n(t) \, \mathcal{W}_2^2(\rho_t, \mu_{t_k}) \, \mathrm{d}t. \tag{44}$$

The same bound from Ambrosio et al. (2008, Thm. 8.3.1) as above further yields $\mathcal{W}_2^2(\rho_t, \mu_{t_k}) \leq 2E |t - t_k|$. Substituting this and the explicit form of the temporal kernel with the change of variables $\tilde{t} \equiv \frac{t - t_k}{\tau_n}$:

$$F_n(\rho, u) - E \leq 2E \sum_{k=1}^K w_n \int_0^1 \lambda_k^n(t) |t - t_k| \, \mathrm{d}t = 2E \sum_{k=1}^K w_n \tau_n \int |\tilde{t}| \, \eta_k(\tilde{t}) \, \mathrm{d}\tilde{t} \xrightarrow{n \to \infty} 0, \tag{45}$$

since $w_n \tau_n \to 0$. Thus, $\limsup_n F_n(\rho, u) \leq E = F_{\mathrm{HC}}(\rho, u)$.

---

[7]A set is relatively compact if its closure is compact.

[8]Commonly referred to as a *recovery sequence*.

*Liminf inequality.* We must show that for every sequence $(\rho^n, u^n) \to (\rho, u)$ (uniformly in $\mathcal{W}_2$), $\liminf_n F_n(\rho^n, u^n) \geq F_{\mathrm{HC}}(\rho, u)$. *Case A:* $\rho_{t_k} = \mu_{t_k}$ *for all $k$.* Then $(\rho, u)$ is HC-admissible, so

$$F_{\mathrm{HC}}(\rho, u) \;=\; \int_0^1 \int \tfrac{1}{2} \|u_t\|^2 \rho_t \;\overset{(i)}{\leq}\; \liminf_n \overbrace{\int_0^1 \int \tfrac{1}{2} \|u_t^n\|^2 \rho_t^n}^{\equiv \Psi(\rho^n, u^n)} \;\overset{(ii)}{\leq}\; \liminf_n F_n(\rho^n, u^n), \tag{46}$$

where (i) is lower-semicontinuity of the kinetic energy $\Psi$ under $(\rho^n, u^n) \to (\rho, u)$ (established above), and (ii) follows from $F_n \geq \Psi(\rho^n, u^n)$ since the OTP penalty is non-negative.

*Case B:* $\mathcal{W}_2^2(\rho_{t_{k^*}}, \mu_{t_{k^*}}) > 0$ *for some $k^*$.* Kernel concentration $(\lambda_{k^*}^n \to \delta(t - t_{k^*}))$ and uniform $\mathcal{W}_2$-convergence imply $\int_0^1 \lambda_{k^*}^n(t) \, \mathcal{W}_2^2(\rho_t^n, \mu_{t_{k^*}}) \, \mathrm{d}t \to \mathcal{W}_2^2(\rho_{t_{k^*}}, \mu_{t_{k^*}}) > 0$. Since $F_n \geq w_n$ times this integral and $w_n \to \infty$, we get $F_n(\rho^n, u^n) \to +\infty = F_{\mathrm{HC}}(\rho, u)$.

$\square$

### A.3. Dynamic OT with a potential

**Theorem 3.2.** *Minimizers of the OTP problem (Eq. 10) satisfy the following E-L equations:*

$$u_t(x) = \nabla \varphi_t(x), \tag{11}$$

$$\partial_t \varphi_t(x) + \frac{\|\nabla \varphi_t(x)\|^2}{2} = \sum_{k=1}^K w_k \lambda_k(t) g_k(x, t), \tag{12}$$

*along with the continuity equation, where $g_k(x, t) \equiv \delta \mathcal{D}_t^k / \delta \rho_t(x)$ is the functional derivative of $\mathcal{D}_t^k$ w.r.t. $\rho_{t_k}(x)$.*

*Proof.* We recall the OTP action (Eq. 10) for convenience:

$$S[\mathcal{L}] = \int_0^1 \int_{\mathbb{R}^d} \left[ \tfrac{1}{2} \rho_t(x) \|u_t(x)\|^2 + \sum_{k=1}^K w_k \lambda_k(t) \mathcal{D}_t^k \rho_t(x) + \varphi_t(x) \left[ \partial_t \rho_t(x) + \nabla \cdot (\rho_t(x) u_t(x)) \right] \right] \mathrm{d}x \mathrm{d}t, \tag{47}$$

where $\mathcal{D}_t^k \equiv \mathcal{D}[\rho_t, \mu_{t_k}]$ is a real-valued functional of $\rho_t$ that varies smoothly with $t$ through its dependence on $\rho_t$. Comparing this with Eq. 23, the variations $\delta S / \delta \varphi$ and $\delta S / \delta u$ recover the continuity equation and $u_t = \nabla \varphi_t$ unchanged.

**Variation w.r.t. $\rho_t$.** Recalling from Eq. 10 that the potential contribution to the action is $\iint (-V_{\mathrm{OTP}}(y, s)) \, \mathrm{d}y \, \mathrm{d}s = \sum_k w_k \iint \lambda_k(s) \, \mathcal{D}[\rho_s, \mu_{t_k}] \, \rho_s(y) \, \mathrm{d}y \, \mathrm{d}s$, direct functional differentiation yields:

$$\frac{\delta}{\delta \rho_t(x)} \iint (-V_{\mathrm{OTP}}(y, s)) \, \mathrm{d}y \, \mathrm{d}s = \sum_k w_k \iint \lambda_k(s) \Big[ g_k(x, t) \, \delta(s - t) \underbrace{\rho_s(y)}_{\text{integrates to 1}} + \mathcal{D}[\rho_s, \mu_{t_k}] \, \delta(s - t) \, \delta(x - y) \Big] \mathrm{d}y \, \mathrm{d}s$$

$$= \sum_k w_k \lambda_k(t) \, g_k(x, t) + \underbrace{\sum_k w_k \lambda_k(t) \, \mathcal{D}_t^k}_{\equiv h(t)}, \tag{48}$$

where $g_k(x, t) \equiv \frac{\delta \mathcal{D}_t^k}{\delta \rho_t(x)}$ and $h(t)$ is an $x$-independent function of $t$ that, as we show below, does not impact the dynamics. Thus, following the derivation in App. A.1, the third E-L equation becomes:

$$\partial_t \varphi_t(x) + \frac{\|\nabla \varphi_t(x)\|^2}{2} = \sum_{k=1}^K w_k \lambda_k(t) \, g_k(x, t) + h(t). \tag{49}$$

**Gauge re-parametrization.** Note that $h(t)$ is $x$-independent and can be absorbed into the field $\varphi_t$ by a gauge transformation:

$$\varphi_t(x) \longmapsto \tilde{\varphi}_t(x) := \varphi_t(x) + c(t), \tag{50}$$

leaves $u_t = \nabla\varphi_t = \nabla\tilde\varphi_t$ (and hence the E-L equations of motion and entire flow) unchanged, since $\nabla c(t) = 0$. Choosing $c(t) := -\int_0^t h(s)\,\mathrm{d}s$ so that $c'(t) = -h(t)$, under $\varphi \mapsto \tilde\varphi$ Eq. 49 becomes:

$$\partial_t[\tilde\varphi_t(x) - c(t)] + \frac{\|\nabla\tilde\varphi_t(x)\|^2}{2} = \sum_{k=1}^K w_k\lambda_k(t)\,g_k(x,t) + h(t)$$

$$\Rightarrow \partial_t\tilde\varphi_t(x) + \frac{\|\nabla\tilde\varphi_t(x)\|^2}{2} = \sum_{k=1}^K w_k\lambda_k(t)\,g_k(x,t),$$

(51)

i.e., Eq. 12. Renaming $\tilde\varphi$ back to $\varphi$ (the gauge choice is conventional) yields the stated form. □

**Corollary 3.3.** *Minimizers of the OTP problem (Eq. 10) satisfy the following sample trajectories:*

$$\ddot X_t = \sum_{k=1}^K w_k\lambda_k(t)\nabla g_k(X_t, t).$$

(13)

*Proof.* The proof follows that of Corollary A.2. We first take the gradient of the H-J equation (Eq. 12):

$$\partial_t\nabla\varphi + (\nabla^2\varphi)\,\nabla\varphi = \sum_{k=1}^K w_k\lambda_k(t)\,\nabla g_k(x,t),$$

(52)

where $\nabla^2\varphi$ is the Hessian of $\varphi$ and $\nabla g_k(x,t)$ is the spatial gradient of $g_k$ at fixed $t$. The time derivative of $\dot X_t$ along the flow then gives Eq. 13:

$$\ddot X_t = \frac{\mathrm{d}u(X_t)}{\mathrm{d}t} = \partial_t\nabla\varphi_t(X_t) + (\nabla^2\varphi)(X_t)\,\nabla\varphi_t(X_t) = \sum_{k=1}^K w_k\lambda_k(t)\,\nabla g_k(X_t, t).$$

(53)

□

## A.4. Proofs of Propositions 4.1–4.2

**Proposition 4.1.** *(Marginal alignment bound). For $\mathcal{W}_2^2$-type potentials of strength $w$, away from singular configurations:* $\mathcal{W}_2^2(\rho_{t_k}, \mu_{t_k}) \leq C_k/w^2 + \mathcal{O}(w^{-3})$, *where $C_k$ is a problem-dependent constant independent of $w$.*

*Proof.* By the one-sample-per-marginal estimator of Sec. 4.2, $\Psi_T(X_T) = x_T \equiv [x_{t_1}, \ldots, x_{t_K}]^\top$ for any $z = (x_0, x_1, \mathcal{B}) \sim \pi_{\mathrm{all}}$ with one GT sample $x_{t_k} \sim \mu_{t_k}$ per intermediate marginal. The fixed-point Eq. 20 is equivalent to $X_T = X_T^{\mathrm{base}} + A(X_T - \Psi_T(X_T))$ (App. E.3); substituting $\Psi_T(X_T) = x_T$ and subtracting $x_T$ from both sides yields

$$X_T - x_T = (\mathbb{1} - A)^{-1}(X_T^{\mathrm{base}} - x_T).$$

(54)

$\mathcal{W}_2$ **bound.** Using Eq. 54:

$$\mathcal{W}_2^2(\rho_{t_k}, \mu_{t_k}) \overset{(i)}{\leq} \mathbb{E}\|X_{t_k} - x_{t_k}\|^2 \overset{(ii)}{\leq} \|P_k(\mathbb{1} - A)^{-1}\|^2\,\mathbb{E}\|X_T^{\mathrm{base}} - x_T\|^2.$$

(55)

*(i)* follows by definition of $\mathcal{W}_2^2$ as the *infimum* of $\mathbb{E}\|X - Y\|^2$ over all joint couplings of $(\rho_{t_k}, \mu_{t_k})$ (Villani, 2009). The joint distribution of $(X_{t_k}, x_{t_k})$ induced by $z \sim \pi_{\mathrm{all}}$ is one such coupling, since $X_{t_k}$ has marginal $\rho_{t_k}$ and $x_{t_k}$ has marginal $\mu_{t_k}$ by construction; plugging this specific joint into the squared cost thus gives an upper bound on the infimum. *(ii)* follows by reading off the $k$-th row of Eq. 54, $X_{t_k} - x_{t_k} = \sum_j[(\mathbb{1} - A)^{-1}]_{kj}(X_{t_j}^{\mathrm{base}} - x_{t_j})$, and applying the Cauchy–Schwarz inequality, where $P_k$ is the projection operator onto the $k$-th canonical row vector.

**Scaling in $w$.** Recalling $A_{ik} = w_k[\mathcal{I}^{(2)}[\lambda_k](t_i) - \mathcal{I}^{(2)}[\lambda_k](1)\,t_i]$ from Eq. 20, all $A_{ik}$ scale linearly in the global potential strength $w$, so $A = w\tilde A$ for a fixed matrix $\tilde A$, which depends only on the time grid and temporal kernels. Away from singular configurations (i.e., $\tilde A$ invertible), the Taylor expansion at large $w$ yields:

$$(\mathbb{1} - A)^{-1} = (\mathbb{1} - w\tilde A)^{-1} = -\frac{1}{w}\tilde A^{-1} + \mathcal{O}(w^{-2}),$$

(56)

so $P_k(\mathbb{1} - A)^{-1} = -P_k \tilde{A}^{-1}/w + \mathcal{O}(w^{-2})$, and hence $\|P_k(\mathbb{1} - A)^{-1}\|^2 = \|P_k \tilde{A}^{-1}\|^2/w^2 + \mathcal{O}(w^{-3})$. The data factor $\mathbb{E}\|X_T^{\text{base}} - x_T\|^2$ is independent of $w$ and finite by the second-moment assumption on the GT marginals. Combining with Eq. 55,

$$\mathcal{W}_2^2(\rho_{t_k}, \mu_{t_k}) \leq \frac{C_k}{w^2} + \mathcal{O}(w^{-3}), \qquad C_k = \left\|P_k \tilde{A}^{-1}\right\|^2 \cdot \mathbb{E}\|X_T^{\text{base}} - x_T\|^2, \tag{57}$$

which is the claimed bound. We note that this bound *applies for any choice of joint coupling* $\pi_{\text{all}}$ (e.g., $\pi_{\text{all}}^{\text{OT}}$ or $\pi_{\text{all}}^{\text{ind}}$, Sec. 4.2); only the constant $C_k$ depends on the coupling. $\qquad\square$

**Proposition 4.2.** *(End-to-end bound). For Eulerian and Lagrangian consistency losses (App. C), combining Prop. 4.1 with flow-map learning bounds from Boffi et al. (2025): $\mathcal{W}_2(\rho_{t_k}^{\theta}, \mu_{t_k}) \leq D_k\sqrt{\mathcal{L}_{\text{OTP-FM}}} + \sqrt{C_k}/w + \mathcal{O}(w^{-3/2})$, where $C_k$ and $D_k$ are problem- and consistency-loss-dependent constants, respectively, both independent of $w$.*

*Proof.* We first decompose the distance by the triangle inequality for $\mathcal{W}_2$ (which holds since it is a metric):

$$\mathcal{W}_2(\rho_{t_k}^{\theta}, \mu_{t_k}) \leq \mathcal{W}_2(\rho_{t_k}^{\theta}, \rho_{t_k}) + \mathcal{W}_2(\rho_{t_k}, \mu_{t_k}), \tag{58}$$

where $\rho_{t_k}$ is the OTP target marginal at strength $w$ and $\rho_{t_k}^{\theta} = (\hat{\Psi}_{0,t_k}^{\theta})_{\#}\rho_0$ is the pushforward of the source $\rho_0$ under the learned consistency-model flow map at time $t_k$. Taking the square root of Prop. 4.1, $\mathcal{W}_2(\rho_{t_k}, \mu_{t_k}) \leq \sqrt{C_k}/w + \mathcal{O}(w^{-3/2})$.

**OTP-FM training error as a *flow-map distillation error*.** For the first term, we bound the *flow-map* matching error via the framework of Boffi et al. (2025), viewing the OTP-FM velocity field $u_t$ as the teacher (with associated true two-time flow map $\Psi_{t_1,t_2}^{\theta}$) and the learned MeanFlow consistency model $\Psi_{t_1,t_2}^{\theta}(x) = x + (t_2 - t_1)v^{\theta}(t_1, t_2, x)$ as the distilled student. As we establish in App. C, the conditional OTP-FM training objective (MeanFlow/ESD or LSD; Eqs. 85, 86) is by construction the squared self-distillation residual of one of two consistency conditions on $\Psi_{t_1,t_2}^{\theta}$ (Eulerian or Lagrangian, respectively).

**Wasserstein bound.** We assume $u_t$ satisfies the one-sided Lipschitz condition of Boffi et al. (2025, Assumption 3.3), $(u_t(x) - u_t(y)) \cdot (x - y) \leq C_t \|x - y\|^2$ with $C_t \in L^1[0,1]$ for all $(t,x,y) \in [0,1] \times \mathbb{R}^d \times \mathbb{R}^d$, to guarantee a unique solution to the flow ODE. Then, applying Boffi et al. (2025, Prop. 3.9 (Lagrangian) or Prop. 3.10 (Eulerian)) on the time interval $[0, t_k]$ in place of $[0,1]$ (the proofs run unchanged after restricting all integrals to $[0, t_k]$) yields

$$\mathcal{W}_2^2(\rho_{t_k}^{\theta}, \rho_{t_k}) \overset{(i)}{\leq} \mathbb{E}\left\|\hat{\Psi}_{0,t_k}^{\theta}(x_0) - \Psi_{0,t_k}(x_0)\right\|^2 \overset{(ii)}{\leq} \kappa_k \mathcal{L}_{\text{OTP-FM}}^{\text{M}}(\theta), \tag{59}$$

where *(i)* follows because $\mathcal{W}_2^2$, by definition, is the infimum of all joint couplings, of which $(\hat{\Psi}_{0,t_k}^{\theta}(x_0), \Psi_{0,t_k}(x_0))$ is a valid one; and *(ii)* is the corresponding bound from Boffi et al. (2025) with constants:

$$\kappa_k = e^{t_k + 2\int_0^{t_k}|C_t|\,\mathrm{d}t} \quad \text{(LSD)}, \qquad \kappa_k = e^{t_k} \quad \text{(ESD/MeanFlow)}. \tag{60}$$

Substituting this into Eq. 58 yields the claimed bound:

$$\mathcal{W}_2(\rho_{t_k}^{\theta}, \mu_{t_k}) \leq D_k\sqrt{\mathcal{L}_{\text{OTP-FM}}} + \frac{\sqrt{C_k}}{w} + \mathcal{O}(w^{-3/2}), \tag{61}$$

where $D_k = \sqrt{\kappa_k}$ and $C_k$ are the constants from Prop. 4.1. $\qquad\square$

## B. Distances between probability measures

### B.1. The $\mathcal{W}_2^2$ distance, KLD, and MMD

The potentials we consider in this work are based on statistical distances between two probability measures $\mathcal{D} : \mathcal{P}(\mathbb{R}^d) \times \mathcal{P}(\mathbb{R}^d) \to \mathbb{R}^+$. Specifically, we consider three popular statistical distances. The first is the Kullback-Leibler divergence (KLD):

$$\mathcal{D}_{\text{KL}}[\alpha, \beta] = \int_{\mathbb{R}^d} \ln\frac{\alpha(x)}{\beta(x)}\mathrm{d}\alpha(x). \tag{62}$$

The second is the $\mathcal{W}_2^2$ distance, a form of the static OT problem discussed in Sec. 2.1, and the third the maximum mean discrepancy (MMD), a type of integral probability metric (IPM). IPMs are defined as the difference in expectation of an optimal witness function $h : \mathbb{R}^d \to \mathbb{R}$ out of a class of real-valued measurable functions $\mathcal{H}$:

$$\mathcal{D}_{\text{IPM}}[\alpha, \beta] = \sup_{h \in \mathcal{H}} \left|\int_{\mathbb{R}^d} h(x)\mathrm{d}\alpha(x) - \int_{\mathbb{R}^d} h(x)\mathrm{d}\beta(x)\right|. \tag{63}$$

They include the $\mathcal{W}_1$ distance and have the attractive property of satisfying the definition of a metric for most common choices of $\mathcal{H}$ (Müller, 1997). In the case of MMD, $\mathcal{H}$ is a unit ball in a reproducing kernel Hilbert space (RKHS) with kernel $k : \mathbb{R}^d \times \mathbb{R}^d \to \mathbb{R}$. An RKHS satisfies the reproducing property: $\forall h \in \mathcal{H}$, $h(x) = \langle h, k(x, \cdot) \rangle_\mathcal{H}$. The squared MMD can be expressed conveniently as (Gretton et al., 2012):

$$\mathcal{D}_{\mathrm{MMD}^2}[\alpha, \beta] = \iint k(x, x') \, \mathrm{d}\alpha(x)\mathrm{d}\alpha(x') - 2 \iint k(x, y) \, \mathrm{d}\alpha(x)\mathrm{d}\beta(y) + \iint k(y, y') \, \mathrm{d}\beta(y)\mathrm{d}\beta(y'). \tag{64}$$

### B.2. Functional derivatives

As highlighted in Sec. 3, the functional derivatives $g$ of statistical distances $\mathcal{D}$, and their gradients $\nabla g$, are important objects for OTP-FM. The expressions for these for the $\mathcal{W}_2^2$ distance, MMD, and KL divergence, provided in Table 1, can be derived by direct differentiation as we detail here.

**The $\mathcal{W}_2^2$ distance**  The derivative can be calculated most easily using the dynamic *dual form* of $\mathcal{W}_2^2$ (Villani, 2009):

$$\mathcal{D}[\rho_t, \mu_{t_k}] := \sup_{\varphi_t} \left\{ \int_{\mathbb{R}^d} \varphi_1(x)\mathrm{d}\mu_{t_k}(x) - \int_{\mathbb{R}^d} \varphi_0(x)\mathrm{d}\rho_t(x) \right\}, \tag{65}$$

such that $\forall t \in [0, 1]$, $\varphi_t$ satisfies the H-J equation (Eq. 3). Taking the functional derivative with respect to $\rho_t(x)$, we get:

$$\frac{\delta \mathcal{D}}{\delta \rho_t} \equiv g_k(x, t) = -\varphi_0^*(x), \tag{66}$$

where $\varphi^*$ is the optimal H-J potential. Finally, using Theorem A.1 and Corollary A.2, we have that:

$$\nabla g_k(x, t) = -\nabla \varphi_0^*(x) = -u_0(x) = x - \psi^*(x), \tag{67}$$

where $\psi^*$ is the optimal (static) OT map between $\rho_t$ and $\mu_{t_k}$.

**MMD**  Differentiating Eq. 64 with respect to $\rho_t(x)$, we get:

$$\frac{\delta \mathcal{D}_{\mathrm{MMD}^2}[\rho_t, \mu_{t_k}]}{\delta \rho_t} = 2 \int k(x, y) \left[ \rho_t(y) - \mu_{t_k}(y) \right] \mathrm{d}y. \tag{68}$$

The gradient, for arbitrary kernel $k$, is then simply:

$$\nabla g_k(x, t) = 2 \int \nabla_x k(x, y) \left[ \rho_t(y) - \mu_{t_k}(y) \right] \mathrm{d}y. \tag{69}$$

Specifically, in this work, we primarily consider the well-known RBF kernel:

$$k_{\mathrm{RBF}}(x, y) = \exp\left( -\frac{\|x - y\|^2}{2\sigma^2} \right), \tag{70}$$

where $\sigma$ is the kernel bandwidth; and experiment briefly with the polynomial kernel (see App. D):

$$k_{\mathrm{poly}}(x, y) = (x^T y + c)^d, \tag{71}$$

where $d$ is the polynomial degree and $c$ a constant. The gradients for these kernels are:

$$\nabla_x k_{\mathrm{RBF}}(x, y) = -\frac{x - y}{\sigma^2} \cdot \exp\left( -\frac{\|x - y\|^2}{2\sigma^2} \right), \tag{72}$$

$$\nabla_x k_{\mathrm{poly}}(x, y) = d \cdot y \cdot (x^T y + c)^{d-1}. \tag{73}$$

**KL divergence**    We consider only the "forward" KL divergence as it has a more useful derivative and gradient:

$$\mathcal{D}_{\mathrm{KL}}[\rho_t, \mu_{t_k}] = \int_{\mathbb{R}^d} \rho_t(x) \left[\ln \rho_t(x) - \ln \mu_{t_k}(x)\right] \mathrm{d}x. \tag{74}$$

Taking the functional derivative:

$$\frac{\delta \mathcal{D}_{\mathrm{KL}}}{\delta \rho_t} = \ln \rho_t(x) + 1 - \ln \mu_{t_k}(x). \tag{75}$$

The gradient is then simply:

$$\nabla g_k(x,t) = \nabla \ln \rho_t(x) - \nabla \ln \mu_{t_k}(x). \tag{76}$$

## C. Consistency models and MeanFlow

### C.1. Flow map matching

Consistency models have been presented in the literature in a wide variety of forms, with variations on, among other things: discrete- vs. continuous-time training; the use of one- vs. two-time flow maps; the requirement (or lack thereof) for a pre-trained "teacher" flow matching model; and training objectives.

*Flow map matching* (Boffi et al., 2025; 2026) provides a useful framework unifying many SOTA continuous-time approaches, such as that of Song et al. (2023), consistency trajectory models (Kim et al., 2024), shortcut models (Frans et al., 2025), and Align Your Flow (Sabour et al., 2025). These models are generally trained via objectives based on one of three self-consistent conditions required for a valid flow map $\Psi_{t_1,t_2}$.

$$\begin{array}{rl}
\text{Semigroup condition:} & \psi_{t_2,t_3}(\psi_{t_1,t_2}(x)) = \psi_{t_1,t_3}(x), \tag{77} \\
\text{Eulerian condition:} & \partial_{t_1} \Psi_{t_1,t_2}(x) + \nabla \Psi_{t_1,t_2}(x) u_{t_1}(x) = 0, \tag{78} \\
\text{Lagrangian condition:} & \partial_{t_2} \Psi_{t_1,t_2}(x) = u_{t_2}(\Psi_{t_1,t_2}(x)), \tag{79}
\end{array}$$

for all $(t_1, t_2, t_3) \in [0,1]^3$ and $x \in \mathbb{R}^d$, where $u_t(x)\colon [0,1] \times \mathbb{R}^d \to \mathbb{R}^d$ is the instantaneous velocity field satisfying $u_t(X_t) = \frac{\mathrm{d}X_t}{\mathrm{d}t}$ along a trajectory $X_{t_2} = \Psi_{t_1,t_2}(X_{t_1})$. Shortcut models, for example, are trained to satisfy the semigroup condition, while most other consistency models the Eulerian; the Lagrangian condition is a relatively newer idea introduced by Boffi et al. (2025; 2026). In our experiments, we consider only the Eulerian and Lagrangian conditions for their desirable theoretical properties (Boffi et al., 2026) and superior performance in computer vision.

By substituting the flow map parametrization in terms of the mean velocity $v_{t_1,t_2}\colon [0,1]^2 \times \mathbb{R}^d \to \mathbb{R}^d$:

$$\Psi_{t_1,t_2}(X_{t_1}) = X_{t_1} + (t_2 - t_1)v_{t_1,t_2}(X_{t_1}), \tag{80}$$

we obtain the Eulerian and Lagrangian conditions in terms of $v_{t_1,t_2}$:

$$\begin{array}{rl}
\text{Eulerian condition:} & v_{t_1,t_2}(x) = u_{t_1}(x) + (t_2 - t_1)(u_{t_1}(x)\,\partial_x v_{t_1,t_2} + \partial_{t_1} v_{t_1,t_2}). \tag{81} \\
\text{Lagrangian condition:} & v_{t_1,t_2}(x) = u_{t_2}(\Psi_{t_1,t_2}(x)) - (t_2 - t_1)\partial_{t_2} v_{t_1,t_2}(x). \tag{82}
\end{array}$$

**Flow map self-distillation.**    We can next inject a training target, referred to as a "teacher" flow map, into these conditions, to obtain a training objective to *distill* the teacher into a consistency model $\Psi^\theta_{t_1,t_2}$ (or, equivalently, $v^\theta_{t_1,t_2}$). Generally, distillation requires a pre-trained flow matching model as the teacher; however, in this work we focus exclusively on the *self-distillation* paradigm, wherein the "teacher" target is formed directly from the CFM-style conditional velocity targets $u_t(x|z)$ (modified in Sec. 4.1 for OTP-FM). Explicitly, we consider the Eulerian and Lagrangian self-distillation (ESD and LSD) training objectives:

$$\mathcal{L}_{\mathrm{ESD}}(\theta) := \mathbb{E}_{t_1,t_2,\, x_{t_1} \sim \rho_{t_1}(\cdot|z),\, z \sim q} \left\| v^\theta_{t_1,t_2}(x_{t_1}) - \left[u_{t_1}(x_{t_1}|z) + (t_2 - t_1)(u_{t_1}(x_{t_1}|z)\partial_x v^\theta_{t_1,t_2} + \partial_{t_1} v^\theta_{t_1,t_2})\right] \right\|^2, \tag{83}$$

$$\mathcal{L}_{\mathrm{LSD}}(\theta) := \mathbb{E}_{t_1,t_2,\, x_{t_1} \sim \rho_{t_1}(\cdot|z),\, z \sim q} \left\| v^\theta_{t_1,t_2}(x_{t_1}) - \left[u_{t_2}(\Psi_{t_1,t_2}(x_{t_1})|z) - (t_2 - t_1)\partial_{t_2} v^\theta_{t_1,t_2}(x_{t_1})\right] \right\|^2, \tag{84}$$

where we see that we aim to minimize the residual between the LHS and RHS of the Eulerian and Lagrangian conditions through a mean-squared error regression loss, with the ground-truth training signal injected via $u_t(x|z)$.

As in many (continuous-time) consistency models, this objective involves derivatives of $v^\theta$ with respect to its inputs $(x, t_1, t_2)$; these can be computed efficiently using the Jacobian-vector product (JVP) operation implemented in modern automatic differentiation libraries such as PyTorch and JAX. Furthermore, to avoid "double-backpropagation" through the JVP and improve training stability, it is standard practice to not backpropagate through the RHS term, represented by the "stop-gradient" operator sg:

$$\mathcal{L}_{\text{ESD}}(\theta) := \mathbb{E}_{t_1, t_2, x_{t_1} \sim \rho_{t_1}(\cdot|z), z \sim q} \left\| v^\theta_{t_1, t_2}(x_{t_1}) - \text{sg}\left[ u_{t_1}(x_{t_1}|z) + (t_2 - t_1)(u_{t_1}(x_{t_1}|z)\partial_x v^\theta_{t_1, t_2} + \partial_{t_1} v^\theta_{t_1, t_2}) \right] \right\|^2, \quad (85)$$

$$\mathcal{L}_{\text{LSD}}(\theta) := \mathbb{E}_{t_1, t_2, x_{t_1} \sim \rho_{t_1}(\cdot|z), z \sim q} \left\| v^\theta_{t_1, t_2}(x_{t_1}) - \text{sg}\left[ u_{t_2}(\Psi_{t_1, t_2}(x_{t_1})|z) - (t_2 - t_1)\partial_{t_2} v^\theta_{t_1, t_2}(x_{t_1}) \right] \right\|^2, \quad (86)$$

We note our stop-gradient version of LSD differs slightly from the original in Boffi et al. (2026) with the time-derivative moved inside the sg operator for further stability. Unlike ESD, LSD does not involve a spatial derivative through $\Psi_{t_1, t_2}$ or the learnt model, which can in principle improve training stability. We experiment with both (using the MeanFlow and improved MeanFlow (iMF) instantiations of ESD below) and find iMF to be most performant, as shown in Apps. F and G.

### C.2. MeanFlow and improved MeanFlow

The MeanFlow objective is an instantiation of the Eulerian condition that has proven particularly performant; indeed, as of this writing, its improved variant is SOTA in one- and two-step generative modeling. The original formulation focuses on the average velocity object between two time points $t_1$ and $t_2$:

$$v_{t_1, t_2}(X_{t_1}) = \frac{1}{t_2 - t_1} \int_{t_1}^{t_2} u_t(X_t) \mathrm{d}t, \quad (87)$$

but is equivalent to the flow-map based description above and in Sec. 2.3.

One modification we make to the original MeanFlow objective is to privilege $t_1$ instead of $t_2$, in order to flow *forward* instead of *backward* in time. The corresponding regression target is derived as follows, following the same steps as Geng et al. (2026a).

First, multiplying both sides of Eq. 87 by $t_2 - t_1$:

$$(t_2 - t_1)v_{t_1, t_2}(X_{t_1}) = \int_{t_1}^{t_2} u_t(X_t)\mathrm{d}t. \quad (88)$$

Next, taking the derivative of both sides with respect to $t_1$ (instead of $t_2$) and rearranging:

$$v_{t_1, t_2}(X_{t_1}) = u_{t_1}(X_{t_1}) + (t_2 - t_1)\frac{\mathrm{d}}{\mathrm{d}t_1} v_{t_1, t_2}(X_{t_1}). \quad (89)$$

Finally, expanding out the time derivative in terms of the partial derivatives, and using $\frac{\mathrm{d}X_{t_1}}{\mathrm{d}t_1} = u_{t_1}(X_{t_1})$, $\frac{\mathrm{d}t_1}{\mathrm{d}t_1} = 1$, and $\frac{\mathrm{d}t_2}{\mathrm{d}t_1} = 0$, we obtain the final regression target:

$$v_{t_1, t_2}(X_{t_1}) = u_{t_1}(X_{t_1}) + (t_2 - t_1)(u_{t_1}(X_{t_1})\partial_x v_{t_1, t_2} + \partial_{t_1} v_{t_1, t_2}). \quad (90)$$

As stated above, this yields the exact same regression target as ESD (Eq. 85).

More recently, Geng et al. (2026b) propose the "improved MeanFlow" (iMF) objective, with the primary difference being the replacement of the second instance of $u_{t_1}(x_{t_1}|z)$ in the MeanFlow target with the instantaneous velocity prediction from the model itself $v^\theta_{t_1, t_1}(x_{t_1})$.[9] Incorporating this and rearranging terms to match their convention, we obtain the iMF objective:

$$\mathcal{L}_{\text{iMF}}(\theta) := \mathbb{E}_{t_1, t_2, x_{t_1} \sim \rho_{t_1}(\cdot|z), z \sim q} \left\| \underbrace{v^\theta_{t_1, t_2}(x_{t_1}) - (t_2 - t_1)\text{sg}\left[v^\theta_{t_1, t_1}(x_{t_1})\partial_x v^\theta_{t_1, t_2} + \partial_{t_1} v^\theta_{t_1, t_2}\right]}_{\equiv V^\theta_{t_1, t_2}} - u_{t_1}(x_{t_1}|z) \right\|^2, \quad (91)$$

where $V^\theta_{t_1, t_2}$ is a compound function of $v^\theta_{t_1, t_2}$ that directly regresses the conditional instantaneous velocity regression target. Geng et al. (2026b) argue this substitution of $v^\theta_{t_1, t_1}(x_{t_1})$ for $u_{t_1}(x_{t_1}|z)$ reduces the regression loss variance and improves training stability. Empirically, we find this modestly outperforms MeanFlow and LSD, as shown in App. G.

---

[9]Alternatively, they propose the possibility of using an auxiliary head to predict this instantaneous velocity, but we elect to use a single head inputting $t_1$ for both time inputs.

# D. Solutions for Gaussian marginals

For the special case of $d$-dimensional isotropic Gaussian marginals, we can derive closed-form solutions for the OTP problem under the ansatz that $\rho_t$ is an isotropic Gaussian with time-dependent mean $m_{\rho_t}$ and variance $\sigma_{\rho_t}^2 \mathbb{1}_d$. In this case, following Lipman et al. (2023), the marginal velocity takes the form:

$$u_t(x) = \frac{\dot{\sigma}_{\rho_t}}{\sigma_{\rho_t}}(x - m_{\rho_t}) + \dot{m}_{\rho_t}. \tag{92}$$

This means the dynamic OT kinetic energy reduces to:

$$
\begin{aligned}
T(x) &= \frac{1}{2} \int \|u_t(x)\|^2 \mathrm{d}\rho_t(x) = \frac{1}{2}\mathbb{E}\|u_t(x)\|^2 \\
&= \frac{1}{2}\left[ \frac{\dot{\sigma}_{\rho_t}^2}{\sigma_{\rho_t}^2} \underbrace{\mathbb{E}\|x - m_{\rho_t}\|^2}_{=\mathrm{tr}(\sigma_{\rho_t}^2 \mathbb{1}_d)=d\sigma_{\rho_t}^2} + 2 \cdot \frac{\dot{\sigma}_{\rho_t}}{\sigma_{\rho_t}} \cdot \dot{m}_{\rho_t} \cdot \underbrace{\mathbb{E}(x - m_{\rho_t})}_{=0} + \|\dot{m}_{\rho_t}\|^2 \right] \\
&= \frac{1}{2}(\|\dot{m}_{\rho_t}\|^2 + d\dot{\sigma}_{\rho_t}^2).
\end{aligned}
\tag{93}
$$

Each distance $\mathcal{D}_t^k \equiv \mathcal{D}[\rho_t, \mu_{t_k}]$ similarly reduces to a function of $(m_{\rho_t}, \sigma_{\rho_t})$ alone, as derived explicitly for different distance metrics below. In general, this means a Lagrangian of the form:

$$\mathcal{L}(m_{\rho_t}, \sigma_{\rho_t}, \dot{m}_{\rho_t}, \dot{\sigma}_{\rho_t}, t) = \frac{1}{2}\left( \|\dot{m}_{\rho_t}\|^2 + d\dot{\sigma}_{\rho_t}^2 \right) + \sum_{k=1}^{K} w_k \lambda_k(t)\, \mathcal{D}_t^k, \tag{94}$$

where the distance $\mathcal{D}_t^k$ depends only on $(m_{\rho_t}, \sigma_{\rho_t})$ and the fixed parameters $(m_{\mu_{t_k}}, \sigma_{\mu_{t_k}})$ of the intermediate marginal. The E-L equations for the generalized coordinates $(m_{\rho_t}, \sigma_{\rho_t})$ are therefore:

$$\ddot{m}_{\rho_t} = \sum_{k=1}^{K} w_k \lambda_k(t)\, \nabla_{m_{\rho_t}} \mathcal{D}_t^k, \tag{95}$$

$$\ddot{\sigma}_{\rho_t} = \frac{1}{d} \sum_{k=1}^{K} w_k \lambda_k(t)\, \frac{\partial \mathcal{D}_t^k}{\partial \sigma_{\rho_t}}. \tag{96}$$

By plugging in the explicit forms for the distances and gradients derived below, this yields second-order ODEs for the time evolution of $m_{\rho_t}$ and $\sigma_{\rho_t}$ that can be efficiently simulated numerically. To solve the boundary value problem, matching $m_{\rho_0} = m_{\mu_0}$, $\sigma_{\rho_0} = \sigma_{\mu_0}$, $m_{\rho_1} = m_{\mu_1}$, and $\sigma_{\rho_1} = \sigma_{\mu_1}$, we use the shooting method (Stoer & Bulirsch, 1980) to reduce it to finding the initial conditions that solve an initial value problem yielding the desired final conditions. Specifically, we use an implicit Runge-Kutta method (Hairer & Wanner, 1996) (the `Radau` method implemented in the `scipy.integrate` library (Virtanen et al., 2020)) to simulate the E-L equations as first-order ODEs in the velocities of the mean and standard deviation, and "Broyden's good method" (van de Rotten, 2003) (`scipy.optimize.broyden1`) to solve for the initial velocities. The exact code is provided in the linked `OTP-FM` repository.

## D.1. The $\mathcal{W}_2^2$ distance

The $\mathcal{W}_2^2$ distance between two Gaussians $\rho_t$ and $\mu_{t_k}$ has the well-known form — also known as the Fréchet distance — that reduces in the case of isotropic Gaussians to:

$$\mathcal{D}_{\mathcal{W}_2^2}[\rho_t, \mu_{t_k}] = \|m_{\rho_t} - m_{\mu_{t_k}}\|^2 + d(\sigma_{\rho_t} - \sigma_{\mu_{t_k}})^2. \tag{97}$$

The gradients are simply:

$$\nabla_{m_{\rho_t}} \mathcal{D}_t^k = 2(m_{\rho_t} - m_{\mu_{t_k}}), \tag{98}$$

$$\frac{\partial \mathcal{D}_t^k}{\partial \sigma_{\rho_t}} = 2d(\sigma_{\rho_t} - \sigma_{\mu_{t_k}}). \tag{99}$$

### D.2. KL divergence

The KL divergence between two Gaussians $\rho_t$ and $\mu_{t_k}$ similarly has a known form (Pardo, 2006), reducing to:

$$\mathcal{D}_{\mathrm{KL}}[\rho_t \| \mu_{t_k}] = \frac{1}{2}\left[d\ln\frac{\sigma_{\mu_{t_k}}^2}{\sigma_{\rho_t}^2} + d\frac{\sigma_{\rho_t}^2}{\sigma_{\mu_{t_k}}^2} + \frac{\|m_{\rho_t} - m_{\mu_{t_k}}\|^2}{\sigma_{\mu_{t_k}}^2} - d\right], \tag{100}$$

with gradients:

$$\nabla_{m_{\rho_t}}\mathcal{D}_t^k = \frac{m_{\rho_t} - m_{\mu_{t_k}}}{\sigma_{\mu_{t_k}}^2}, \tag{101}$$

$$\frac{\partial \mathcal{D}_t^k}{\partial \sigma_{\rho_t}} = d\left(\frac{\sigma_{\rho_t}}{\sigma_{\mu_{t_k}}^2} - \frac{1}{\sigma_{\rho_t}}\right). \tag{102}$$

### D.3. MMD (RBF kernel)

We first consider MMD with the RBF kernel (Eq. 70) with bandwidth $\gamma$. The squared MMD between two distributions is (Eq. 64):

$$\mathcal{D}_{\mathrm{MMD}^2}[\rho_t, \mu_{t_k}] = \mathbb{E}_{x,x'\sim\rho_t}[k(x,x')] - 2\mathbb{E}_{x\sim\rho_t,y\sim\mu_{t_k}}[k(x,y)] + \mathbb{E}_{y,y'\sim\mu_{t_k}}[k(y,y')]. \tag{103}$$

For the self-terms, if $x, x' \sim \mathcal{N}(m, \sigma^2 I_d)$ are independent, then $x - x' \sim \mathcal{N}(0, 2\sigma^2 I_d)$. Using the moment generating function of a chi-squared distribution:

$$\mathbb{E}_{x,x'\sim\mathcal{N}(m,\sigma^2 I)}[k(x,x')] = \mathbb{E}\left[\exp\left(-\frac{\|x-x'\|^2}{2\gamma^2}\right)\right] = \left(\frac{\gamma^2}{\gamma^2 + 2\sigma^2}\right)^{d/2}. \tag{104}$$

For the cross-term, if $x \sim \mathcal{N}(m_1, \sigma_1^2 I_d)$ and $y \sim \mathcal{N}(m_2, \sigma_2^2 I_d)$ are independent, then $x - y \sim \mathcal{N}(m_1 - m_2, (\sigma_1^2 + \sigma_2^2)I_d)$. Hence:

$$\mathbb{E}_{x\sim\rho_t,y\sim\mu_{t_k}}[k(x,y)] = \left(\frac{\gamma^2}{\gamma^2 + \sigma_{\rho_t}^2 + \sigma_{\mu_{t_k}}^2}\right)^{d/2}\exp\left(-\frac{\|m_{\rho_t} - m_{\mu_{t_k}}\|^2}{2(\gamma^2 + \sigma_{\rho_t}^2 + \sigma_{\mu_{t_k}}^2)}\right). \tag{105}$$

Defining the effective variances $\tau_\rho^2 = \gamma^2 + 2\sigma_{\rho_t}^2$, $\tau_\mu^2 = \gamma^2 + 2\sigma_{\mu_{t_k}}^2$, and $\tau_+^2 = \gamma^2 + \sigma_{\rho_t}^2 + \sigma_{\mu_{t_k}}^2$, and writing $\Delta m = m_{\rho_t} - m_{\mu_{t_k}}$:

$$\mathcal{D}_{\mathrm{MMD}^2}[\rho_t, \mu_{t_k}] = \left(\frac{\gamma^2}{\tau_\rho^2}\right)^{d/2} + \left(\frac{\gamma^2}{\tau_\mu^2}\right)^{d/2} - 2\left(\frac{\gamma^2}{\tau_+^2}\right)^{d/2}\exp\left(-\frac{\|\Delta m\|^2}{2\tau_+^2}\right). \tag{106}$$

Finally, the gradients are:

$$\nabla_{m_{\rho_t}}\mathcal{D}_{\mathrm{MMD}^2} = \frac{2\Delta m}{\tau_+^2}\left(\frac{\gamma^2}{\tau_+^2}\right)^{d/2}\exp\left(-\frac{\|\Delta m\|^2}{2\tau_+^2}\right), \tag{107}$$

$$\frac{\partial \mathcal{D}_{\mathrm{MMD}^2}}{\partial \sigma_{\rho_t}} = -2d\sigma_{\rho_t}\left(\frac{\gamma^2}{\tau_\rho^2}\right)^{d/2}\frac{1}{\tau_\rho^2} + 2\sigma_{\rho_t}\left(\frac{\gamma^2}{\tau_+^2}\right)^{d/2}\exp\left(-\frac{\|\Delta m\|^2}{2\tau_+^2}\right)\left[\frac{d}{\tau_+^2} - \frac{\|\Delta m\|^2}{\tau_+^4}\right]. \tag{108}$$

### D.4. MMD (polynomial kernel)

We next consider the polynomial kernel (Eq. 71) with degree 2. Higher-degree kernels can be computed similarly but are more cumbersome.

As the quadratic kernel is $k(x,y) = (x^\top y + c)^2$, we first compute $\mathbb{E}[(x^\top y)^2]$. For independent $x, x' \sim \mathcal{N}(m, \sigma^2 I_d)$:

$$\mathbb{E}[(x^\top x')^2] = \|m\|^4 + d\sigma^4 + 2\sigma^2\|m\|^2. \tag{109}$$

For independent $x \sim \mathcal{N}(m_1, \sigma_1^2 I_d)$ and $y \sim \mathcal{N}(m_2, \sigma_2^2 I_d)$:

$$\mathbb{E}[(x^\top y)^2] = (m_1^\top m_2)^2 + d\sigma_1^2\sigma_2^2 + \sigma_1^2\|m_2\|^2 + \sigma_2^2\|m_1\|^2. \tag{110}$$

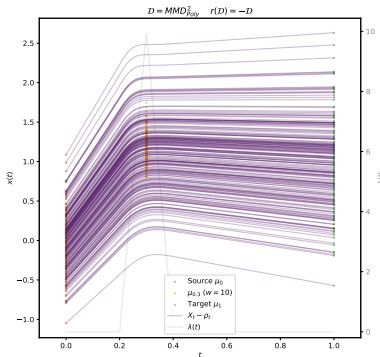

*Figure 4.* A characteristic example of the exact dynamic OT solution (under the Gaussian ansatz) with MMD and a polynomial kernel failing to enforce the intermediate marginal at high strength.

Hence, combining all terms with $\mathbb{E}[(x^\top y + c)^2] = \mathbb{E}[(x^\top y)^2] + 2c\mathbb{E}[x^\top y] + c^2$:

$$
\begin{aligned}
\mathcal{D}_{\mathrm{MMD}^2}[\rho_t, \mu_{t_k}] =\ & \|m_{\rho_t}\|^4 + d\sigma_{\rho_t}^4 + 2\sigma_{\rho_t}^2\|m_{\rho_t}\|^2 + 2c\|m_{\rho_t}\|^2 + c^2 \\
& - 2\Big[(m_{\rho_t}^\top m_{\mu_{t_k}})^2 + d\sigma_{\rho_t}^2\sigma_{\mu_{t_k}}^2 + \sigma_{\rho_t}^2\|m_{\mu_{t_k}}\|^2 + \sigma_{\mu_{t_k}}^2\|m_{\rho_t}\|^2 + 2cm_{\rho_t}^\top m_{\mu_{t_k}} + c^2\Big] \\
& + \|m_{\mu_{t_k}}\|^4 + d\sigma_{\mu_{t_k}}^4 + 2\sigma_{\mu_{t_k}}^2\|m_{\mu_{t_k}}\|^2 + 2c\|m_{\mu_{t_k}}\|^2 + c^2.
\end{aligned}
\tag{111}
$$

Finally, the gradients are:

$$
\nabla_{m_{\rho_t}}\mathcal{D}_{\mathrm{MMD}^2} = 4\Big[(\|m_{\rho_t}\|^2 + \sigma_{\rho_t}^2 - \sigma_{\mu_{t_k}}^2 + c)m_{\rho_t} - (m_{\rho_t}^\top m_{\mu_{t_k}} + c)m_{\mu_{t_k}}\Big],
\tag{112}
$$

$$
\frac{\partial \mathcal{D}_{\mathrm{MMD}^2}}{\partial \sigma_{\rho_t}} = 4\sigma_{\rho_t}\Big[d(\sigma_{\rho_t}^2 - \sigma_{\mu_{t_k}}^2) + (\|m_{\rho_t}\|^2 - \|m_{\mu_{t_k}}\|^2)\Big].
\tag{113}
$$

We find this polynomial kernel to not be performant, even for this simple case of Gaussian marginals. A characteristic example is shown in Fig. 4, for the same marginal configuration as in the second MMD RBF example in Fig. 2. As we see, while the force of the polynomial kernel drives the mean as expected, it is not as effective on the variance; hence, we do not pursue it further in this work.

## E. Details of the OTP-FM algorithm

### E.1. Time sampling

We sample the training times $t_1$ and $t_2$ according to the joint distribution:

$$
p_{t_1,t_2} = \frac{1}{K+1}\sum_{k=0}^{K}\frac{\mathbb{1}_{[t_k,t_{k+1}]}(t_1)}{t_{k+1}-t_k}\cdot\frac{\mathbb{1}_{[t_1,1]}(t_2)}{1-t_1};
\tag{114}
$$

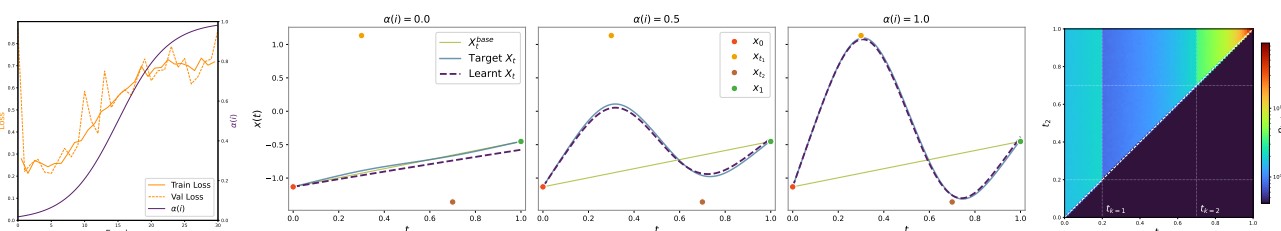

*Figure 5. Left:* Example loss curves and $\alpha(i)$ schedule. *Center:* Progression of target conditional trajectories during training for the $\mathcal{W}_2^2$ distance based on example samples from four 1D marginals. *Right:* Example time sampling distribution $p_{t_1,t_2}$.

i.e., uniformly sampling $t_1$ but with each of the $K + 1$ intervals between consecutive marginals weighted equally, and $t_2$ uniformly on the interval $[t_1, 1]$, so that $t_2 > t_1$ by construction (Fig. 5). This follows the convention of most consistency models of training only the upper or lower triangle of $v^\theta(t_1, t_2)$; however, we note there is no requirement in general to restrict $t_2 > t_1$ and, indeed, allowing $t_2 < t_1$ may be desirable in some applications for flowing "backwards". We use positional embeddings (Vaswani et al., 2017) to encode $t_1$ and $\Delta t = t_2 - t_1$ as inputs to the model.

### E.2. Fixed-point iteration convergence and homotopy continuation

As stated in Sec. 4.3, the curriculum on the correction strength in OTP-FM is an application of the homotopy continuation method (Allgower & Georg, 2003). Here, we analyze the convergence of the FP iterations and why the homotopy curriculum $\alpha(i)$ helps stabilize them.

**Contraction analysis**   Stacking the $K$ intermediate positions as $X_T \equiv [X_{t_1}, \ldots, X_{t_K}]^\top$, the FP iterations of Eq. 21 can be written as $X_T^{(m+1)} = F_\alpha(X_T^{(m)})$ with

$$F_\alpha(X_T)_i = X_{t_i}^{\mathrm{base}} + \alpha \sum_{k=1}^K A_{ik} \nabla g_k(X_{t_k}, t_k, \mathcal{B}), \tag{115}$$

where $A_{ik}$ is as in Eq. 20. If each force $\nabla g_k(\cdot, t_k, \mathcal{B})$ is Lipschitz-continuous with constant $L_k$, then the Jacobian of $F_\alpha$ is uniformly bounded:

$$\|J F_\alpha\| \leq \alpha \max_i \sum_{k=1}^K |A_{ik}| L_k \equiv L_F. \tag{116}$$

By the Banach fixed-point theorem, whenever $L_F < 1$, $F_\alpha$ is a contraction and the iterations $X_T^{(m)}$ converge linearly to a unique fixed point at rate at most $L_F$.

**Lipschitz constants**   We can compute $L_k$ directly from the Jacobians of the gradient estimators in Table 1:

- $\mathcal{W}_2^2$: $\nabla g_k(X_{t_k}, t_k, \mathcal{B}) = X_{t_k} - x_{t_k}$ with $x_{t_k}$ fixed (Sec. 4.2), so $L_k^{\mathcal{W}_2^2} = 1$ and $F_\alpha$ is in fact linear in $X_T$; we solve it directly by matrix inversion (App. E.3) and FP iterations are unnecessary.

- $\mathcal{D}_{\mathrm{MMD}^2}$ with an RBF kernel $k_{\sigma_k}(x, y) = \exp(-\|x - y\|^2/2\sigma_k^2)$: $L_k^{\mathrm{MMD}} \leq 4/\sigma_k^2$.

- $\mathcal{D}_{\mathrm{KL}}$ with a KDE score estimator: $L_k^{\mathrm{KL}} \lesssim \sup_x \|J\nabla \ln \hat{\rho}_{t_k}(x)\| + \sup_x \|J\nabla \ln \hat{\mu}_{t_k}(x)\|$, controlled by the KDE bandwidth and diverging in low-density regions. The Gaussian-score variant (Sec. 4.2) instead yields $L_k^{\mathrm{KL,Gauss}} \leq \|\Sigma_{\rho_{t_k}}^{-1}\|_2 + \|\Sigma_{\mu_{t_k}}^{-1}\|_2$, finite away from degenerate covariances.

The $\mathcal{D}_{\mathrm{MMD}^2}$ bound follows from direct differentiation of the corresponding expression in Table 1. For $\mathcal{D}_{\mathrm{KL}}$, $J\nabla g_k(x, t) = J\nabla \ln \hat{\rho}_{t_k}(x) - J\nabla \ln \hat{\mu}_{t_k}(x)$ and the triangle inequality yields the KDE bound; for the Gaussian estimator, $\nabla \ln \mathcal{N}(x; m, \Sigma) = -(x - m)^\top \Sigma^{-1}$ has constant Jacobian $-\Sigma^{-1}$, so $\|J\nabla g_k(x, t)\|_2 \leq \|\Sigma_{\rho_{t_k}}^{-1}\|_2 + \|\Sigma_{\mu_{t_k}}^{-1}\|_2$.

Intuitively, narrower kernels and stronger potentials (larger $w_k$, hence larger $|A_{ik}|$) yield larger $L_F$ and slower or non-contractive iterations; KL with KDE scores is particularly sensitive due to the divergence of $L_k^{\mathrm{KL}}$ in sparse regions. This matches our empirical findings and motivates both damped Anderson acceleration (Anderson, 1965) of the iterations and the homotopy curriculum described next, though, as we discuss in Sec. 7, we recommend the $\mathcal{W}_2^2$ potential for efficient, straightforward training, avoiding these complexities altogether.

**Homotopy continuation**   To help stabilize the FP iterations, we introduce a homotopy curriculum $\alpha(i)$ that gradually increases the correction strength $\alpha$ from 0 to 1 over training iterations. At $\alpha = 0$, $F_0(X_T)_i = X_{t_i}^{\mathrm{base}}$ has $L_F = 0$ and the trivially attractive fixed point $X_T = X_T^{\mathrm{base}}$. By continuity of $JF_\alpha$ in $\alpha$ and the implicit function theorem, there exists a locally unique, smooth solution branch $\alpha \mapsto X_T^{\mathrm{FP}}(\alpha)$ that remains attractive on any sub-interval where $L_F < 1$. A sufficiently slow continuation in $\alpha$ from 0 to 1 thus allows the training dynamics to track this branch, even when $F_1$ alone may not be globally contractive. The schedule $\alpha(i)$ can be fixed as a hyperparameter, or dynamically adjusted during training — for example, based on the residual $\|F_\alpha(X_T) - X_T\|$ — to ensure a sufficiently slow continuation. We find a fixed sigmoid schedule (Fig. 5) sufficient and practical, and ablate this choice in Sec. 6.2. Surprisingly, even for the $\mathcal{W}_2^2$ distance, for which we have a closed-form solution, we find the homotopy curriculum improves the model.

## E.3. Direct solution of the fixed-point problem for the $\mathcal{W}_2^2$ distance

As described briefly in Sec. 4.2, for the $\mathcal{W}_2^2$ distance, the fixed-point problem reduces to a linear system of equations that can be solved directly by matrix inversion. We see this by writing out the fixed-point equations for the $\mathcal{W}_2^2$ distance, with $\nabla g_k(X_{t_k}, t_k, \mathcal{B}) = X_{t_k} - \psi(X_{t_k})$, where $\psi$ is the static OT transport map between $\rho_{t_k}$ and $\mu_{t_k}$:

$$X_{t_i}(x_0, x_1, \mathcal{B}) = X_{t_i}^{\text{base}}(x_0, x_1) + \sum_{k=1}^{K} (X_{t_k} - \psi(X_{t_k})) \underbrace{\left[ w_k \left( \mathcal{I}^{(2)}[\lambda_k](t) - \mathcal{I}^{(2)}[\lambda_k](1)t \right) \right]}_{\equiv A_{i,k}}, \quad \forall i \in \{1, ..., K\}. \quad (117)$$

Rewriting this in matrix form, with $A \in \mathbb{R}^{K \times K}$, $X_T \equiv [X_{t_1} ... X_{t_K}]^T \in \mathbb{R}^{K \times D}$ and $\Psi_T(X_T) \equiv [\psi(X_{t_1}) ... \psi(X_{t_K})]^T \in \mathbb{R}^{K \times D}$:

$$
\begin{align}
X_T &= X_T^{\text{base}} + A(X_T - \Psi_T(X_T)) \notag \\
\Rightarrow \quad (\mathbb{1} - A)X_T &= X_T^{\text{base}} - A\Psi_T(X_T) \quad\quad (118) \notag \\
\Rightarrow \quad X_T &= (\mathbb{1} - A)^{-1} \left( X_T^{\text{base}} - A\Psi_T(X_T) \right) \notag
\end{align}
$$

Thus, the fixed-point problem reduces to inverting and multiplying by a $K \times K$ matrix, where the inversion can be precomputed. Though in general each application of this equation may change the map $\psi$, thereby still leaving us with a complex fixed-point problem that needs to be solved iteratively, when conditioned on single samples from each marginal (as we do in OTP-FM), the map is fixed and a single application of Eq. 118 yields the fixed-point.

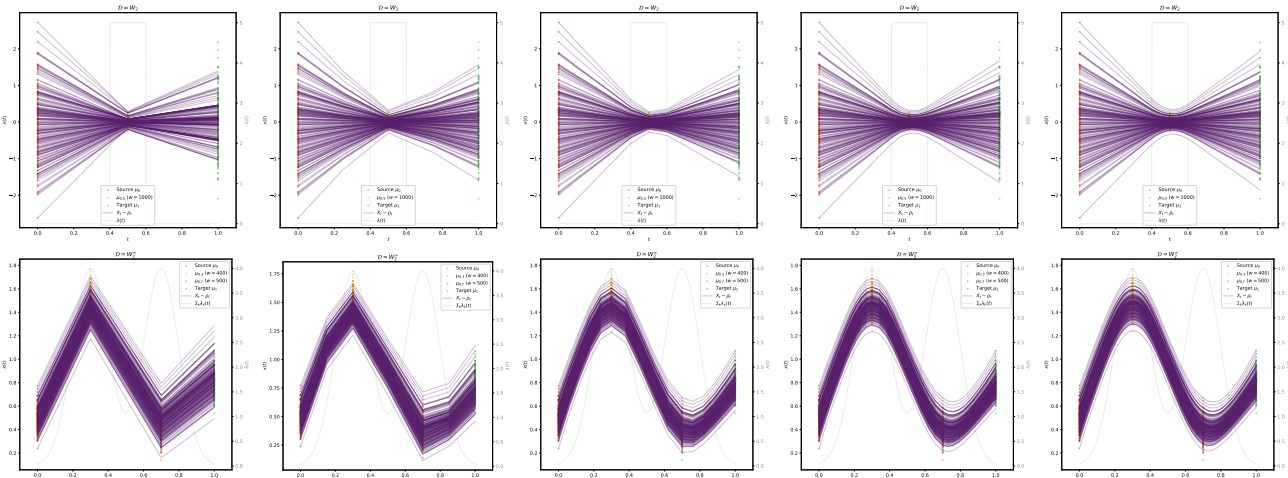

*Figure 6.* Example OTP-FM trajectories inferred from 1, 2, 5, 10, and 50 steps per marginal (left to right), demonstrating the effectiveness of the MeanFlow consistency model.

# F. Experimental details and studies on synthetic data

### F.1. Architecture

We use a three-layer MLP for the MeanFlow velocity model with 256 nodes per hidden layer and the SiLU activation function after each hidden layer. As stated in Sec. 4.3, we embed the times $t_1$ and $\Delta t = t_2 - t_1$ using positional embeddings with 64D each, and the position input with a single embedding layer of 64D as well.

### F.2. Training

We use the Adam optimizer with a learning rate of $3 \cdot 10^{-3}$ and a batch size of 512 samples per marginal. We use a sigmoid curriculum for the correction strength $\alpha(i)$ with a mean at half the total number of training iterations $N$ and a slope of $12/N$:

$$\alpha(i) = \frac{1}{1 + \exp\left[-\frac{12}{N}\left(i - \frac{N}{2}\right)\right]}, \tag{119}$$

as shown in Fig. 5. The training epochs vary between 10 and 50 for convergence depending on the potential strengths and distance metrics: generally, stronger potentials and distances with harder FP problems (MMD and KLD) require more epochs and/or a slower curriculum.

### F.3. MeanFlow, LSD and few-step inference

We train with a 0.75 probability of sampling $t_2 = t_1$ (i.e., on the $u_{t_1,t_2}$ diagonal) and 0.25 probability of sampling $t_2 > t_1$ according to the joint distribution $p_{t_1,t_2}$ (Eq. 114). We find one or two steps per marginal sufficient for reasonably accurate inference of the intermediate $X_{t_k}$ for the MMD and KLD distances, as illustrated in Fig. 6. We maintain an EMA of the model weights $u_\theta^{\text{EMA}}$ for evaluating the $X_{t_k}$ during training as well as for inference, with a decay factor of 0.99. We also compare MeanFlow with the LSD training objective (App. C.1) with the same training settings, and find largely similar performance, leaving a more rigorous comparison to future work. We use 50 steps per marginal for inference in all experiments.

### F.4. Weaknesses of kernel-based distances

As previously discussed in App. D, MMD with the polynomial kernel is not effective even for the exact solution for 1D Gaussian marginals (Fig. 4). The RBF kernel, on the other hand, is performant for the exact solutions, but is limited with respect to the strength of the potential it is able to enforce for OTP-FM. This is illustrated in Fig. 8, where we see that for high strength potentials, the dynamic OT MMD solution effectively enforces $\rho_{t_k}$ to match the intermediate marginal, whereas the OTP-FM MMD solution does not. The KLD solution using a KDE for the score estimate faces similar issues.

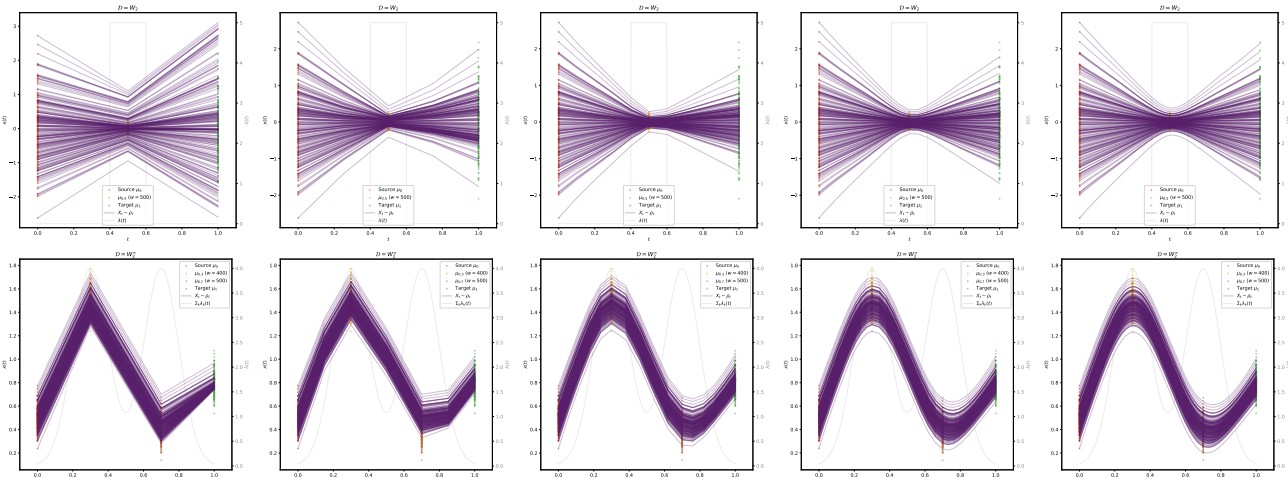

*Figure 7.* Example OTP-FM trajectories trained with the LSD consistency objective and inferred from 1, 2, 5, 10, and 50 steps per marginal (left to right), demonstrating comparable performance to the MeanFlow model.

On the other hand, the $\mathcal{W}_2^2$, $\mathcal{W}_2^\infty$, and KLD with a Gaussian score estimator are able to learn this. This suggests that kernel-based estimates of the MMD and KLD $\nabla g$ provide too noisy and unstable a training target for the OTP-FM velocity model.

### F.5. Comparing $\lambda_k(t)$ shapes and widths

We experiment with three forms of the time-dependence of the potentials $\lambda_k(t)$, with half-width $\tau$:

$$\text{Triangle:} \quad \lambda_k(t)(\tau, t) = \frac{1}{\tau} \max\left(0, 1 - \frac{|t - t_k|}{\tau}\right), \tag{120}$$

$$\text{Rectangle:} \quad \lambda_k(t)(\tau, t) = \frac{1}{2\tau} \theta\left(\tau - |t - t_k|\right), \tag{121}$$

$$\text{Gaussian:} \quad \lambda_k(t)(\tau, t) = \frac{1}{\tau\sqrt{2\pi}} \exp\left(-\frac{(t - t_k)^2}{2\tau^2}\right) \tag{122}$$

where $\theta(x)$ is the Heaviside step function. Different configurations of shapes and widths are shown in the main text (Fig. 2) and compared directly for the same marginals and potential strengths in Fig. 9. The width of the potential has an intuitive effect on the smoothness of the trajectories, with narrower potentials having sharper kinks (approximating a delta function in the limiting case of $\tau \to 0$). We do not observe a significant empirical effect from the shape of the $\lambda_k(t)$.

### F.6. Comparing loss functions

Finally, as described in Sec. 4.3, we compare three different variations of the regression loss function: 1) simple squared L2 loss; 2) adaptive weights as described in the main text with $p = 1$ (Sec. 4.3); and 3) a learnt weighting scheme as proposed in Karras et al. (2024):

$$\mathcal{L} = \mathbb{E}\left[e^{-w(t)} \left\|v^\theta(t_1, t_2, x) - \text{sg}(v_{\text{tgt}}(t_1, t_2, x))\right\|^2 + w(t)\right], \tag{123}$$

where $w(t)$ is a learnt log-variance predicted by the model. These are motivated by the observation that the flow matching loss is in fact a sum over multiple losses at each time step, each of which may be easier or harder to optimize. While the naive squared L2 loss simply weights each time step equally (for uniform time sampling), the two latter schemes attempt to weight the time steps based on the difficulty, either by simply dividing the squared L2 loss by the magnitude of the loss, or by encouraging the network to learn (and minimize) the log-variance of its own predictions for each time step. We show an example comparison in Fig. 10 and find the adaptive weights with $p = 1$ to be both most performant and most stable to train in this experiment.

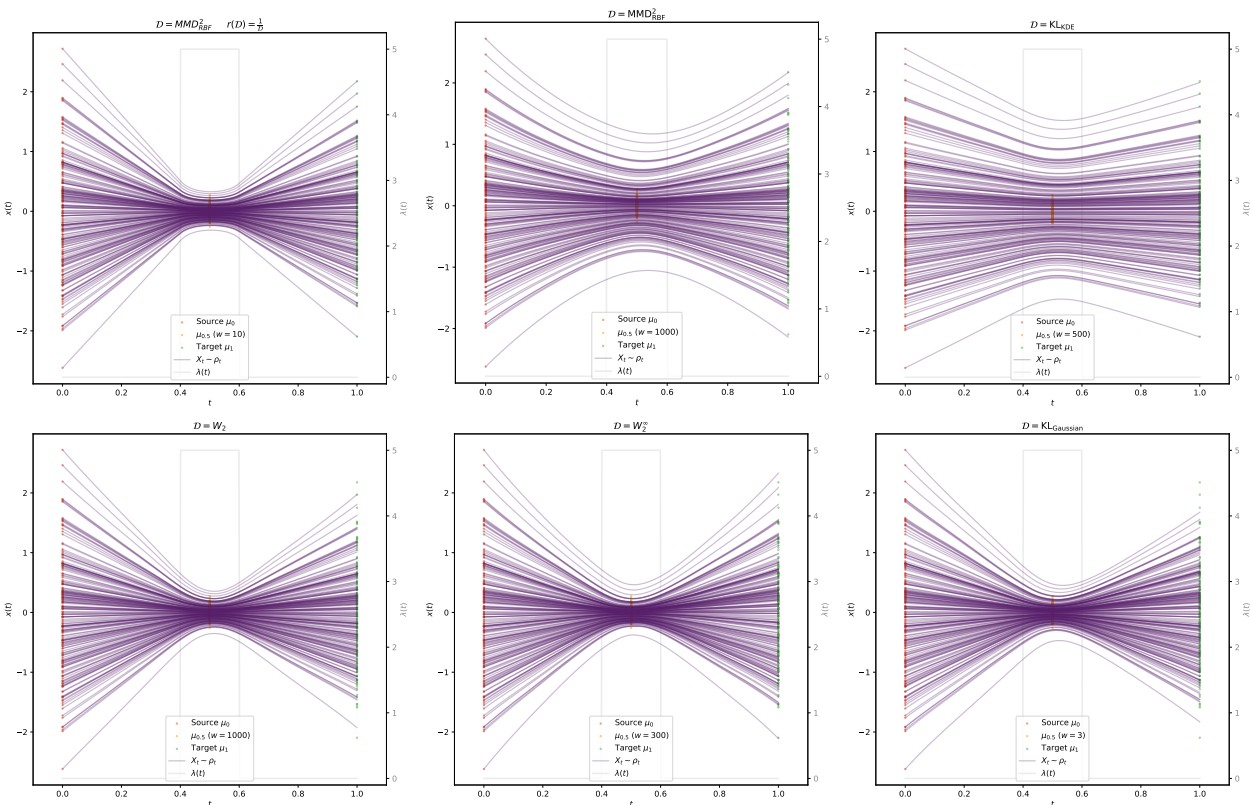

*Figure 8.* The exact dynamic OT solution with MMD and the RBF kernel (top left) demonstrates that the intermediate marginal is effectively enforced by the potential. However, kernel-based estimates of the MMD (top center) and KLD (top right) $\nabla g$ do not allow OTP-FM to learn the intermediate marginal well (even for very high strength potentials). Finally, the $\mathcal{W}_2^2$ (bottom left), $\mathcal{W}_2^\infty$ (bottom center), and KLD with a Gaussian score estimator (bottom right) do learn the intermediate marginal.

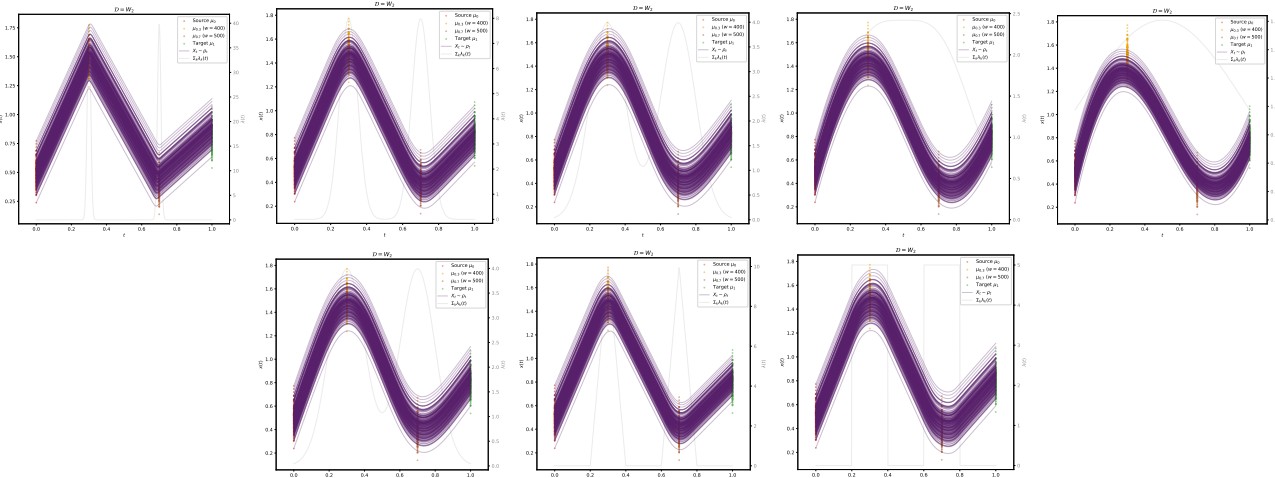

*Figure 9.* *Top:* Varying the half-width $\tau$ of the $\lambda_k(t)$s, with Gaussian shape, from 0.01, 0.05, 0.1, 0.2, to 0.4 (left to right). *Bottom:* Varying the shape of the $\lambda_k(t)$s, with $\tau = 0.1$, between Gaussian, triangular, and rectangular (left to right).

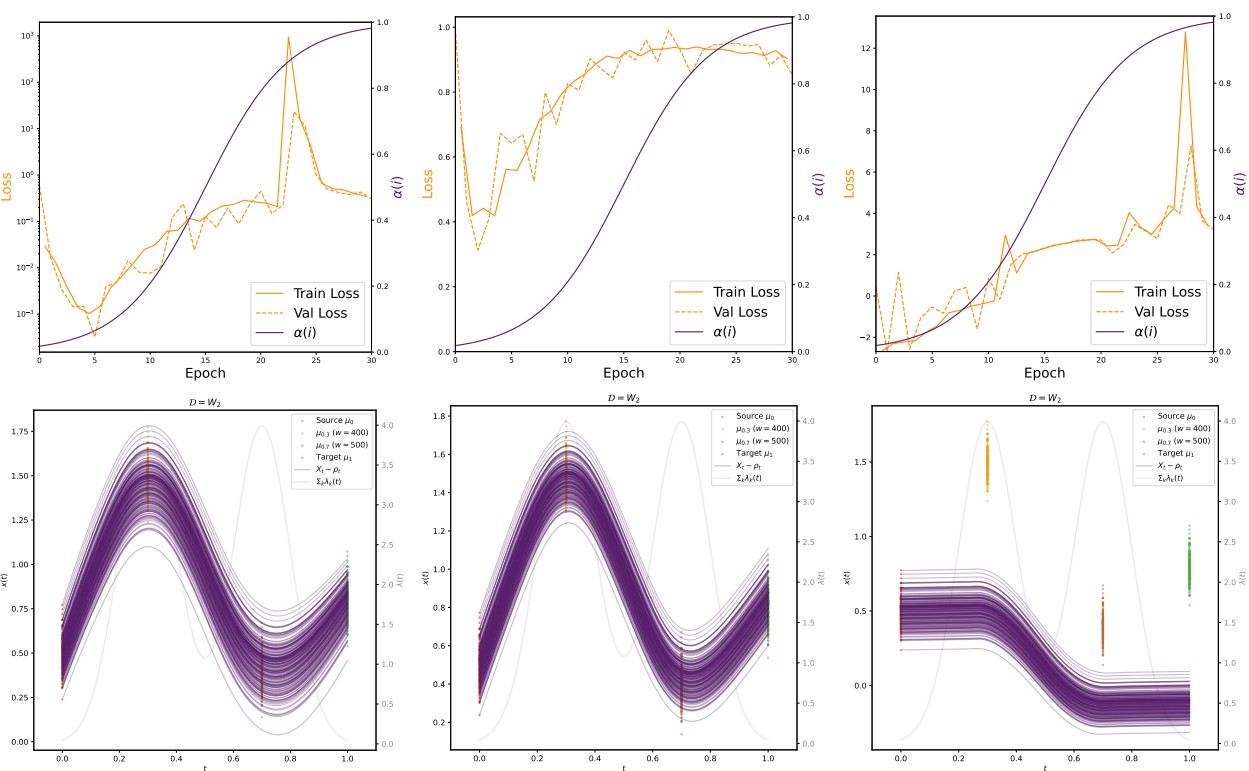

*Figure 10. Top:* Example loss curves for the three different loss functions: squared L2 (top left, log scale), adaptive weights with $p = 1$ (top center), and learnt log-variance (top right). *Bottom:* The corresponding OTP-FM trajectories for each loss function (for the same marginals and potential strengths).

# G. Experimental details and ablation studies on single-cell RNA sequencing datasets

### G.1. Embryoid body scRNA-seq dataset

We use the preprocessed embryoid body (EB) data provided by Moon et al. (2019), which includes the 100-dimensional PCA representation per cell. The dataset comprises a total of 16,819 cells collected over 27 days across the five discrete time intervals $t_0 \in [0, 3]$, $t_1 \in [6, 9]$, $t_2 \in [12, 15]$, $t_3 \in [18, 21]$, and $t_4 \in [24, 27]$ days, with 2,381, 4,163, 3,278, 3,665, and 3,332 cells per interval, respectively. We consider four experimental protocols on this dataset:

- **EB 5D L1O**: training in the 5-dimensional PCA space with one of the intermediate marginals $t_1$, $t_2$, or $t_3$ held out per fold, averaging $\mathcal{W}_1$ in the normalized space over the three held-out folds, following Neklyudov et al. (2024);

- **EB 100D L0O**: training in the 100-dimensional PCA space on all five marginals (no held-out marginal), averaging MMD over the four training times after $t_0$, following Persiianov et al. (2026);

- **EB 100D L1O**: same as L1O but in the 100-dimensional PCA space, averaging MMD only at the held-out times in the unnormalized PCA space, following Persiianov et al. (2026);

- **EB 100D L2O**: training in the 100-dimensional PCA space with both $t_1$ and $t_3$ held out (training on $\{t_0, t_2, t_4\}$), averaging MMD over the four non-source times $t_1$–$t_4$, following Theodoropoulos et al. (2026).

Each PC feature is normalized to zero mean and unit variance across the full dataset for training, and evaluation metrics are computed in the original (un-normalized) PCA space, except for EB 5D L1O which evaluates in normalized space following the convention of Neklyudov et al. (2024). For the EB protocols we evaluate on a random subsample of 2,000 cells per time interval.

### G.2. CITE-seq scRNA-seq dataset

The CITE-seq dataset (Burkhardt et al., 2022) consists of single-cell RNA sequencing measurements of CD34+ hematopoietic stem and progenitor cells (HSPCs) from four human donors, collected on four days, $t_0$—$t_3$, over a 10-day period. We use the pre-computed 50-PC representation provided by Wang et al. (2026), consisting of 31,240 cells distributed across the four time intervals as 7,476, 6,999, 9,511, and 7,254 cells per day, respectively. For training, the PC features are centered and divided by the maximum per-feature standard deviation, following Neklyudov et al. (2024); this normalization preserves the relative scale across PCs. We consider two experimental protocols on this dataset, each with two leave-one-out folds (holding out day 3 and day 4 separately, with the source day 2 and target day 7 always present):

- **CITE 5D L1O**: using the first 5 PCs and averaging $\mathcal{W}_1$ in the normalized PCA space over the two held-out folds, following Kapusniak et al. (2024);

- **CITE 50D L1O**: using all 50 PCs and averaging $\mathcal{W}_1$ in the original (un-normalized) PCA space over the two held-out folds, following Neklyudov et al. (2024).

For both protocols we evaluate on the full set of cells at each time point for an apples-to-apples comparison with the baselines (Neklyudov et al., 2024; Kapusniak et al., 2024, and others).

### G.3. Architecture and training

For all single-cell experiments we use an MLP velocity network with residual connections every two layers, SiLU activations, LayerNorm, and dropout 0.2, trained with the Adam optimizer at a base learning rate of $3 \cdot 10^{-3}$ and a batch size of 256 samples per marginal. All training runs are performed on a single NVIDIA L40S GPU and final model checkpoints are chosen based on the minimum MMD or $\mathcal{W}_1$ scores. We use 64-dimensional positional embeddings for both $t_1$ and $\Delta t = t_2 - t_1$, and a 64-dimensional linear embedding for the position input, following the architecture described in App. F. We use eight to ten hidden layers of 768 nodes each (4.9M—6.5M parameters), training each of the $\mathcal{W}_2^2$, $\mathcal{W}_2^\infty$, MMD, and KLD potentials with the iMF consistency loss (App. C.1), adaptive loss weighting with $p = 1$, tuning the potential strengths and $\lambda_k(t)$'s for each dataset (exact configurations are provided in our linked codebase). We use a sigmoid curriculum for $\alpha(i)$ with slope 24 and midpoint at iteration 0 (i.e., the schedule starts at $\alpha = 0.5$ and quickly saturates to 1). Intermediate potential times $\{t_k\}$ are auto-computed as evenly spaced fractions of the training interval. We train for between 200 and 300

*Table 5.* Ablation of the consistency model choice and loss function on EB 100D L2O ($\overline{\text{MMD}}$ on held-out times, lower is better). Baseline in **bold**; others single-seed.

| Consistency | $\overline{\text{MMD}}\downarrow$ | | Loss function | $\overline{\text{MMD}}\downarrow$ |
|---|---|---|---|---|
| **IMF** | **0.0675** | | **Adaptive** | **0.0675** |
| MeanFlow | 0.0690 | | MSE | 0.0698 |
| LSD | 0.0696 | | Weighted | 0.0746 |

epochs depending on the distance metric, with training times shown in Figs. 3 and 12. We use MMD with the RBF kernel of bandwidth 10 and KLD with KDE score estimates of bandwidth 3, both with a maximum of 5 Picard iterations per training iteration. We tested the KLD with a Gaussian score estimator as well but this did not converge.

### G.4. Ablation studies

Table 5 continues the ablation study from Sec. 6.2, varying the consistency model and loss function on the EB 100D L2O experiment. The baseline model uses the same parameters specified in Sec. 6.2, trained with the iMF formulation and adaptive loss function. Figure 11 shows the ground truth and simulated trajectories on this experiment for the baseline model and different potential parameters.

### G.5. Additional results

Figure 12 shows additional timing vs. performance plots for the EB 5D, EB 100D L1O, and CITE 50D experiments, and Fig. 13 shows trajectory visualizations for the EB 100D L2O experiment for OTP-FM with different potentials and all baseline models for which we have access to trajectories, either by our own training (as specified in App. G.6) or provided by the authors.[10]

### G.6. Benchmarking methods

Below we describe each baseline method, the codebase and configuration used in our evaluation and timing studies, and the provenance of the results in Table 3, in the order that they appear. All timing measurements (Figs. 3 and 12) were performed on a single cloud VM with one NVIDIA L40S GPU and one AMD EPYC 7R13 processor (8 cores / 16 threads, 32 MiB L3 cache) and 124 GiB RAM. Timings report the wall-clock training time per fold of one held-out marginal configuration (Sec. 6.3), measured on an otherwise idle machine with no concurrent jobs. We use the author-provided configurations whenever available for the respective dataset, otherwise adapting the most similar configuration available in their repository. For methods whose per-fold training time exceeds ten minutes, we measure the time per training iteration (or per epoch) from a ten-minute run and extrapolate by the total iteration count specified in the original configuration; for all other methods the reported value is of the full training run. We exclude evaluation time and precomputation of OT couplings from the timing measurements; for the datasets considered in this work, we measured the OT precomputation time to be between 5–15s and, hence, negligible in comparison to the training time.

**TrajectoryNet** TrajectoryNet (Tong et al., 2020) learns a continuous normalizing flow with dynamic OT regularization for trajectory inference. We use the provided code[11] with the authors' EB 5D configuration; the EB 5D L1O value is taken directly from Tong et al. (2020). We were not able to run the EB 100D and CITE experiments ourselves because of the prohibitively slow training time. For the same reason, timing values for EB 100D, CITE 5D, and CITE 50D are extrapolated from limited-iteration runs using configurations adapted from the EB 5D config.

**NLSB** Neural Lagrangian Schrödinger Bridge (NLSB) (Koshizuka & Sato, 2023) models population dynamics via a Lagrangian-regularized neural SDE. We use the provided code[12] with the authors' EB 5D configuration; the EB 5D L1O value is taken from Koshizuka & Sato (2023), the EB 100D L0O and L1O values from Chen et al. (2023), and we ran the CITE 5D L1O setting ourselves with a configuration adapted from EB 5D. We were not able to complete the EB 100D L2O

---

[10]We thank Panagiotis Theodoropoulos for providing the trajectories for 3MSBM.

[11]https://github.com/KrishnaswamyLab/TrajectoryNet

[12]https://github.com/take-koshizuka/NLSB

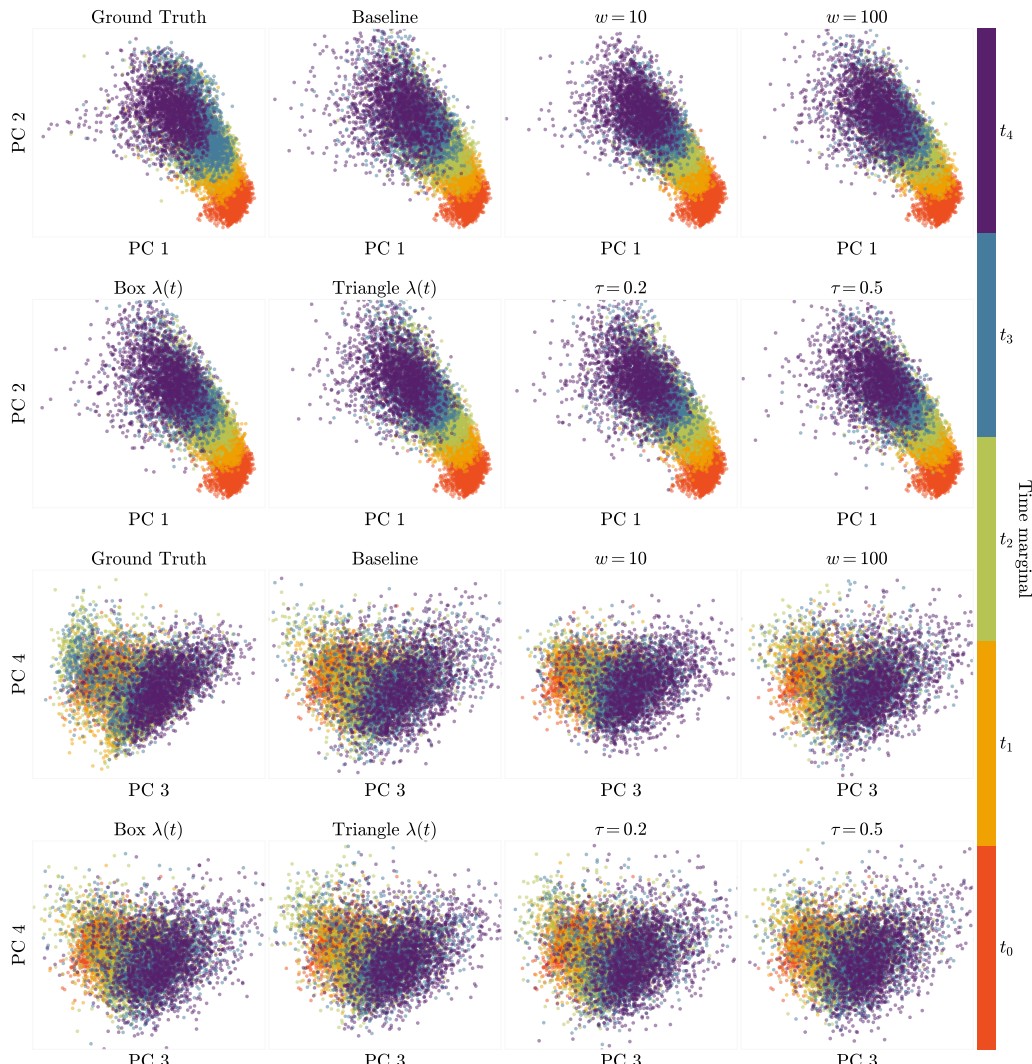

*Figure 11.* Ground truth and simulated trajectories for the EB 100D dataset in the leave-two-out setting ($t_1$ and $t_3$ marginals held out) for different OTP-FM potential parameters. The "baseline" model uses $\mathcal{D} = \mathcal{W}_2^2$ with a Gaussian $\lambda(t)$, with half-width $\tau = 0.33$ and strength $w = 1000$. All other models use the same parameters except for the difference specified.

and CITE 50D runs because of the prohibitively slow training time. Their timings are extrapolated from ten-minute runs using configurations adapted from the EB 5D config.

**MIOFlow**  MIOFlow (Huguet et al., 2022) combines a Geodesic Autoencoder (GAGA) with a neural ODE for trajectory inference; crucially it trains the neural ODE by matching predicted and observed marginals under the $\mathcal{W}_2^2$ distance instead of the maximum-likelihood loss used by continuous normalizing flows (CNFs). This sidesteps the instantaneous change-of-variables computation that CNFs require at every integration step, leading to much faster training than conventional simulation-based approaches. We use the provided code[13]. The repository does not provide scRNA-specific configurations matching our benchmark, so we adapt the 5D demo configuration for all four datasets. The CITE 50D L1O values are taken from Huguet et al. (2022), the EB 100D L1O and L0O values from Chen et al. (2023), and we ran the EB 5D L1O, EB 100D L2O, and CITE 5D L1O settings ourselves, bypassing the GAGA encoding and decoding.

---

[13]https://github.com/KrishnaswamyLab/MIOFlow

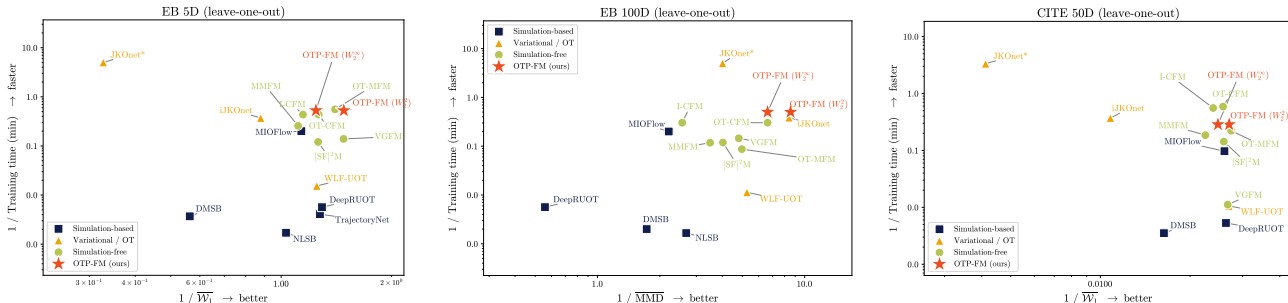

*Figure 12.* Training time vs. performance of different methods for the EB 5D L1O (left), EB 100D L1O (center), and CITE 50D L1O (right) experiments. Top right is better.

**DMSB** DMSB (Chen et al., 2023) is a neural SDE method that lifts multi-marginal Schrödinger bridges to phase space to smooth the dynamics. We use the provided code[14] with the authors' EB 100D and CITE 50D configurations; the EB 5D L1O value is reported from Chen et al. (2023), the CITE 5D L1O value from Kapusniak et al. (2024), and we ran the EB 100D and CITE 50D L1O settings ourselves. We note that Chen et al. (2023) provides EB 100D L1O + L0O in the z-score *normalized* space while we report unnormalized MMD values; hence, we retrain DMSB to obtain unnormalized values for this setting as well. The EB 5D and CITE 5D timings use configurations adapted from the EB 100D and CITE 50D configs, respectively.

**DeepRUOT** DeepRUOT (Zhang et al., 2025) is a neural SDE method that jointly learns velocity, growth, and score networks under a regularized unbalanced OT objective. We use the provided code[15] and adapt the original four-phase EB 5D training schedule for all datasets. The EB 5D L1O, CITE 5D L1O, and CITE 50D L1O values are taken from Zhang et al. (2025), and we ran the EB 100D settings ourselves. Timing values are obtained by extrapolating the per-phase iteration timing measured within 10 minutes per dataset.

**WLF-UOT** WLF-UOT (Neklyudov et al., 2024) represents an alternative paradigm for trajectory inference, solving the variational problem directly. We use the provided code[16] with the authors' EB 5D and CITE 50D unbalanced-OT configurations; the EB 5D L1O value is taken from Neklyudov et al. (2024), the CITE 5D and 50D L1O values from Kapusniak et al. (2024), and we ran the EB 100D settings ourselves with configurations adapted from those provided. For consistency with the other baselines, for evaluation we generate all samples starting from $t = 0$. We note that Neklyudov et al. (2024) also report values incorporating potentials based on the *held-out* marginals, which we do not include for fairness.

**JKOnet\*** JKOnet (Bunne et al., 2022) interprets diffusion as energy-minimizing trajectories in Wasserstein space, following the work of Jordan, Kinderlehrer, and Otto (JKO) and solves this minimization problem. We experiment with an improved version, JKOnet\* (Terpin et al., 2024). We use the provided code[17] with the authors' EB 5D configuration; the other datasets are run with configurations adapted from the EB 5D config. For EB 5D L1O, EB 100D L2O, CITE 5D L1O, and CITE 50D L1O we ran the method ourselves, while the EB 100D L0O and L1O values are from Persiianov et al. (2026).

**iJKOnet** iJKOnet (Persiianov et al., 2026) proposes an alternative, adversarial scheme for the JKO optimization problem above. We use the provided code[18] with the authors' EB 5D and CITE 50D configurations. For EB 5D L1O, EB 100D L2O, CITE 5D L1O, and CITE 50D L1O we re-ran the method, while the EB 100D L0O and L1O values are from Persiianov et al. (2026). EB 100D and CITE 5D timings use configurations adapted from EB 5D and CITE 50D, respectively.

**3MSBM** 3MSBM (Theodoropoulos et al., 2026) extends multi-marginal Schrödinger bridges to phase space via a score-matching-style objective to avoid simulation-based training, with the score targets calculated by solving a dynamic

---

[14] https://github.com/TianrongChen/DMSB
[15] https://github.com/zhenyiizhang/DeepRUOT
[16] https://github.com/necludov/wl-mechanics
[17] https://github.com/antonioterpin/jkonet-star
[18] https://github.com/MuXauJl11110/iJKOnet

programming problem during training. We completed one training run of 3MSBM on the EB 100D L2O setting following the code provided[19] and the authors' EB-specific configuration but it did not converge. We therefore only report results directly from Theodoropoulos et al. (2026) (EB 100D L2O) and include trajectories that are provided by the authors in Fig. 13.

**I-CFM, OT-CFM, and [SF]²M**   I-CFM and OT-CFM (Tong et al., 2024) train flow-matching velocity models on independently sampled or OT-aligned pairs of consecutive marginals, respectively, while [SF]²M (Tong et al., 2023) uses score-matching to learn a Schrödinger bridge between marginals. In all cases, trajectories are stitched together piecewise between consecutive training marginals, as described in App. A.2. We use the provided `torchcfm` code,[20] with hyperparameters adapted from the single-cell tutorials. For OT-CFM we precompute a single full-dataset EMD coupling per consecutive time pair rather than re-solving per minibatch, for a fair comparison with OTP-FM. The EB 5D L1O value is taken from Tong et al. (2024; 2023), the CITE 5D and 50D L1O values from Kapusniak et al. (2024), and the EB 100D experiments we ran ourselves.

**OT-MFM**   OT-MFM (Kapusniak et al., 2024) encourages CFM trajectories to follow the data manifold by replacing straight interpolants with approximate geodesics of a data-dependent Riemannian metric. We use the provided code[21] with the authors' EB 5D, CITE 5D, and CITE 50D LAND-metric configurations; the EB 5D L1O and CITE 5D and 50D L1O values are taken from Kapusniak et al. (2024), and we ran the EB 100D experiments ourselves using a configuration adapted from the EB 5D config.

**VGFM**   VGFM (Wang et al., 2026) jointly learns velocity and *growth* fields using a loss derived from semi-relaxed OT on top of the CFM loss. We use the provided code[22] with the authors' EB 5D, CITE 5D, and CITE 50D notebook configurations (both the "warm-up" and training phases); the EB 5D L1O and CITE 5D and 50D L1O values are taken from Wang et al. (2026), and we ran the EB 100D experiments ourselves with a configuration adapted from the EB 5D notebook.

**MMFM**   MMFM (Rohbeck et al., 2025) trains CFM between multiple marginals using a cubic-spline interpolation as the conditional velocity regression target. We train MMFM following the code provided,[23] using an identical model architecture to OTP-FM's for each respective configuration, over 5 training seeds on all five evaluation settings.

---

[19] https://github.com/panostheo98/3MSBM
[20] https://github.com/atong01/conditional-flow-matching
[21] https://github.com/kksniak/metric-flow-matching
[22] https://github.com/DongyiWang-66/VGFM
[23] https://github.com/Genentech/MMFM

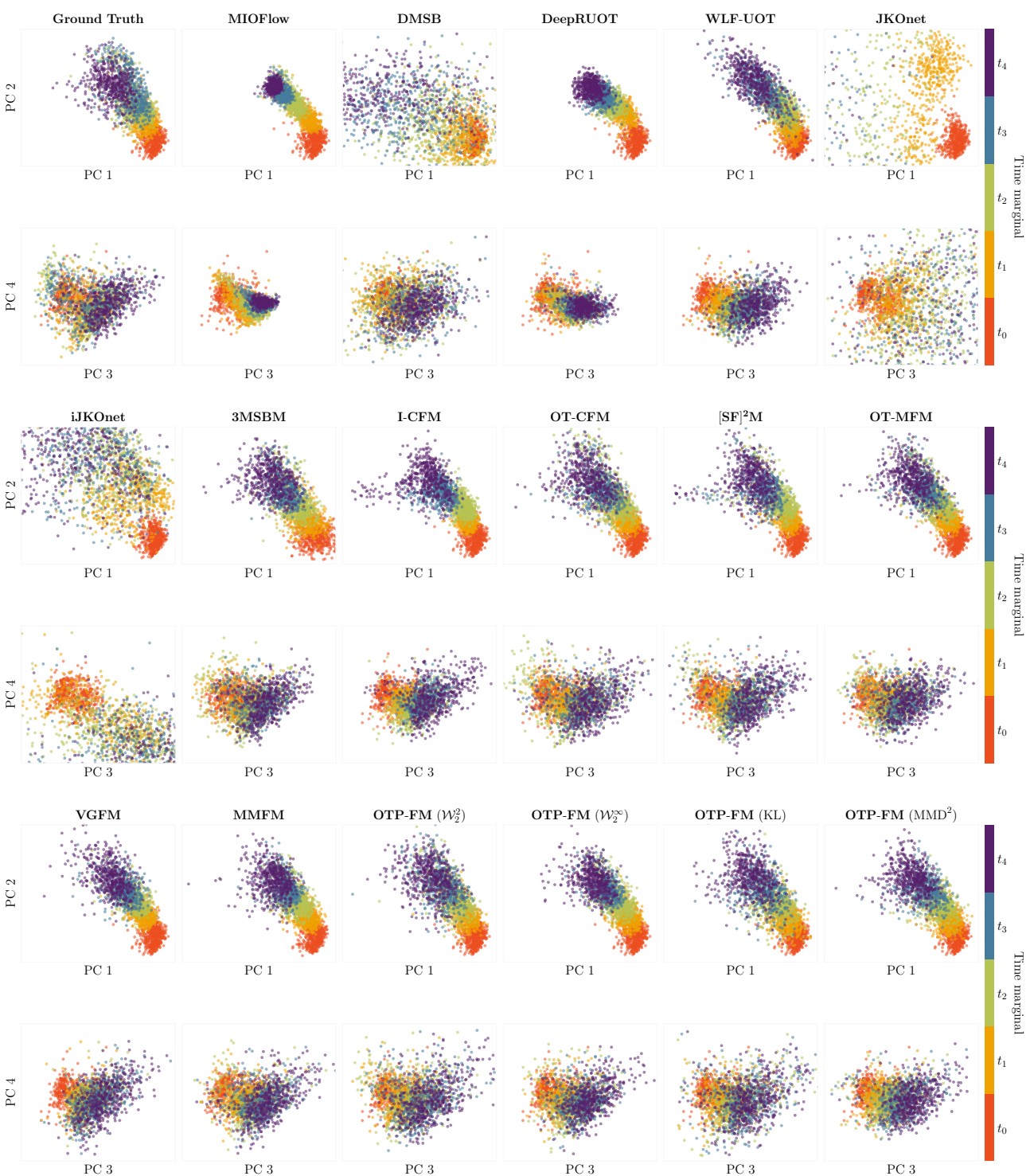

*Figure 13.* Ground truth and simulated trajectories for the EB 100D dataset in the leave-two-out setting ($t_1$ and $t_3$ marginals held out) in all four PCs across all five time intervals, comparing OTP-FM with different potentials with baseline models.

*Table 6.* Per-timepoint results on the GoM dataset, showing the mean and standard deviation across five seeds of the $\mathcal{W}_2$ distance to the four held-out marginals and the average distance to the rest (lower is better). 3MSBM results are reported from Theodoropoulos et al. (2026), while MMFM results are from our training runs. The best performing method per timepoint is in **bold** and the second-best in *italics*.

| Method | $t_1\ \mathcal{W}_2$ | $t_3\ \mathcal{W}_2$ | $t_5\ \mathcal{W}_2$ | $t_7\ \mathcal{W}_2$ | Rest $\mathcal{W}_2$ |
|---|---|---|---|---|---|
| 3MSBM | $0.20 \pm 0.03$ | $0.18 \pm 0.03$ | $\mathbf{0.07 \pm 0.02}$ | $\mathit{0.09 \pm 0.00}$ | $\mathbf{0.05 \pm 0.03}$ |
| MMFM | $0.20 \pm 0.00$ | $0.20 \pm 0.01$ | $\mathit{0.08 \pm 0.01}$ | $0.12 \pm 0.02$ | $0.08 \pm 0.01$ |
| OTP-FM ($\mathcal{W}_2^2$) | $\mathbf{0.07 \pm 0.00}$ | $\mathbf{0.15 \pm 0.01}$ | $0.09 \pm 0.01$ | $\mathbf{0.08 \pm 0.00}$ | $\mathit{0.06 \pm 0.01}$ |
| OTP-FM ($\mathcal{W}_2^\infty$) | $\mathbf{0.07 \pm 0.00}$ | $\mathbf{0.15 \pm 0.00}$ | $0.11 \pm 0.00$ | $0.11 \pm 0.01$ | $0.07 \pm 0.00$ |
| OTP-FM (MMD$^2$) | $0.15 \pm 0.02$ | $0.33 \pm 0.12$ | $0.25 \pm 0.16$ | $0.15 \pm 0.05$ | $0.22 \pm 0.10$ |
| OTP-FM (KL) | $0.13 \pm 0.01$ | $0.26 \pm 0.09$ | $0.15 \pm 0.06$ | $0.10 \pm 0.02$ | $0.16 \pm 0.06$ |

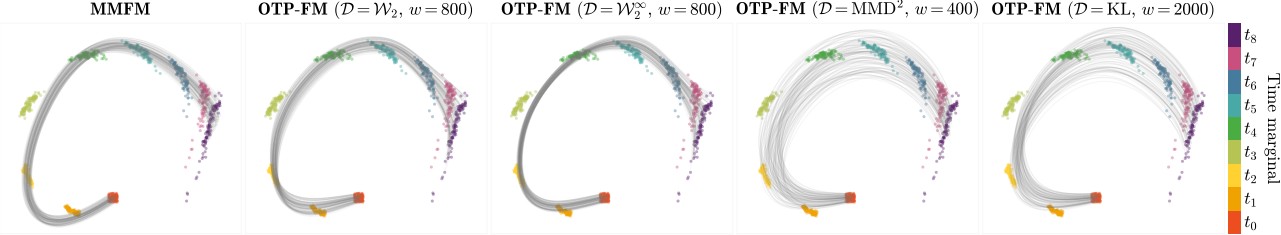

| MMFM | OTP-FM ($\mathcal{D}=\mathcal{W}_2$, $w=800$) | OTP-FM ($\mathcal{D}=\mathcal{W}_2^\infty$, $w=800$) | OTP-FM ($\mathcal{D}=$MMD$^2$, $w=400$) | OTP-FM ($\mathcal{D}=$KL, $w=2000$) |

*Figure 14.* Ground truth and simulated trajectories for the GoM dataset, comparing MMFM and OTP-FM with different potentials.

# H. Experimental details on the Gulf of Mexico dataset

## H.1. Dataset

This dataset comprises high-resolution bathymetric measurements of the Gulf of Mexico (GoM) in the region between $98°$E and $77°$E in longitude and from $18°$N to $32°$N in latitude. The original dataset was released to the public by the United States Department of Defense and contains hourly observations between January 1st, 2001 and August 31st, 2024 (HYCOM Consortium & Center for Ocean-Atmospheric Prediction Studies (COAPS), 2024). Shen et al. (2025) then extracted the velocity field around a vortex from the June 1st, 2024, 5pm timepoint, which they then used to simulate 1,000 particle trajectories. Finally, these trajectories were each randomly sampled at one of nine time points $t_0 — t_8$, such that each particle is only observed at a single time point, yielding 111 samples per marginal. These are the ground truth marginals shown in Fig. 14.

## H.2. Architecture and training

We use a ten-layer MLP with residual connections every two layers and 128 nodes per hidden layer for the velocity model. We use the Adam optimizer with a learning rate of $10^{-3}$, a batch size of 64 samples per marginal, and the same sigmoid curriculum for $\alpha(i)$ as in the synthetic data experiments. In this case, we find the simple MSE loss to be most performant. We train for a total of 2000 epochs for each distance metric, corresponding to a total training time of 4 minutes for the $\mathcal{W}_2^2$ and $\mathcal{W}_2^\infty$ distances and 12 minutes for the MMD and KLD distances on a single NVIDIA L40S GPU. We use MMD with the RBF kernel and KLD with KDEs for the score estimates, both with a bandwidth of 3 and a maximum of 20 Picard iterations per training iteration. The final model checkpoints are chosen based on the average $\mathcal{W}_2^2$ distance over the held-out times.

## H.3. Additional results

Table 6 shows the per-timepoint results on the GoM dataset, including OTP-FM with the KL and MMD$^2$ potentials, and Fig. 14 shows the ground truth and simulated trajectories for the GoM dataset, comparing MMFM and OTP-FM with different potentials.

*Table 7.* Per-timepoint results on the Beijing air quality dataset, showing the mean and standard deviation across five seeds of the $\mathcal{W}_2$ distance to the four held-out marginals and the average distance to the rest (lower is better). 3MSBM results are reported from Theodor-opoulos et al. (2026), who do not report the training scores, while MMFM results are from our training runs. The best performing method per timepoint is in **bold** and the second-best in *italics*.

| Method | $t_2\ \mathcal{W}_2$ | $t_5\ \mathcal{W}_2$ | $t_8\ \mathcal{W}_2$ | $t_{11}\ \mathcal{W}_2$ | Rest $\mathcal{W}_2$ |
|---|---|---|---|---|---|
| 3MSBM | *12.87 ± 1.87* | 79.36 ± 12.65 | 27.89 ± 5.88 | 32.26 ± 4.15 | |
| MMFM | 23.80 ± 3.58 | 17.51 ± 3.61 | 44.34 ± 7.66 | 69.91 ± 12.68 | **11.88 ± 2.81** |
| OTP-FM ($\mathcal{W}_2^2$) | **12.02 ± 1.35** | 12.09 ± 1.07 | 22.55 ± 1.45 | **21.10 ± 1.46** | *18.20 ± 0.54* |
| OTP-FM ($\mathcal{W}_2^\infty$) | 23.70 ± 11.97 | 14.45 ± 3.61 | *21.35 ± 11.70* | 26.59 ± 2.07 | 24.50 ± 4.61 |
| OTP-FM (MMD$^2$) | 45.08 ± 0.46 | *11.76 ± 0.43* | 22.28 ± 1.19 | *24.21 ± 0.66* | 27.62 ± 0.22 |
| OTP-FM (KL) | 42.31 ± 0.58 | **5.21 ± 0.26** | **18.63 ± 1.03** | 26.31 ± 1.57 | 26.15 ± 0.41 |

# I. Experimental details on the Beijing air quality dataset

## I.1. Dataset

This final dataset includes hourly measurements of the concentrations of six main air pollutants and six meteorological variables from 12 monitoring sites in Beijing, between 2013 and 2017 (Chen, 2017). We follow the experimental setup of Theodoropoulos et al. (2026), whereby we focus on one station — Dingling's — measurements of one important pollutant: particulate matter smaller than 2.5 $\mu$m, or PM2.5. We aggregate the hourly measurements by month, and sample every other month in the period of March 2013 — March 2015; i.e., 13 total marginals: $t_0$ — $t_{12}$. We hold out $t_2$, $t_5$, $t_8$, and $t_{11}$ from the training to test the interpolation ability of the model, leaving nine marginals (seven intermediate potentials) for training. Each marginal contains around 740 samples.

## I.2. Architecture and training

We use a four-layer MLP with residual connections every two layers and 128 nodes per hidden layer for the velocity model. We use the Adam optimizer with a learning rate of $3 \cdot 10^{-3}$, a batch size of 128 samples per marginal, and the same sigmoid curriculum for $\alpha(i)$ as in the synthetic data experiments. We find simple MSE to again be most performant. We train for a total of 600 epochs for the $\mathcal{W}_2^2$ and $\mathcal{W}_2^\infty$ distances and 100 epochs for the MMD and KLD distances, corresponding to a total training time of 5 minutes for the former and 2 minutes for the latter distances on a single NVIDIA L40S GPU. We use MMD with the RBF kernel of bandwidth 3 and KLD with the Gaussian score estimates, both with a maximum of 5 Picard iterations per training iteration. The final model checkpoints are chosen based on the average $\mathcal{W}_2^2$ distance over the held-out times.

## I.3. Additional results

Table 7 shows the per-timepoint results on the Beijing air quality dataset, and Fig. 15 shows the ground truth and simulated trajectories for the Beijing air quality dataset, comparing MMFM and OTP-FM. This dataset interestingly highlights a key failure mode of prescribed approaches such as MMFM: as evident from Fig. 15, the cubic spline interpolation assumed in MMFM manifestly does not capture the data. In contrast, the flexibility of the OTP-FM design space allows the data to dictate the optimal interpolation, yielding a better fit to the held-out marginals.

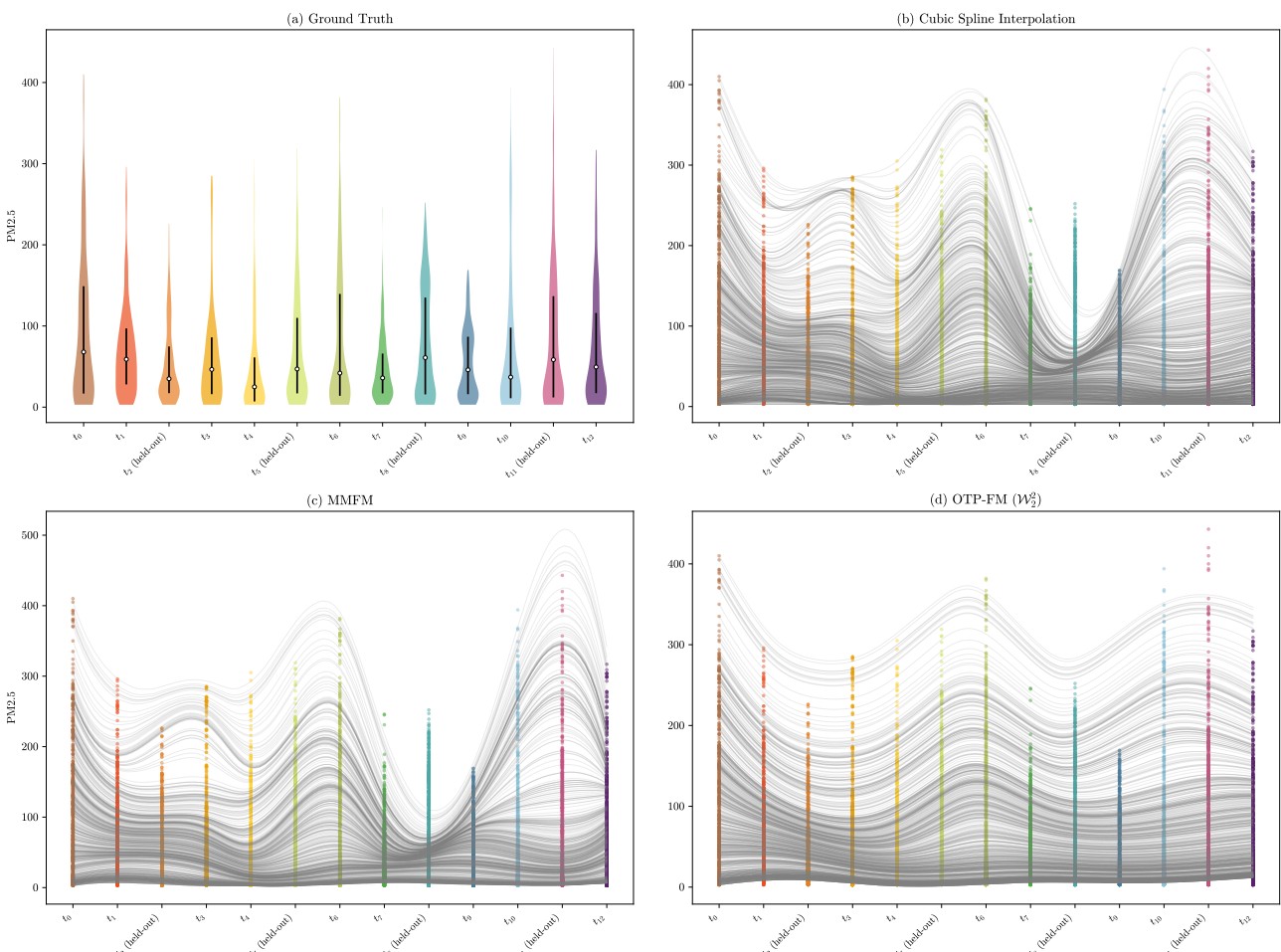

*Figure 15.* Visualizing the Beijing PM2.5 air quality dataset. (a) Ground truth marginal distributions as violin plots showing the density at each timepoint, the median, and the interquartile range. The marginals are heavily concentrated at low PM2.5 but with long tails. (b) Cubic spline interpolation through OT-coupled ground-truth samples, excluding held-out marginals; this represents MMFM's training targets. (c) Trajectories inferred by MMFM. (d) Trajectories inferred by OTP-FM ($\mathcal{W}_2^2$ potential).

