# OpenReview forum: "Multimarginal flow matching with optimal transport potentials"
_ICML.cc/2026/Conference — ICML 2026 regular_

### Official Review · Reviewer_nZvx · 2026-03-05

**Soundness:** 2
**Presentation:** 1
**Significance:** 2
**Originality:** 2
**Overall Recommendation:** 3
**Confidence:** 3

**Summary:**

The paper proposes the OTP-FM method for trajectory inference, where the objective is to recover the underlying density dynamics given only samples observed at discrete time points. The approach builds on the dynamic formulation of optimal transport (OT) [1], viewed through a Lagrangian framework that includes a kinetic energy term.

The authors extend this formulation by introducing a Vlasov-type potential energy term, which depends on the full density rather than on individual particles. This term is defined as a discrepancy between the ground-truth intermediate marginals and the marginals predicted by the model. The resulting variational problem is solved by learning a vector field that governs the density evolution.

To parameterize the vector field, the authors adopt the MeanFlow formulation [2], since one of the drawbacks of FM is that evaluating these positions requires numerically simulating the flow ODE, generally requiring a large number of network eval-uations and time steps for high accuracy. They introduce one modification to the original MeanFlow objective: instead of privileging time $t_2$, they privilege $t_1$, allowing the model to learn forward-in-time dynamics rather than backward ones (see Appendix C.2).

The method is evaluated on a one-dimensional synthetic example and on three real-world datasets.

[1] Benamou, Jean-David, and Yann Brenier. "A computational fluid mechanics solution to the Monge-Kantorovich mass transfer problem." _Numerische Mathematik_ 84.3 (2000): 375-393.

[2] Geng, Zhengyang, et al. "Mean Flows for One-step Generative Modeling." _The Thirty-ninth Annual Conference on Neural Information Processing Systems_.

**Compliance With Llm Reviewing Policy:**

Affirmed.

**Final Justification:**

Score 2 (reject) -> 3 (weak reject). l still think the paper requires a fresh round of revision. See my response to the author's rebuttal.

**Key Questions For Authors:**

- Could the authors include and discuss the related works mentioned above in the citation list? A clear positioning of the current work within the infinite zoo of trajectory inference papers could help to better understand the uniqueness and advantages of the current method.

- Could the authors provide more explanation details for a broader
 audience regarding solving the fixed-point problem in equation (23)?

- Could the authors provide experimental results on the EB dataset in 100 dimensions using the standard leave-one-out evaluation protocol (e.g., see [1] for details), in order to enable a comparison with the aforementioned methods?

[1] Neklyudov, Kirill, et al. "A Computational Framework for Solving Wasserstein Lagrangian Flows." Forty-first International Conference on Machine Learning.

**Limitations:**

The paper does not include a dedicated **Limitations** section. One potential limitation concerns the computation of $\nabla g_k$, which appears to require accurate estimation of the marginal densities. In practice, density estimation is challenging and may introduce significant approximation error.

Another limitation is the apparent complexity of the approach in terms of hyperparameter tuning and implementation details. The method seems to involve multiple design choices and sensitive parameters. A clearer discussion of these aspects is needed.

**Strengths And Weaknesses:**

**Strengths**.

The paper proposes a new method for trajectory inference that may serve as a valuable addition to existing approaches, particularly due to its simulation-free objective for multimarginal Flow Matching. The method can be viewed as a generalization of Multimarginal Flow Matching (MMFM) [1], where OT-inspired interpolation is used instead of the empirical cubic splines proposed in [1].

The exposition in Section 2 (Background) is clear and well written. Most of the “Dynamic OT with potentials” section (Section 3), together with the relevant supplementary material in the appendix, is presented in a structured and technically sound manner.

**Weaknesses**.

**1. Motivation, Theory, and Exposition**

Although Sections 2 and 3 provide a clear introduction, the later sections lack clarity and rigor.

- **Problem statement.** The paper does not explicitly and rigorously formulate the mathematical problem it aims to solve. In particular, it is not clearly stated that $\mu_{t_k}$ represents the ground-truth marginals whose dynamics the model aims to approximate. This weakens the motivation.

- **Vlasov-type potential.** The motivation for introducing a Vlasov-type potential (i.e., a potential depending on the full density $\rho_t$) is not sufficiently justified. In Section 2.3, Vlasov potentials are mentioned, but Equation (13) is unclear: the left-hand side $V(x,t)$ depends on $x$, whereas the right-hand side appears not to. This inconsistency requires clarification.

- **Derivations in Section 4.** While the idea of integrating second-order constraints in Equation (17) to obtain the velocity (18) and trajectories (19) is intuitive, the subsequent derivations become difficult to follow. The logical connection between equations is not sufficiently explained. For example:

    - Equation (23) appears to contain a possible misprint when compared to Equation (22).

    - Lemma 4.1 and Section 4.2 lack detailed explanation.

    - It is unclear how the proposed method should be implemented in practice and why it is expected to work.

Overall, Section 4 requires a substantial revision to improve clarity, rigor.

**2. Experiments and Related Work**

The paper lacks citations to several important works on trajectory inference and related frameworks. See the relevant papers and references therein, including:

- Metric Flow Matching [2]
- Lagrangian-based framework [3]
- JKO-based approaches [4]
- Unbalanced Schrödinger Bridge framework [5]

The paper does not clearly position itself within the broader trajectory inference literature.

Moreover:

- The authors adopt a **leave-two-out** evaluation protocol, whereas the standard practice in the above-mentioned works is **leave-one-out**. This choice makes direct comparison difficult and raises concerns about the fairness and comparability of the results.

- The paper does not provide comparisons with classical TrajectoryNet [6] and OT-based CFM baseline [7]. Therefore, it remains unclear whether the proposed modifications lead to improvements in performance.

**Minor Comments**.

- There appears to be an incorrect equation reference above Equation 59 in the appendix.
- Figure 2 may contain a mistake in the legend, as it includes two entries labeled “Intermediate.”

**Conclusion**

At its current stage, it is difficult to clearly understand the core contribution of the method described in Section 4. The lack of rigorous problem formulation and insufficient comparisons with related methods make it unclear why the proposed approach should perform better than existing alternatives. Section 4 requires major revision, and the problem statement should be explicitly and formally presented in the paper.

[1] Rohbeck, Martin, et al. "Modeling complex system dynamics with flow matching across time and conditions." _The Thirteenth International Conference on Learning Representations_. 2025.

[2] Kapusniak, Kacper, et al. "Metric flow matching for smooth interpolations on the data manifold." Advances in Neural Information Processing Systems 37 (2024): 135011-135042.

[3] Neklyudov, Kirill, et al. "A Computational Framework for Solving Wasserstein Lagrangian Flows." Forty-first International Conference on Machine Learning.

[4] Persiianov, Mikhail, et al. "Learning of population dynamics: Inverse optimization meets jko scheme." The Fourteenth International Conference on Learning Representations.

[5] Wang, Dongyi, et al. "Joint Velocity-Growth Flow Matching for Single-Cell Dynamics Modeling." The Thirty-ninth Annual Conference on Neural Information Processing Systems.

[6] Tong, Alexander, et al. "Trajectorynet: A dynamic optimal transport network for modeling cellular dynamics." _International conference on machine learning_. PMLR, 2020.

[7] Tong, Alexander, et al. "Improving and generalizing flow-based generative models with minibatch optimal transport." _Transactions on Machine Learning Research_.

---

> ### Author Rebuttal · Authors · 2026-03-31
>
> We thank the reviewer for the positive comments and constructive feedback. We address the key concerns below.
> # Related work
> We will significantly expand the related-work discussion to clearly outline the contributions of OTP-FM with respect to the rich trajectory-inference literature.
> We summarize the salient points below and include new experimental results to enable comparisons.
>
> Trajectory inference methods can broadly be grouped as requiring simulating an underlying ODE/SDE during training (TrajectoryNet, DBSM etc.), which are generally computationally expensive to train (see Experiments), vs. simulation-free methods.
> Flow- and score-matching methods such as CFM pioneered the latter, but mainly focus on trajectories between two endpoint marginals; naively stitching trajectories between multiple marginals, leading to unphysical kinks.
> MMFM and 3MSBM are, to our knowledge, the only simulation-free methods that explicitly target multimarginal interpolations, hence our focus on them.
> Crucially, they prescribe specific smooth interpolations such as cubic splines, which need not describe the dynamics of physical systems (e.g., see response to Reviewer tn6E on Beijing results).
> Instead, **OTP-FM offers multimarginal, simulation-free training with flexible dynamics** through the broad design space of potentials, allowing the data to determine the ideal trajectory interpolation by optimizing over this space.
> MFM and VGFM are important *orthogonal* improvements to CFM, learning the data manifold and modeling growth, resp.; future work will explore their integration.
> Finally, WLF and JKO methods are conceptually related in introducing potentials into dynamic OT, but solve the OT problem over the full dataset using fundamentally different computational paradigms, rather than using it to obtain conditional targets within a FM framework.
> # Experiments
> We perform multiple experiments on the EB (5D and 100D) and CITE (50D) datasets to ensure at least one point of comparison with each method ([Tables 1-2](https://tinyurl.com/2tus2h9u), anonymous upload).
> We also measure training times ([Table 3](https://tinyurl.com/bdfhcezb)) and summarize the timing-performance comparison altogether in [Fig. 1](https://tinyurl.com/389b7p25).
> These comprehensively demonstrate that **OTP-FM**, using our
> recommended $W_2$ potentials (see Limitations),
> **significantly advances the frontier on performance and efficiency** in
> trajectory inference.
> We will add CITE 100D and Multi 50D/100D with 5-seed averages as well in the final paper to complete the benchmark suite.
> # Motivation, theory, exposition
> ## Problem statement
> We will state explicitly our aim to learn physically plausible trajectories whose densities $\rho_t$ align with given empirical marginals $\{\mu_{t_k}\}$ observed at discrete times $t_k$.
> The alignment and physical plausibility are measured quantitatively by the ability to reproduce seen marginals and interpolate unseen marginals, respectively.
>
> ## Motivation for Vlasov-type potentials
> Our response to Reviewer HaPt motivates their use to solve the above problem. We will make explicit in Eq. 13 that V depends on $x$ through $\rho(x)$ and $\mu(x)$.
>
> ## Fixed-point (FP) problem
> We will clarify the exposition around Eq. 23 (we confirm its correctness) and provide a complete derivation.
> The key point is that after conditioning on endpoints and coupling $B$, we solve a *conditional* variational problem and use its velocity as the CFM target, in the same spirit as standard CFM.
> We derive the solution (Eqs. 20-23) in the form of a self-consistent FP relation.
> One training step is then viewed as a Picard-style update to solve the FP problem: evaluate the RHS of Eq. 23 from the current prediction, and regress the discrepancy to zero.
> We draw the analogy in this sense to MeanFlow itself (Sec. 2).
>
> The FP problem view informs important algorithmic choices.
> As the problem is not necessarily contractive (App E.3), we apply the  *homotopy continuation* technique as a curriculum: intuitively, we first learn the easier base velocity (i.e., CFM between the endpoints alone), then gradually introduce the correction term, tracking the solution of a simple problem to the full one (Fig 5) - this *directly motivates the decomposition in Eqs. 20-23*.
> The particular solution to Eq. 23 depends on the potential: for $W_2$ it has a particularly simple closed-form, while MMD and KL require more involved solvers within each training iteration (Sec. 4.3).
> We will add a high-level overview expanding upon these important aspects of the algorithm.
> ## Practical implementation
> We provide pseudocode and extensive training and architectural details in Algo 1 and App. E-F. We will also release the codebase.
> # Limitations
> Please see responses to Reviewers HaPt and tn6E on hyperparameter tuning and limitations, respectively.
>
> ---
>
> We hope these clarifications and added experiments address the reviewer's concerns. We are happy to answer further questions!

---

> > ### Author Rebuttal · Reviewer_nZvx · 2026-04-02
> >
> > I thank the authors for their efforts during the rebuttal. The additional experiments help better position the method within the trajectory inference setting. However, I still have concerns regarding the universality of the approach. As pointed out by **Reviewer ukiM**, the method appears to primarily incorporate physical intuition into conditional flow matching. While it improves training speed and achieves near state-of-the-art performance, I consider the use of a potential-based formulation to be a relatively minor contribution.
> >
> > Furthermore, I am still worried that the explicit mathematical formulation of the problem that the authors solve is never accurately stated in the main text. E.g., it is even non-trivial to understand the phrase “eq. (5) with potential (13)” which seems to be the main formula in the beginning of the main section. The method lacks a clearly defined explicit loss function, making it challenging to assess what objective is being optimized and how the training procedure relates to standard formulations. In particular, Algorithm 1 in Appendix E refers to computing the loss $\mathcal{L}_{\text{OTP-FM}}$, but this quantity seems not to be defined anywhere in the paper.
> >
> > While I became more positive about the empirical performance, I am concerned that the method itself is not clearly understandable and well explained (this is my main remaining concern so far). The authors promised to make the edits to the paper. Unfortunately, the ICML review process does not allow reviewers to assess the revised version of the paper. As a result, I cannot verify whether the updated manuscript will be sufficiently improved (primarily in terms of clear method derivation) to meet the acceptance criteria. Therefore, I do not feel comfortable recommending acceptance. I raise my score to **3 (weak reject)** but select **option (c)** for the author's response.

---

> > > ### Author Response · Authors · 2026-04-06
> > >
> > > We thank the reviewer for raising their score and acknowledging OTP-FM's empirical performance. We address the remaining concerns regarding significance and clarity of method derivation below.
> > >
> > > ## 1. Significance
> > >
> > > We respectfully disagree that the potential formulation is a minor addition of physical intuition.
> > > OTP-FM is a well-motivated generalization of CFM to the multimarginal setting.
> > > The central question is: how should intermediate marginals constrain trajectories?
> > > Standard multimarginal CFM performs piecewise OT, effectively enforcing them as hard density-level constraints, producing unphysical kinks.
> > > OTP-FM relaxes these to smooth constraints — which we interpret as potentials in the action, yielding:
> > > 1) a **principled, physically plausible** generalization of the hard-constraint formulation, recovering piecewise OT (I-CFM / OT-CFM) in the high-strength, highly-localized limit (see below);
> > > 2) **flexible dynamics** via a broad design space, allowing the data to determine the interpolation unlike prescribed approaches such as MMFM; while
> > > 3) retaining **theoretical guarantees**: $W_2$ bounds on marginal alignment controlled by potential strengths and loss (Props. 1–3, response to Reviewers ukiM, HaPt).
> > >
> > > OTP-FM is thus the first simulation-free multimarginal method with flexible, optimization-derived interpolation, with empirically demonstrated SOTA performance and training efficiency across diverse scientific domains.
> > >
> > > ## 2. Method derivation
> > > We present the Problem → Solution → Loss derivation as it will appear, as completely as possible given the spatial constraints, so the clarity of presentation can be directly assessed.
> > >
> > > **Problem.** Given empirical marginals $\\{µ\_{t_k}\\}_{k=0}^{K+1}$ at discrete times $\\{t_k\\}$, learn a velocity field $u_θ$ whose flow 1) induces marginals $ρ_t$ aligned with $\\{µ\_{t_k}\\}$ while 2) describing physically plausible interpolated trajectories, as measured by distributional distances to marginals in and held-out from the training set, respectively.
> > >
> > > **Strategy.** Generalize CFM, which uses conditional-OT regression targets, to use multimarginal conditional *OT+potentials* (OTP) targets, allowing smooth trajectories while retaining simulation-free training.
> > >
> > > **Variational OTP formulation.** We generalize dynamic OT to incorporate soft density-level constraints, interpreted as potential terms in the action:
> > > $$L_{OTP} = \min_{u_t, ρ_t} \int_0^1 \int_{R^d} \left[\tfrac{1}{2}\|u_t\|^2 ρ_t(x) + \sum_k w_k λ_k(t) D_k(ρ_t(x), \mu_{t_k}(x))\right] dxdt,$$
> > > subject to the continuity equation, where $D_k$ is a statistical distance ($W_2$, MMD, or KL), $w_k$ the potential strength, and $λ_k(t)$ the local time dependence.
> > > This makes the reference "Eq. (5) with potential (13)" explicit.
> > > The limit $w_k \to ∞, λ_k \to \delta(t - t_k)$ recovers piecewise OT, while
> > > finite $w_k$ and smooth $λ_k$ represent a principled relaxation with smooth and flexible dynamics.
> > >
> > > **Conditional solution.**
> > > To obtain CFM-like regression targets, we derive exact solutions to the conditional problem, conditioned on endpoints $(x_0, x_1)$ and a minibatch $B$ (Eq. 23):
> > >
> > > $$
> > > X_t(x_0, x_1, B) = \underbrace{x_0 + (x_1 - x_0)t}\_{X_t^{base}}+\underbrace{\sum_k w_k ∇g_k(X\_{t_k}, t_k, B)  \left[I^{(2)}[λ_k] (t) - I^{(2)}[λ_k] (1) t\right]}\_{X_t^{corr}},
> > > $$
> > >
> > > where $g_k$ is the functional derivative of $D_k$ w.r.t. $ρ_t$ and $I^{(2)}[λ_k]$ denotes the double time-integral of $λ_k$.
> > > The base term is the standard CFM straight-line path; the correction term captures the potential-driven deviation.
> > > The dependence of $∇g_k$ on $X_{t_k}$ yields a self-consistent fixed-point problem, motivating the algorithmic choices as in our initial response (homotopy curriculum, base+correction decomposition, $W_2$ recommendation).
> > >
> > > **Training loss.** The explicit OTP-FM objective is MeanFlow regression to this solution:
> > >
> > > $$
> > > \boxed{L_{\text{OTP-FM}}(θ) = \mathbb{E}\_{t_1, t_2, z}\|u_θ(X\_{t_1}, t_1, t_2) - \mathrm{sg}(u\_{tgt})\|^2},\quad u\_{tgt}=\dot{X}\_{t_1} + (t_2 - t_1)(\dot{X}\_{t_1} ∂_x u_θ + ∂\_{t_1} u_θ)
> > > $$
> > >
> > > where $z = (x_0, x_1, B)$, $u\_{tgt}$ is the MeanFlow target constructed from $u_t(x|z)=\dot{X}_t$, $X\_{t_1}$ the conditional position, and $\mathrm{sg}$ denotes stop-gradient.
> > >
> > > ## 3. Structural revision
> > >
> > > We will reorganize Secs. 3–4 to follow the logical sequence above, focusing on the temporally localized conditional problem and leaving the more general causal formulation to the Appendix - also addressing the confusion raised by Reivewer HaPt.
> > > This streamlining and the extra allowed page provides space for the explicit derivations and algorithmic discussion above along with the expanded related work and new experiments.
> > >
> > > ---
> > >
> > > We hope this demonstrates that the method and its derivation are well-defined, how the clarity concern will be addressed concretely, and, combined with the significance of the contribution and acknowledged empirical results, provides the reviewer improved evidence for acceptance.

---

### Official Review · Reviewer_tn6E · 2026-03-09

**Soundness:** 2
**Presentation:** 2
**Significance:** 3
**Originality:** 3
**Overall Recommendation:** 3
**Confidence:** 2

**Summary:**

This paper introduces Optimal Transport-Potential Flow Matching (OTP-FM) to tackle multimarginal trajectory inference. The authors formulate intermediate marginal distributions as time-dependent potential energy terms within the dynamic optimal transport framework. To achieve a tractable, simulation-free training objective, they apply a time-localization approximation, reducing the dynamics to a fixed-point problem. This is solved iteratively using a consistency model (MeanFlow) paired with numerical stabilization techniques. The framework supports various statistical distances for the potentials. Empirically, OTP-FM demonstrates competitive or state-of-the-art interpolation performance over existing methods across synthetic data real-world scientific datasets.

**Compliance With Llm Reviewing Policy:**

Affirmed.

**Final Justification:**

The authors addressed several points, but my primary concern "the inconsistency between the dynamics and the coupling" has not been satisfactorily resolved in the rebuttal. Given the importance of this issue to the work, I am keeping my current score.

**Key Questions For Authors:**

1. **Coupling vs. Dynamics Consistency**: The prior couplings (independent, 2-Wasserstein OT in the paper) are computed independently of the potential fields, yet the resulting continuous dynamics are explicitly driven by these potentials. Is it possible to formulate a coupling mathematically consistent with these potential-driven dynamic assumptions (similar to works extending OT to stochastic [1], unbalanced [2,3], or accelerated dynamics [4])? Addressing this would significantly improve the framework's theoretical cohesion.

2. **Table Formatting Errors**: Could the authors correct the erroneous bolding in Tables 2 and 3? Sub-optimal results are currently bolded in several places, which makes it difficult to accurately assess the empirical comparisons.

3. **Performance Discrepancies**: OTP-FM performs similarly to MMFM and 3MSBM on the EB and GoM datasets, but shows a much more noticeable advantage on the Beijing air dataset. What specific characteristics of the Beijing dataset explain this performance gap?

4. **Hyperparameter Best Practices**: The algorithm introduces a large design space (statistical distances, couplings and other hyperparameters). Could the authors provide a recommended best practice pipeline or heuristic for tuning these settings on a novel dataset to ensure practical usability?

[1] Tong A, Malkin N, Fatras K, et al. Simulation-free schr\" odinger bridges via score and flow matching[J]. arXiv preprint arXiv:2307.03672, 2023.

[2] Wang D, Jiang Y, Zhang Z, et al. Joint velocity-growth flow matching for single-cell dynamics modeling[J]. arXiv preprint arXiv:2505.13413, 2025.

[3] Peng Q, Wang Z, Ying J, et al. WFR-FM: Simulation-Free Dynamic Unbalanced Optimal Transport[J]. arXiv preprint arXiv:2601.06810, 2026.

[4] Yue A, Dong A, Xu H. OAT-FM: Optimal Acceleration Transport for Improved Flow Matching[J]. arXiv preprint arXiv:2509.24936, 2025.

**Limitations:**

No. The authors have not adequately discussed the limitations regarding the method's applicability and scope. Please add a discussion on the boundary conditions and potential failure modes of OTP-FM. Specifically, clarify under what data characteristics the proposed method might break down or be outperformed by baselines like MMFM. Discussing when not to use this method will greatly strengthen its practical value for the community.

**Strengths And Weaknesses:**

**Strengths**

1. **Solid Mathematical Framework (Soundness & Originality)**: The paper presents a rigorous theoretical foundation. Bridging dynamic optimal transport and generative flow matching by formulating intermediate marginals as time-dependent potential energy terms is a well-motivated approach.

2. **Efficiency & Practicality (Significance)**: The method is practical and computationally appealing. The finding that the independent coupling limit ($W_2^\infty$) achieves $O(M)$ complexity while maintaining competitive performance is a useful contribution for large-scale scientific datasets.

**Weaknesses**
1. **Misleading Table Formatting (Presentation)**: A notable presentation issue is that the bold formatting in the experimental tables is sometimes incorrect or inconsistent (Table 2,3). Sub-optimal results are occasionally bolded, which creates confusion when evaluating the empirical results of OTP-FM against baselines. The tables need careful proofreading.

2. **Lack of Discussion on Applicability and Scope (Significance)**: The submission lacks a discussion regarding the boundaries of the proposed method. It is unclear under what specific data distributions, physical conditions, or degrees of sparsity the "potential field" assumption might fail or be inferior to geometric spline-based methods like MMFM.

3. **Hyperparameters (Soundness)**: The framework introduces several hyperparameters (potential strengths, widths, scheduling) without clear tuning heuristics, raising questions about its out-of-the-box usability on novel datasets.

---

> ### Author Rebuttal · Authors · 2026-03-31
>
> We thank the reviewer for their positive comments on the theoretical soundness, originality, and practical efficiency, as well as their constructive feedback. We respond to the key concerns below.
>
> # Coupling vs Dynamics
> We first clarify explicitly that we do *not* claim  to solve the OT + potentials problem over the entire dataset,
> which is what would be required to obtain fully consistent couplings.
> This is as yet unsolved (except for the Gaussian case, Fig. 1), in contrast to the works referenced, who are able to use existing unbalanced / entropic OT algorithms to obtain their couplings.
> We aim to solve the more tractable *conditional* problem, *given* the coupling, to obtain a regression target for FM.
> This is a generalization of CFM, which similarly learns *conditional*-OT paths between two (possibly randomly coupled) samples [1].
> We elaborate upon this and show that our CFM approach is sufficient to control the alignment between the marginals regardless of the coupling in reponse to Reviewer ukiM.
>
> However, we agree that the choice of $\pi_{all}$ is an interesting optimization to explore in future work.
> Two possibilities are 1) to indeed try and solve the marginal variational problem, e.g. using Ref. [2] (which we benchmark against in response to Reviewer nZvx); or 2) a simple rectified flow approach: retraining using the previously learnt couplings, which has proven to produce more optimal flows [3].
>
> [1] Lipman+ 2022, 2210.02747; [2] Kapuśniak+ 2024, 2405.14780; [3] Liu+ 2022, 2209.03003
>
> # Table formatting
> Our reasoning was that mean values within 1 std dev are statistically compatible (at 68% confidence level) so the performance can't be differentiated; however, we understand this is not standard practice and have updated the tables to avoid confusion: [Tables 1-3](https://tinyurl.com/yxta66we), uploaded anonymously.
>
> # Beijing dataset performance
> We add a new figure showing this failure mode of MMFM's interpolation on the Beijing dataset ([Fig 1](https://tinyurl.com/3z6bceca)).
> We see manifestly that **this data is not described by cubic splines**.
> For 3MSBM, we hypothesize that their phase space smoothening cannot capture the long tails of these marginals (we cannot confirm this as we are unable to reproduce their trajectories).
> These results directly support our motivation for flexible dynamics with OTP-FM, rather than the prescriptive approach of MMFM and 3MSBM (see response to Reviewer HaPt).
> We note also that OTP-FM either significantly outperforms (particularly on FGD) or is at least close to other methods on the EB dataset, as well as in our new experiments (response to Reviewer nZvx).
> The GoM dataset is an exception due to its small size (111 samples per marginal), precluding any model from generalizing well.
>
> # Hyperparameter tuning
> We detail this in response to Reviewer HaPt with specific recommendations.
>
> # Limitations
> We will add a clear discussion of limitations. We summarize the main points below due to spatial constraints.
>
>  - **Computing the force $\nabla g$**: This is a clear challenge for KL, MMD, and other potential statistical distances.
>  KL and MMD generally require kernel-based estimates, which provide noisy gradients even for simple distributions (App F.1) and, hence, exhibit worse performance.
>  On the other hand, $W_2$ potentials overcome this with their direct and linear gradients (no density estimation needed), providing a clean and efficient signal.
>  - While matching the intermediate marginals encourages more plausible trajectories, it does not guarantee that interpolated states are physically meaningful, particularly for sparsely-sampled or noisily-labeled ground-truth data (e.g. in disease progression). Methods such as MFM [4] and pixel MeanFlow [5] that explicitly aim to **learn the data manifold** for the entire trajectory may be preferred for stronger guarantees in such cases; as these are orthogonal advancements to that of OTP-FM, future work will explore their integration.
>  - Incorporating **domain-specific inductive biases** such as modeling cell death / growth (e.g. with VGFM [6]), **conditional inference** as in MMFM, and optimizing the couplings (e.g. via Refs. [2-3], as discussed above) are all important areas for future improvement.
>  - Finally, the apparent complexity of our **broad design space** can be seen as a limitation avoided by a more prescriptive approach such as MMFM. However, we point out the drawback of imposing an ad-hoc interpolation strategy and hope to mitigate this complexity by providing tuning guidelines in our response to Reviewer HaPt.
>
> [4] Kapuśniak+ 2024, 2405.14780; [5] Lu+ 2026, 2601.22158; [6] [10] Wang+ 2025, 2505.13413
>
> ---
>
> We believe the added discussion, all of which will be expanded upon in the paper, significantly strengthen the work.
> We kindly ask the reviewer to consider increasing their score if we have addressed their concerns;
> we are happy to answer any further questions!

---

> > ### Author Rebuttal · Reviewer_tn6E · 2026-04-01
> >
> > Thank you for the detailed rebuttal. It resolves the vast majority of my concerns.
> >
> > However, my core concern regarding "Coupling vs. Dynamics" remains unresolved.If the coupling (e.g., $\pi_{all}$) is obtained via standard Wasserstein-2 ($W_2$) optimal transport, its corresponding dynamic is inherently a constant-speed, straight-line trajectory. This seems to directly contradict your motivation to learn "flexible dynamics."
> >
> > What is the theoretical or practical justification for learning complex, non-linear dynamics based on a coupling that inherently assumes straight-line paths?
> >
> > If you can clearly resolve this theoretical inconsistency, I will gladly raise my score.

---

> > > ### Author Response · Authors · 2026-04-06
> > >
> > > We thank the reviewer once again for their thoughtful consideration of our work and the opportunity to clarify this important point in detail as follows.
> > >
> > > ### 1. Clarifying our contribution as independent of the coupling choice
> > >
> > > CFM commonly defines regression targets as the conditional dynamic OT solution between endpoints $x_0, x_1$: a straight line $u_t(x|x_0, x_1) = x_1 - x_0$.
> > > The coupling $\pi(x_0, x_1)$ from which conditioning samples are drawn is an *independent choice* that does not affect CFM's theoretical grounding.
> > >
> > > Our core contribution is to generalize CFM to the *conditional multimarginal* dynamic OT *+ potentials* (OTP) problem.
> > > As with CFM, the coupling is an independent algorithmic choice; there is no *theoretical inconsistency* because our solution (Eqs. 20-23) remains exact and theoretical claims (clarified in response to Reviewer ukiM) are *independent* of the coupling.
> > > The competitive performance of $W_2^∞$ (independent coupling) further empirically demonstrates the robustness of OTP-FM's ability to infer complex dynamics to the coupling choice.
> > > We will revise Secs. 2-4 to make this framing explicit.
> > >
> > > ### 2. Why flexible nonlinear dynamics are retained with OT couplings
> > >
> > > However, as the reviewer points out, there *may appear* to be an inconsistency with our use of static W2 couplings and our aim of "flexible, complex, non-linear dynamics".
> > > We respond to their points directly.
> > >
> > > > "If the coupling is obtained via W2 OT, its  corresponding dynamic is inherently a constant-speed, straight-line trajectory.This seems to directly contradict your motivation to learn 'flexible dynamics.'"
> > >
> > > The **straight-line property is not a property of the $W_2$ coupling** (the static plan $\pi^{OT}$) — it is a property of the **$V = 0$ *dynamic* OT solution** (Corollary 2.2).
> > > The *combination* of the static $W_2$ coupling and $V=0$ dynamics in CFM leads to straighter flows; this may be the source of the perceived inconsistency.
> > >
> > > Crucially, however, the coupling and trajectories are different mathematical objects: $\pi^{OT}$ does *not specify the dynamics*.
> > > We introduce more complex dynamics, that are **not** straight lines, by solving the conditional *OTP* problem, with the Euler-Lagrange solution defined by Eq. 23:
> > > $$
> > > X_t(x_0, x_1, B) = \underbrace{x_0 + (x_1 - x_0)t}\_{X_t^{base}\ (\text{straight line})} + \underbrace{\sum_k w_k \nabla g_k (X\_{t_k}, t_k, B) \left[\mathcal{I}^{(2)}\[\lambda_k\](t) - \mathcal{I}^{(2)}\[\lambda_k\](1) t\right]}\_{X_t^{corr}\ (\text{potential-driven, flexible, nonlinear})}
> > > $$
> > > This shows explicitly that the static coupling's "corresponding straight-line dynamic" under $V = 0$, $X_t^{base}$, is **not** what OTP-FM computes - we introduce the nonlinear correction $X_t^{corr}$.
> > > The **flexibility** we highlight is in shaping the solution and dynamics via potential strength $w_k$, time dependence $λ_k(t)$, and type (different $\nabla g_k$'s), as illustrated in Fig. 1 and ablations in App F.1-F.2.
> > >
> > > ### 3. Why consider OT couplings?
> > >
> > > > "What is the theoretical or practical justification for learning complex, non-linear dynamics based on a coupling  that inherently assumes straight-line paths?"
> > >
> > > We reiterate the coupling does **not** assume straight-line paths — it is a static coupling, not a dynamical assumption; the dynamics of the OTP solution remain nonlinear regardless of the pairing.
> > > Nevertheless, since we are *not* motivated by the "straighter flows" argument of standard CFM, it is important to clarify why we do consider static OT couplings.
> > > There are two strong motivations that we will highlight in the text.
> > >
> > > 1) **Theoretical**: In the limiting case of $W_2$ potentials with δ-function time-dependence, the OTP problem reduces to standard OT between each consecutive pair of marginals.
> > > (Intuitively, each intermediate marginal becomes a hard constraint in the variational problem; we
> > > will add a derivation to the Appendix.)
> > > The exact marginal solution to this is obtained by sequential piecewise OT between each marginal (Ref. [1] App. B) — yielding exactly our $\pi^{OT}$ coupling.
> > > Thus, $\pi^{OT}$ corresponds to **the exact solution to the marginal OTP problem in the δ-function (high-strength, highly localized) potentials limit**.
> > > Since the variational problem varies continuously with the potential strengths, $\pi^{OT}$ remains a well-motivated, tractable choice for the finite-strength regime used in practice.
> > > Jointly optimizing the coupling with the potential-driven dynamics away from this limit is interesting future work.
> > >
> > > 2) Static OT couplings are further **practically** motivated: consistent, nearby conditioning samples provide lower-variance regression targets for CFM, proven in Ref. [2] Lemma 3.2.
> > >
> > > [1] Rohbeck+ 2025 MMFM
> > > [2] Pooladian+ 2023 2304.14772
> > >
> > > ---
> > >
> > > We hope this clearly resolves 1) why there is no theoretical inconsistency, 2) how our practical and theoretical claims remain valid regardless of the coupling, and 3) our justifications for considering OT couplings.

---

### Official Review · Reviewer_HaPt · 2026-03-09

**Soundness:** 2
**Presentation:** 3
**Significance:** 2
**Originality:** 2
**Overall Recommendation:** 4
**Confidence:** 4

**Summary:**

This paper proposes OTP-FM, a multimarginal flow matching framework that incorporates intermediate marginal distributions through time-dependent potential terms derived from a dynamic optimal transport perspective. The goal is to surpass the endpoint-only constraints and learn trajectories that better model the smooth evolution dynamics. The method is evaluated on several real world datasets, where it shows improved performance compared to existing multimarginal baselines.

**Compliance With Llm Reviewing Policy:**

Affirmed.

**Final Justification:**

The author have addressed my major concerns.

**Key Questions For Authors:**

- Why should intermediate marginals be incorporated specifically as a potential term in the action? Is there a stronger theoretical justification for this choice beyond physical intuition?

- The final OTP-FM method relies on several approximations and training heuristics. Which parts of the practical algorithm are theoretically justified by the original dynamic OT formulation, and which are mainly empirical design choices? Can the authors justify them emperically through systematic ablation studies?

- Are there principled guidelines for choosing the potential form in different applications, or is this currently determined mainly by empirical tuning?

- Can authors show the reliance of the method's accuracy and scalability on mini-batch size in the coupling?

I would be happy to adjust my evaluation of this manuscript if the authors could address these questions.

**Limitations:**

No, the paper does not discuss the method's failure mode thoroughly.

**Strengths And Weaknesses:**

- **Strengths**
  -  Multimarginal flow matching is a meaningful setting for temporal snapshot data in areas such as biology, ocean science, and climate.
  - The framework is fairly flexible.
  - The paper is generally well-written.

- **Weaknesses**
  - The the motivation and theoretical grounding seems not strong enough. The paper does not sufficiently justify why marginal distribution should be incorporated specifically as a potential term in the action, or why this is theoretically preferable. At present, the argument is more a physics-inspired modeling intuition than a well-supported theoretical principle.
  - There is also a noticeable gap between the theory and the final algorithm. The theoretical development starts from dynamic OT with potentials, but the final OTP-FM method is obtained only after several approximations and engineering choices. Its trainability and stability seem to depend substantially on these additional mechanisms, rather than following directly from the original theory. In this sense, the theory mainly provides motivation and a conceptual framework, rather than a rigorous foundation for the correctness and stability of the final method.
  - Relatedly, although the paper presents a broad design space over different potentials, distance functions, kernels, and strengths, the experiments suggest that the most effective and stable choices are mainly $W_2^2$ and $W_2^\infty$, while MMD and KLD are less competitive. The paper does not yet explain why some potentials work better than others, or when each choice should be preferred.
  - The evidence on scalability and computational cost is still limited. The implementation uses minibatch OT, but the paper does not systematically study how this approximation affects performance or stability. This is important for large-scale scientific data, since the practical value of the method should not rely on the assumption that global couplings can be precomputed.

---

> ### Author Rebuttal · Authors · 2026-03-31
>
> We thank the reviewer for their positive and constructive feedback.
> We respond to the concerns below.
>
> # Motivation for marginals as potentials
> We provide theoretical justification for the W2 potentials in response to Reviewer ukiM: their strength directly controls alignment with intermediate marginals.
> They are also motivated by the naive alternative of stitching trajectories piecewise between marginals (as in CFM): this is the special case of OTP-FM with δ-function $W_2^2$/$W_2^∞$ potentials (OT-CFM/I-CFM)!
> This clarifies that such "potentials" emerge organically to define the multimarginal task; indeed, our use of the term is physical intuition for merely a density-level constraint term in the variational problem.
> OTP-FM is naturally motivated as a softening of the δ constraint to smooth the unphysical kinks in the naive approach.
> It yields a broad design space for the dynamics, which we argue is an advantage over methods such as MMFM and 3MSBM that prescribe a single ad-hoc smoothing strategy (e.g. cubic splines) that need not describe the physical system.
> Instead, the general form of the potentials minimizes this bias, effectively allowing the data to determine the optimal interpolation via optimization over this space.
>
> # Theory vs. algorithm
> In Sec. 4, the potentials are 1) restricted to be linear in the statistical distance D, eliminating the need to compute D itself, and 2) temporally localized, i.e. forces depend on positions at $t_k$ only, enabling efficient simulation-free training.
> The latter is the only approximation with respect to Sec. 3, but still yields a valid variational problem; in fact, the W2 bounds in response to Reviewer ukiM are based on the Sec. 4 conditional formulation.
> Fig. 1 further demonstrates identical qualitative behaviour from both forms.
> Thus, **we do not lose theoretical or practical justification** from these simplifications.
>
> The second point we will clarify is that we do *not* claim to solve the full OT-plus-potentials problem over the marginal distributions.
> We solve the *conditional* problem of Sec. 4.1 given endpoints and minibatch B, using finite-sample estimators of the forces ∇g.
> Eq. 23 defines the **exact** solution to this problem, analogous to standard CFM, which similarly solves *conditional*-OT between endpoints to obtain its regression target; we generalize this to the multimarginal case.
>
> # Empirical choices & hyperparameter recommendations
> While discussed in Sec. 4, we acknowledge the current paper does not give sufficiently clear limitations and recommendations regarding different hyperparameters;
> we summarize these below and will include them in the text and codebase to **easily enable out-of-the-box training**.
>
> **Choice of potentials & estimating ∇g**:
> This is the main ablation study of the work, and based on the results and discussion in Sec. 4.3 we clearly recommend the W2 potentials, as:
> 1) Calculating ∇g is $O(|B|)$ rather than quadratic;
> 2) Their finite-sample ∇g estimates provide a much cleaner training signal, directly toward ground-truth samples, than the higher-variance kernel-based estimates of MMD and KL (likely cause of their poorer performance); and
> 3) Their FP problem has a closed-form solution, whereas MMD and KL require iterative solvers within a single training iteration for stable training.
>
> **Choice of joint distribution $π_{all}$**:
> We test both pre-computed OT couplings across the whole dataset and random couplings.
> (Neither is an approximation in our formulation because we solve the problem *conditioned on* $B\sim π_{all}$; furthermore, the W2 bounds above are valid for both.)
> We do **not rely on minibatch OT**, which the reviewer correctly notes has limited scalability.
> (We mention it as an option but find the overhead impractical even on small datasets.)
> Instead, our recommendation is to use pre-computed OT couplings when feasible and random couplings otherwise.
> Empirically, the gap between $W_2^2$ and $W_2^∞$ narrows on the larger / higher-D datasets, suggesting that $W_2^∞$ scales well in both efficiency and performance (consistent with SOTA image models like MeanFlow using random couplings).
>
> **Solving the fixed-point problem**:
> We apply homotopy continuation algorithmically as a sigmoid training curriculum.
> App. E.4 justifies this theoretically, and a new EB ablation ([Table 1](https://tinyurl.com/47tnpekj), anon. upload) supports it over no or linear curricula.
>
> **Potential configurations**:
> We argue above why flexibility in this area is an advantage of OTP-FM; however, a strong baseline recommendation is equally-spaced Gaussian potentials of strength 500 (optimal on EB and close on the others).
> Table 1 above and Table 5 (App F.2) serve as references for sensitivity to curriculum and potential hyperparameters for further optimization if desired, with strength the most impactful.
>
> # Limitations
> See response to Reviewer tn6E.
>
> ---
> We hope these clarifications address the reviewer's concerns and are happy to answer further!

---

> > ### Author Rebuttal · Reviewer_HaPt · 2026-04-03
> >
> > I appreciate the authors' efforts in the rebuttal. I still have the following remaining questions:
> >
> > 1. While I acknowledge the authors' statement that the algorithm is inspired rather than dicretly solving the OT+Potential framework, its theoretical rigor needs further justification. Specifically, please provide the explicit statement and proof regarding the error bound $W_2^2(p_{t_k}^\theta, \mu_{t_k})$ controlled by $L_{\text{OTP-FM}}$, as mentioned in the response to reviewer ukiM. This result is essential to bridge the gap between the training objective and the accuracy of the inferred trajectory.
> >
> > 2. I appreciate the extended explanation regarding empirical choices. To better understand the algorithm's implementation robustness, could you provide a theoretical analysis of the fixed-point iteration’s convergence and rate? Additionally, please clarify the computational complexity involved in the OT coupling (especially if pre-computed as a full OT coupling) calculation. To assess its practical scalability, I suggest evaluating the algorithm on a larger-scale benchmark like CITE [1], which is widely recognized in the field. For fairness, I also suggest comparing the running time of OTP-FM (including the running time solving OT coupling) with MMFM and 3MSBM.
> >
> > [1]Lance, C., Luecken, M. D., Burkhardt, D. B., Cannoodt, R., Rautenstrauch, P., Laddach, A., ... & Theis, F. J. (2022). Multimodal single cell data integration challenge: results and lessons learned. BioRxiv, 2022-04.

---

> > > ### Author Response · Authors · 2026-04-06
> > >
> > > Thank you to the reviewer for this opportunity to further improve and clarify these important points.
> > > We provide as much detail as possible within the spatial constraints and will add complete derivations to the paper.
> > >
> > > ## 1. Direct $W_2(p^θ_{t_k},\mu_{t_k})$ bound
> > >
> > > **Prop 3.** In the case of the $W_2^{2/\infty}$ potentials, with strength scale $w>0$ and away from singular potential configurations, and an Eulerian or Lagrangian self-distillation (ESD or LSD, see App. C) OTP-FM training objective $L_{\text{OTP-FM}}(θ)$,
> > > there exist constants $C_k,D_k$ independent of $w$ such that for each intermediate time $t_k$,
> > > $$
> > > W_2(p^θ_{t_k},\mu_{t_k}) \le D_k\sqrt{L_{\text{OTP-FM}}(θ)} + \sqrt{C_k} / w,
> > > $$
> > > where $p_t^θ$ is the marginal generated by the learnt model and $\\{\mu_{t_k}\\}_{k=1}^K$ are the ground-truth intermediate marginals.
> > >
> > > **Proof sketch**: By the triangle inequality,
> > > $$
> > > W_2(p^θ_{t_k},\mu_{t_k}) \le W_2(p^θ_{t_k},p_{t_k}) + W_2(p_{t_k},\mu_{t_k}),
> > > $$
> > > where $p_t$ is the marginal induced by the conditional OT+potentials velocity (Eqs. 20-22).
> > > We derive the bound $C_k / w^2$ for the square of the second term on the RHS in Prop. 2 (response to Reviewer ukiM).
> > > For the first term, we use the endpoint $W_2$ bounds for ESD / LSD from Ref. [1] Prop. 2.4, restricting the flow-map learning problem to $[0, t_k]$:
> > > $$W_2(p^θ_{t_k},p_{t_k})\le D_k \sqrt{L_{\text{OTP-FM}}(θ)},$$
> > > where $D_k$ is a constant depending on the choice of ESD vs LSD and Lipschitz constant of the learned velocity (see [1]).
> > > Combining these two bounds thus yields Prop. 3.
> > > The MeanFlow loss used in the main text is an instantiation of ESD (App. C.2) so this result directly applies.
> > >
> > > **Takeaway**: The alignment between the learnt and ground-truth intermediate marginals is controlled by the OTP-FM training loss and potential strengths.
> > >
> > > [1] Boffi+ 2025, 2505.18825
> > >
> > > ## 2. Theoretical analysis of fixed-point (FP) iteration convergence
> > >
> > > For $W_2$ potentials, the FP equations reduce to a linear system that we solve directly by matrix inversion (Sec. 4.3 & App. E.4), avoiding any FP iterations (the sigmoid curriculum is simply a practical training optimization in this case that empirically improves results).
> > >
> > > ---
> > > For MMD/KL, we extend the discussion in App E.3.
> > > Let $X_T=[X_{t_1},\ldots,X_{t_K}]$ be the matrix of intermediate sample positions, and write Eq. 24 as the map $X_T^{(m+1)}=F_α(X_T^{(m)})$, where
> > > $$
> > > F_α(X_T)_i=X\_{t_i}^{base}+α\sum_k A\_{ik}∇g_k(X\_{t_k}),
> > > $$
> > >
> > > with $A_{ik}=w_k [I^{(2)}[λ_k] (t_i)-I^{(2)} [λ_k] (1)t_i]$.
> > > If each $∇g_k$ is Lipschitz continuous with constant $L_k$, then the norm of the Jacobian of $F_α$ satisfies
> > > $$\|J F_α\|\le α \max_i \sum_k |A_{ik}|L_k \equiv L_F,$$
> > > defining a Lipschitz constant $L_F$ for $F_α$.
> > > By the Banach FP theorem, the **FP iterations are contractive, and therefore linearly convergent with rate bounded by $L_F$, if $L_F < 1$**.
> > >
> > > We can derive $L_k$ explicitly for the gradient estimators considered from the Jacobian of the expressions in Table 1 of the text: MMD with an RBF kernel of bandwidth $\sigma_k$, $L_k^{MMD}\le 4/\sigma_k^2$; and KL with a KDE score estimator, $L_k^{KL}\lesssim \sup_x\|J∇\log\hat\rho_{t_k}(x)\|+\sup_x\|J∇\log\hat\mu_{t_k}(x)\|$, which depends on the KDE bandwidth and diverges in low-density regions.
> > >
> > > **Takeaway**: Intuitively, narrower kernels lead to slower convergence, and KL in particular is less stable for sparse data.
> > > Convergence is also slower and less stable for higher potential strengths $w_k$ (higher $L_F$).
> > > This matches our empirical findings and motivates homotopy continuation (gradually increasing α) and damped Anderson updates for KL and MMD to improve stability, though **we recommend $W_2$ potentials for efficient, straightforward training**, avoiding these complexities altogether.
> > > We will add complete derivations to App. E.
> > >
> > > ## 3. Computational complexity of OT coupling and benchmarking
> > >
> > > We *only* ever precompute it across the entire dataset. We use the exact network simplex method of $O(N^3)$ complexity.
> > >
> > > We added results on CITE in response to Reviewer nZvx ([Table 1](https://tinyurl.com/2tus2h9u)), benchmarked timing ([Table 3](https://tinyurl.com/bdfhcezb)), and summarized the **timing-performance** comparison in [Fig. 1](https://tinyurl.com/389b7p25). **OTP-FM leads in both**.
> > >
> > > The OT precomputation for EB and CITE takes 5.7s and 13.5s (AMD EPYC 7R13 CPU wall time), respectively - a small fraction of the training time, which does **not** affect the relative timing rankings (though we will update Table 3 to include this).
> > > MMFM *also* uses precomputed OT couplings and is 1.5-4x slower than OTP-FM, while our 3MSBM training took 17 hours on EB (500x slower).
> > >
> > > Finally, while the OT computation has been efficient for all benchmarks so far, we recognize it may not be feasible for significantly larger datasets; this is why we ablate with random couplings ($W_2^\infty$), with no precompute cost, that show competitive performance while offering efficient scalability.

---

### Official Review · Reviewer_ukiM · 2026-03-11

**Soundness:** 3
**Presentation:** 3
**Significance:** 3
**Originality:** 3
**Overall Recommendation:** 4
**Confidence:** 4

**Summary:**

This paper proposes OT Potential Flow Matching (OTP-FM) for learning flow matching models under intermediate marginal distribution constraints. To address this problem, the authors introduce an additional potential term based on the distance to intermediate-time marginal distributions, and use a MeanFlow-style approach to learn the corresponding flow map. Building on this formulation, the paper develops a conditional method based on temporal marginals together with a practical training algorithm. The effectiveness of OTP-FM is demonstrated on several benchmarks, including EB, GoM, and Beijing Air Quality. The method shows strong empirical performance across four different choices of distance-based potentials, while incurring little additional computational cost. Overall, I find the paper strong and practically valuable, although some of its theoretical claims and experimental validation could be sharpened.

**Compliance With Llm Reviewing Policy:**

Affirmed.

**Final Justification:**

My concerns have been resolved, and I will maintain my recommendation while increasing my confidence.

**Key Questions For Authors:**

I do not have additional questions beyond those already listed in the weaknesses above.

**Limitations:**

yes

**Strengths And Weaknesses:**

### **Strengths**
1. The paper is well written and easy to follow.
The discussions of the distance choices, methodological pipeline, and numerical implementation are detailed and generally clear.

2. The proposed method is novel.
In particular, the potential-based formulation, as well as the use of MeanFlow and a conditional construction to improve efficiency, are interesting and technically meaningful.

3. The empirical results are strong.
With only limited additional computational overhead, the proposed method outperforms MMFM and 3MSBM on most metrics across multiple benchmarks.

### **Weaknesses**
1.  **Insufficient Validation of Training Agnosticism:** In Line 118, the authors state that “OTP-FM is agnostic to the particular training procedure.” However, the supporting evidence appears limited; as far as I can tell, this claim is only illustrated in the appendix using a Gaussian toy example. This is not sufficient to convincingly support such a general statement. Could the authors provide additional ablation studies on real benchmarks, for example in a format similar to Tables 2–4?

2. **Theoretical Equivalence:** The MMFM paper provides a theoretical guarantee by proving the equivalence of the loss gradients/objectives (i.e., $\nabla\mathcal{L} _ {MMFM} = \nabla\mathcal{L}_{FM}$). Does the proposed OTP-FM method possess a similar theoretical property or equivalence bound? A brief discussion or proof regarding the loss landscape equivalence would greatly strengthen the theoretical foundation of this work.

3. **Minor comments:**

- At Line 198, “satisifies” should be corrected to “satisfies”.

- Figure 3 appears to be missing a ground-truth panel.

- I would recommend avoiding vector graphics for figures with a large number of point clouds and line segments, as large vectorized figures can slow down rendering and significantly reduce readability.

---

> ### Author Rebuttal · Authors · 2026-03-31
>
> We thank the reviewer for their positive comments on the presentation, novelty, and results, as well as their constructive feedback. We address the key concerns below.
>
> # Training agnosticism
>
> We have performed additional experiments on the three real-world datasets using the Lagrangian self-distillation (LSD) [1] and new "improved MeanFlow" (IMF) consistency model formulations [2], using the $W_2^{2/\infty}$ potentials and identical hyperparameters ([Tables 1-3](https://tinyurl.com/3b6ue57u), uploaded anonymously).
> Overall, we observe **robust performance across training procedures**, with a modest average improvement using IMF.
>
> [1] Boffi+ 2025, arXiv:2505.18825; [2] Geng+ 2026, 2512.02012
>
>
> # Theoretical results
>
> We strengthen our theoretical grounding with the following propositions.
>
> **Prop. 1 (Conditional-to-marginal FM equivalence).** Let $z=(x_{t_0},x_{t_1},B)\sim q(z)$ and let $X_t(z)$ be the OTP-FM conditional trajectory with conditional velocity $u_t(x\mid z)=\dot X_t(z)$ (Eqs. 20-23). Define a conditional path $p_t(x\mid z)=\mathcal N(X_t(z),\sigma^2 I)$, inducing the marginal $p_t(x)=\int p_t(x\mid z)q(z)\,dz$.
> Then the OTP-FM conditional regression loss $L_{\mathrm{OTP\text{-}FM}}(\theta)$ has the same parameter gradient as the marginal FM loss on the induced marginal velocity $u_t(x)=\int u_t(x\mid z)\,p_t(z\mid x)\,dz$, i.e.
> $$
> \nabla_\theta L_{\mathrm{OTP\text{-}FM}}(\theta)=\nabla_\theta L_{\mathrm{FM}}(\theta).
> $$
>
> **Proof sketch.** This is exactly the conditional-to-marginal gradient equivalence proved in Theorem 3.2 of [3], applied with the OTP-FM conditioning variable $z$.
>
> **Takeaway.** As in CFM, OTP-FM indeed learns the marginal $u_t(x)$ induced by the conditional $u_t(x|z)$; the difference is that our conditional trajectories solve *dynamic-OT with middle-marginal potentials* (Eq. 20), not the standard endpoint-only conditional-OT problem [4].
>
> ---
> We know the corresponding induced $p_t$ matches the endpoint GT marginals $\mu_0,\mu_1$ by construction (Sec. 2 and [3, 4]). We next show that OTP-FM with $W_2^{2/\infty}$ potentials also controls the $W_2$ distance between the *intermediate* induced and GT marginals, $\{p_{t_k}\}$ and $\{\mu_{t_k}\}$, resp..
>
>
> **Prop. 2 (Informal).** In the case of the $W_2^{2/\infty}$ potentials, of strength scale $w > 0$ and away from singular potential configurations, the OTP-FM dynamics imply
> $$
> W^2_2(p_{t_k},\mu_{t_k})\le C_k / w^2,
> $$
> where $C_k$ is a problem-dependent constant independent of $w$.
>
> **Proof sketch.**
> Let $X_T = [X_{t_1},\dots,X_{t_K}]$ be a matrix of positions along a
> conditional trajectory (Eq. 23) at the intermediate marginal times,
> $X_T^{base}$ positions along the corresponding base path, and $x_T=[x_
> {t_1},\dots,x_{t_{K}}]$ samples from the GT marginals, either sampled randomly for the $W_2^\infty$ potential or OT-coupled to the $X_{t_k}$ for $W_2^2$. Then,
> $$
> W^2_2(p_{t_k},\mu_{t_k})\le^{(i)} \mathbb E\|X_{t_k} - x_{t_k}\|^2 =^{(ii)} \|P_k (I - A)^{-1}\|^2 \mathbb E\|X_T^{base} - x_T\|^2,
> $$
> where $P_k$ is the projection operator, $A$ is the matrix in Eq. 82, and the expectation is over the joint distribution over $(X_{t_k}, x_{t_k})$.
> $(i)$ thus follows from the definition of $W_2$, and $(ii)$ uses the closed-form solution to Eq. 23 for the linear gradients of the $W_2^{2/\infty}$ potentials (App. E.4).
> $A$ scales linearly with $w$, and the expectation term is independent of it for the random or endpoint-fixed couplings we consider (Sec. 4.3); thus, we obtain Prop. 2, assuming invertibility of $I - A$ (i.e., away from singular potential configurations).
>
>
> **Takeaway.** The $W_2$-bound is controlled by the potential strengths, supporting the intuition that stronger potentials "pull" trajectories towards the GT marginals.
> Analogous bounds are harder to derive for the KL and MMD potentials, consistent with their poorer performance.
> Finally, we note Prop. 2 can be combined with Wasserstein bounds from Ref. [1] between the *learnt* $p^\theta_t$ and induced marginals $p_t$, to obtain a direct bound on $W^2_2(p^\theta_{t_k},\mu_{t_k})$ in terms of $L_{\mathrm{OTP\text{-}FM}}$.
>
> ## Summary
>
> Taken together, these results show:
> 1. **Regression against the conditional OTP-FM velocity ($L_{\mathrm{OTP\text{-}FM}}$) is gradient-equivalent to regressing the induced marginal velocities** (as in CFM and MMFM); and
> 2. The **$W_2$ bound between these marginals and the intermediate ground-truth marginals is controlled by the potential strengths** for W2-type potentials.
>
> We will add complete derivations to the Appendix.
>
> [3] Tong+ 2023, 2302.00482; [4] Lipman+ 2022, 2210.02747
>
> ---
>
> We believe the additional experimental and theoretical results significantly strengthen the manuscript.
> We kindly request the reviewer to consider increasing their score if we have addressed their concerns, and we are happy to answer any further questions!

---

> > ### Author Rebuttal · Reviewer_ukiM · 2026-04-02
> >
> > Thank you for your detailed rebuttal, which has addressed my concerns. I will raise my confidence to 4.

---

### Decision · Program_Chairs · 2026-04-30

**Decision:**

Accept (regular)

**Comment:**

The paper proposes a new method for flow matching when few intermediate marginals are observed. The idea is to make a couple of simplifying assumptions on the appropriate dynamic OT equations and learn the corresponding flow velocity. Simulations illustrate the efficacy of the methodology.

Overall, the reviews were positive about the problem motivation and the solution proposed. However, some re-work on the presentation is needed as highlighted by some reviews. Quite some reading in between lines may be needed to fully understand the details of the proposal.  Also, some discussion on how much ground is lost by approximating (16) by (17)? this seems to be  a key simplification.

Once these gaps in the presentation are appropriately handed the paper may be more appropriate for a publication. For now, I am recommending an accept with the above apprehensions and will be ok if this paper is dropped against a better alternative.